# The Study of the Extracellular Matrix in Chronic Inflammation: A Way to Prevent Cancer Initiation?

**DOI:** 10.3390/cancers14235903

**Published:** 2022-11-29

**Authors:** Asia Marangio, Andrea Biccari, Edoardo D’Angelo, Francesca Sensi, Gaya Spolverato, Salvatore Pucciarelli, Marco Agostini

**Affiliations:** 1General Surgery 3, Department of Surgery, Oncology and Gastroenterology, University of Padova, Via Giustiniani 2, 35128 Padova, Italy; 2Fondazione Istituto di Ricerca Pediatrica, Città della Speranza, 35129 Padova, Italy; 3Department of Women’s and Children’s Health, University of Padova, 35128 Padova, Italy

**Keywords:** chronic inflammation, cancer, 3D culture model, extracellular matrix, decellularization

## Abstract

**Simple Summary:**

The extracellular matrix (ECM) is a key orchestrator in several diseases’ initiation and progression. Alterations in its components and the remodeling process influence the inflammatory microenvironment (IME), inflammatory response chronicization and promote the passage toward a tumor microenvironment (TME). This review analyzes the influence of ECM alterations and remodeling process in both chronic inflammation and cancer; then, the correlation between the two pathologic conditions is described. Furthermore, some data obtained in cancer research studies with the employment of three-dimensional decellularized ECM models are reported. The final aim is to evaluate the potential utility of these models in the study of chronic inflammation diseases to promptly prevent cancer initiation.

**Abstract:**

Bidirectional communication between cells and their microenvironment has a key function in normal tissue homeostasis, and in disease initiation, progression and a patient’s prognosis, at the very least. The extracellular matrix (ECM), as an element of all tissues and cellular microenvironment, is a frequently overlooked component implicated in the pathogenesis and progression of several diseases. In the inflammatory microenvironment (IME), different alterations resulting from remodeling processes can affect ECM, progressively inducing cancer initiation and the passage toward a tumor microenvironment (TME). Indeed, it has been demonstrated that altered ECM components interact with a variety of surface receptors triggering intracellular signaling that affect cellular pathways in turn. This review aims to support the notion that the ECM and its alterations actively participate in the promotion of chronic inflammation and cancer initiation. In conclusion, some data obtained in cancer research with the employment of decellularized ECM (dECM) models are described. The reported results encourage the application of dECM models to investigate the short circuits contributing to the creation of distinct IME, thus representing a potential tool to avoid the progression toward a malignant lesion.

## 1. Introduction

The past 25 years have seen a massive shift in the approach to understand the biology of solid tumors.

For a long time, researchers have focused on studying individual cancer cells and the mechanisms that lead them to their malignant phenotype. To date, an overwhelming number of research studies are directed towards a more in-depth knowledge of the tumor, seen as a complex ecosystem of cancer cells, ancillary stromal cells and extracellular matrix (ECM) [1,2,3]. Indeed, researchers analyze all the tumor tissue, including cellular and non-cellular components, considering them together as tumor microenvironment (TME).

The ECM is the most abundant TME component that is taking a leading role in cancer research due to its influence on cancer onset, progression, diffusion and treatment response [4,5,6]. This complex structure is composed of various macromolecules including collagens, glycoproteins (such as fibronectin and members of laminin family), proteoglycan and polysaccharides secreted in an organotypic proportion by resident cells of each tissue and organ. Fine-tuned deposition and degradation of the different ECM components confers to the different tissue and organs their peculiar physical and biological properties. All ECM components, assembled together, form a framework which interacts with cells through a bidirectional communication allowed by surface receptors [4]. This connection between cells and ECM components confers to the ECM a pivotal role in regulating cell adhesion, survival, proliferation, migration and responses to growth factors [7]. Contextualized in a tumor scenario, the ECM is involved in tumor initiation, growth and progression.

In the course of tumor progression, the acquisition of mutations in cancer cells and the changing of surrounding stromal cells phenotype impact on ECM, leading to bio-functional and biomechanical changes. In this frame, different studies demonstrated that the interplay of the change of both cells and/or ECM components is responsible for either the containment of tumor growth or leads to tumor progression with a poor prognosis [8,9].

This evidence substantiates the attempts of research communities to integrate ECM architecture in three-dimensional (3D) models employed for pre-clinical research on cancer biology and drug-screening assay. Indeed, using an appropriate system of microenvironment and tumor culture is the first step towards a better understanding of the complex interactions between cancer cells and their surroundings [10].

The increased knowledge regarding tumor-ECM interplay responds to this emerging request to overcome the limitations of bi-dimensional (2D) models and the development of new models for cancer research. Currently, for this purpose, the field of 3D disease models is currently thriving and 3D models which better recapitulate the intricate dynamic of native tumor and its unique microenvironment have been developed [11].

In this context, the development of novel methodologies for tissue decellularization that enables cell removal without affecting the ECM structure is are paving the way for the guided recapitulation of solid tumor ECM composition in a patient-personalized manner [11]. The decellularization process allows the obtainment of the decellularized matrix (dECM), from a patient sample, which is employed as scaffold for 3D models. These models have huge advantages in that they preserve biological activity and a portion of the main structural proteins and soluble factors, allowing for a more accurate tissue reconstruction [12].

Furthermore, dECM could be successfully re-colonized by cells, providing a suitable microenvironment for optimal cell growth, differentiation and function, and could mimic tissue-like constructs in vitro [12,13]. Nevertheless, dECM models need a long and meticulous process to define an appropriate protocol.

Thus, the successful development of 3D patient-derived dECM scaffolds for cancer research provides the grounds to extend the employment of these models to investigate pathological conditions which can predispose to cancer initiation. For this purpose, this review shifts the attention on the correlation between chronic inflammation disease and cancer initiation and progression. Coussens and Werb asserted that chronic inflammation accounts for neoplastic transformation, and upwards of 15% of malignancies worldwide can be attributed to infections [14]. In a later review, Singh et al. estimated that up of 20% of cancers were linked to chronic inflammation caused by chronic infection, tobacco and other environmental factors [15]. Nevertheless, the relationship between inflammation and cancer harkens back over 150 years ago to 1863, when Rudolf Virchow suggested that chronic inflammation from tissue injury stimulates the proliferation of cells leading to cancer [16,17]. The proliferation of cells alone does not cause cancer, but the presence of inflammatory cells, growth factors and DNA-damage-promoting agents contribute to stroma activation, promoting neoplastic risk.

Furthermore, alterations in ECM potentiate or promote the sequential pathologic events that lead to cancer onset. In the course of chronic inflammation, the assaulting agents fail to be removed and the proliferating cells that sustain DNA damage and/or mutagenic assault continue to proliferate in a microenvironment rich in inflammatory cells and growth/survival factors that support their growth [14]. For these reasons, Dvorak in 1986 refers to a tumor as “a wound that do not heal”, discussing how stroma alterations in wound healing predisposes to cancer initiation [18].

Recent studies shed some light on two molecular and cellular pathways which link inflammation and cancer: in the intrinsic pathway, genetic events causing neoplasia initiate the expression of inflammation-related programs that guide the construction of an inflammatory microenvironment, while in the extrinsic pathway, inflammatory conditions facilitate cancer development [15].

Nevertheless, despite the fact that the functional relationship between chronic inflammation and cancer initiation is widely accepted, many of the molecular and cellular mechanisms mediating this relationship remain unknown [14].

This review analyzes the influence of ECM alterations in both TME and in inflammatory microenvironments (IME), highlighting their role in both diseases. Furthermore, some of the successful results obtained in cancer research with the employment of 3D patient-derived dECM scaffolds are reported. In conclusion, the correlation between the chronic inflammation and cancer initiation is discussed. Considering the relevance of ECM in both TME and IME and how the inflammation could trigger cancer initiation, the review’s aim is to reflect on the potentiality of patient-derived dECM models as tools to investigate the triggers of chronic inflammation which increases the risk of cancer initiation and progression, including infection or autoimmune disease (e.g., inflammatory bowel) [15]. This specific employment of inflamed 3D patient-derived dECM scaffolds may improve our knowledge about chronic inflammation resolution and about inflammatory mechanisms that could favor the passage from IME to TME. Furthermore, since the links between cancer and inflammation are assumed to have huge implications for prevention and treatment [19], the improvement of the inflamed repopulated 3D patient-derived dECM model may be useful for adopting new preventive approaches against cancer initiation and progression.

### 1.1. ECM as More Than a Physical Support: ECM Major Components, Properties and Functions

The ECM, typically situated in large intercellular spaces of connective tissue, consists of a voluminous meshwork of fibrous proteins, mainly represented by collagens, glycosaminoglycans (GAGs), proteoglycans (PGs) and matricellular proteins with different physical and biochemical properties. These components are combined between them with different ratio giving rise to distinct structures with different properties and functions. Table 1 summarizes the major ECM components and their functions.

Two basic forms of specialized ECM can be distinguished: basement membranes (BM), a thin network of highly crosslinked glycoproteins, and the loose fibril-like interstitial matrix (IM). BM, produced by epithelial, endothelial and stromal cells, is mainly composed of collagens, laminins, fibronectin (FN) and linker proteins such as nidogen and entactin, which connect collagens with other protein components. This composition makes BM structure more compact and less porous than IM. Indeed, IM, which is produced primarily by stromal cells, is rich in fibrillar collagens, proteoglycans, and various glycoproteins, such as tenascin C and FN. It contributes greatly to the tensile strength of tissues thanks to its charged and hydrated structure [24,27]. Nevertheless, some specialized ECM structures combine features of both forms of ECM, BM and IM, such as the reticular fibers network of secondary lymphoid organs [6,28]. In each existing version, ECM is not merely intercellular filling, but represents a bioactive element with multiple functions related to its physical, biochemical and biomechanical essential properties in cell behavior regulation [24,29]. ECM’s physical properties, molecular architecture, rigidity, porosity, insolubility, spatial arrangement and orientation allow the function of ECM: to shape and maintain the form of tissue and organ integrity. It also functions as a barrier and anchorage site, and has both a negative and positive influence on cell migration [24]. Its biomechanical properties are mainly related to the fibrous protein content, such as type collagen I, which provides tissue with tensile strength and resistance to deformation [30]. 

### 1.2. ECM-Cells Interplay

Cells can sense all mechanical and biochemical properties of ECM through cell surface and transmembrane receptors. These receptors bind cells to the ECM components and transmit mechanical and chemical signals from ECMs to cells. Integrins are typical matrix receptors which mediate ECM and cell communication. Their function is to connect their extracellular domain with matrix molecules and interact with actin cytoskeleton, through their cytoplasmic tails, thus regulating cell adhesion and motility [31,32,33]. Integrins can be activated through two different mechanisms, either inside-out or outside-in signaling [30]. Indeed, biochemical signals coming from intracellular space can induce conformational changes in the integrin extracellular domain. These changes facilitate ECM ligand binding through the promotion of talin and kindlin recruitment to the cytoplasmatic tails. Regarding the outside-in signaling in ECM, ligands bind to the extracellular domain of integrin and contribute to the recruitment of talin and kindlin to their cytoplasmatic tail. Talin molecules connect the actin cytoskeleton to the integrin and the activation signaling induces conformational alterations that lead to the intracellular recruitment of scaffolding proteins, such as focal adhesion kinase (FAK), which promote cell signaling. The integrin-ECM ligand linkage stimulates FAK/SRC complex assembly at the tails of integrins, recruiting various downstream effectors which activate Pl3K/Akt and Ras/MEK/ERK pro-survival signaling [34]. As a consequence, a stiff ECM exhibits an increase of PI3K/Akt and extracellular-regulated kinase (ERK) signaling, recruiting integrin clustering and promoting more cell survival than a softer matrix [35]. Integrin-mediated adhesion to ECM signaling can also regulate the levels of various cyclin-dependent kinases, controlling cycle progression and promoting cell proliferation. Furthermore, integrin signaling can guide cell migration acting on focal adhesion assembly and Rho GTPase signaling. The discoidin domain receptors (DDrs), instead, are a family of transmembrane receptors tyrosine kinase which bind to structural collagens. After autophosphorylation, DDrs create binding sites for signaling molecules, such as Pl3K, ERK and myosin IIA, that act on cell fate. Another family of transmembrane proteoglycans are syndecans, that interact with ECM ligands such as FN and collagen through their extracellular domain [36].

For biomechanical properties, cells can sample them and tune intracellular signaling pathways through a process termed mechano-transduction [37]. As it turns out, the elasticity of ECM (which can range from soft and compliant to stiff and rigid) helps to coordinate how a cell senses and perceives external forces and stimuli [38,39], providing major environmental cues that affect cell behavior [40,41]. Cells use actomyosin contractibility to remodel the ECM and to sense its material properties or stiffness while integrins, DDr receptors and FAKs mediate cells response [30]. For instance, perceiving a stiff ECM, cells exert higher actomyosin force, thus deforming the surrounding matrix and promoting talin-mediated recruitment and retention of scaffolding protein (e.g., vinculin). This process strengthens the integrins and actin cytoskeleton binding and improves adhesion growth and maturation [42]. As a result, fostering the maturation of new focal adhesion, a stiff ECM emphasizes the integrin adhesion signaling. In the opposite situation, cell interaction with a compliant ECM, the actomyosin contractile force applied is lower and insufficient to provoke the talin and vinculin recruitment which is critical for focal adhesion maturation [30].

### 1.3. ECM Deposition

Fibroblasts are the main producers of ECM components, whether in physiological or pathological conditions. These cells, once activated, become myofibroblasts, secrete ECM components, and, exerting contractile functions, thus confer mechanical alterations of three-dimensional spatial topology of the ECM. Myofibroblasts originate from different cell types and combine phenotypic and functional features of both fibroblasts and smooth muscle cells. Their activation is triggered by several pro-inflammatory factors, among which transforming growth factor β (TGF-β) represents the most important one [39]. The balance of activation and the ensuing inactivation of myofibroblasts is essential to keep physiological conditions and to preserve ECM homeostasis, especially during tissue regeneration and wound repair. Under pathological conditions, indeed, sustained inflammatory stimuli and ensuing TGF-β production by immune cells and tumor cells, alter myofibroblasts’ activation/inactivation equilibrium, causing its hyperproliferation and overactivation which are signatures of fibrosis and the regulated stroma formation in cancer [43,44].

### 1.4. Degradation of ECM and MMPs Activity

The ECM components are normally cleaved and degraded by pericellular target-specific remodeling and degrading enzymes. Among them, various proteases such as soluble and membrane-bound metalloproteinases (MMPs), disintegrin and metalloproteinases (ADAMs), disintegrin and metalloproteinase with thrombospondin motifs (ADAMTS), cathepsins, bone morphogenic protein1 and Tolloid-like proteinases, as well as hyaluronindase and heparanase, are mainly involved. MMPs are of outstanding importance in ECM remodeling, and are crucially involved in cancer progression more than the other ECM-degrading enzymes [45]. The MMP family comprises 28 members and, depending on their type, these can have gelatinolytic and collagenolytic activity toward the ECM. MMPs’ proteolytic activity is triggered by an activation cascade through which it is secreted as zymogens (pro-MMPs) without biological activity and kept inactive until the disruption of the interaction between the conserved cysteine in the propetide domain and the zinc ion bound to the catalytic domain. The activation cascade can include endogenous inhibitors, such as the case of MMP-2 which requires a tissue inhibitor of metalloproteinases (TIMP)-2 to be activated by MMP-13. TIMPs are a protein family which, forming a complex with their own N-terminal domain and chelating the catalytic zinc ion in the active center of MMP, function as natural MMPs inhibitors. Nevertheless, TIMPs can also interact with their C-terminal domain with the hemopexin domain of MMPs and thereby activate them [46]. In particular, the formation of a ternary complex of pro-MMP-9 with MMP-3 and TIMP-1 can activate MMP-9, while TIMP-2 contributes to MMP-2 activation, forming a complex with pro-MMP-2 and MMP-14 [47]. TIMPs have also been shown to modulate other biological functions, such as apoptosis, cell growth and angiogenesis. Notably, many cancer types exhibit TIMP dysregulation, which affects ECM integrity and promotes metastatic ability [46].

The several MMPs are stringently regulated from transcriptional to post-translational level to maintain their expression and activity limited in spatio-temporal distribution and activity. This regulation process is frequently lost in many cancers [48]. Nevertheless, metabolic conditions also influence MMPs’ activity, such as in hypoxia conditions. In particular hypoxia-inducible factor (HIF)-1 increases the degrading action of MMP-1 and MMP-9 [49].

MMPs exert different molecular functions. They can cleave the insoluble ECM components into soluble fragments, can activate or inactivate soluble proteins and can release soluble ectodomains of membrane-bound proteins as autocrine or paracrine signals [50]. Among MMPs, MT1-MMP, also known as MMP-14, exhibits the highest value of substrate specificity, cleaving the interstitial collagen I, II and III while activating MMP-2, and can also indirectly promote cleavage type IV [51]. Other membrane-bound MMPs, such as MMP-15, can cleave type I collagen while MMP-16 is able to cleave type III collagen and both can also activate pericellular pro-MMP-2. [52,53]. Nevertheless, MMPs’ activity is not limited to the degradation of ECM components. Indeed, some MMPs’ substrates and some cleaved fragments can regulate cell proliferation, differentiation, apoptosis and, furthermore, chemotaxis, migration and angiogenesis.

Some pericellular molecules are also susceptible to MMPs’ cleavage activity, such as protease inhibitors, such as α 1- anti-chymotrypsin, α 1-proteinase inhibitor, α 2-macroglobulin, plasminogen activator inhibitor-1 (PAI-1), plasmin C1-inhibitor, and serine proteinase inhibitor-E2 (SERPINE2), and furthermore, other molecules, such as cytokines (e.g., pro-IL-1β, pro-IL-8, Motif Chemochine Ligand 5 (CXCL5), CXCL9, CXCL10, CXCL11 precursor, CXCL11 and CXCL12 (stromal cell-derived factor 1—SDF) and growth factors (e.g., pro-TGF-β).

Additionally, several MMPs’ substrates are intracellular proteins involved in transcription, translation and apoptosis processes, such as the case of many cytoskeletal protein nuclear laminins, chaperones, regulators of transcription and translation [53].

### 1.5. ECM Dynamics under Pathological Conditions

Consistent with the numerous cell biological functions in which the ECM participates, ECM remodeling and degradation are processes that need to be tightly regulated. Not surprisingly, pathological conditions, such as inflammation and cancer, and the breaking of the balance of tissue homeostasis of both processes, cause alterations in the composition and properties of ECM. Thus, despite multiple regulatory mechanisms, ECM dynamics can go awry when activities of ECM remodeling proteins are deregulated, resulting in devastating consequences, manifested in various human diseases. Many ECM features are mostly related to development of disease, such as the interconnection between the different properties of ECM. Indeed, when the ECM stiffness increases due to a pathological condition, its biochemical properties change and, vice versa, cells can perceive the variation and changing the kind of force exerted on ECM. This feature explains why stiff linearized cross-linked collagen bundles promote cell migration, whereas the presence of a denser network of stiff cross-linked matrix fibers, instead, impedes migration unless matrix MMPs are simultaneously activated [54]. The reciprocity represents another important feature of cell-ECM interaction and is involved in the development of disease [55]. Cells constantly act on ECM to modify one or more of its properties, creating, breaking down, rearranging and realigning ECM components. Instead, any changes in the ECM as a result of cellular activities will in turn influence adjacent cells and modify their behavior [24,56,57].

## 2. Role of ECM in IME

### 2.1. The Immune Microenvironment and ECM Components Interactions

The immune microenvironment is strongly influenced by ECM features, while many immune type cells have a key role in ECM deposition and remodeling processes, in both homeostasis and pathological conditions.

Inflammation is a multifactorial network of chemical signals which initiate and maintain the host response aimed to heal the injured tissue. The inflammation process involves primary cell players, such as neutrophils, monocyte-macrophages, mast cells (MCs) and T lymphocytes. These cells produce cytokines, chemokines, growth factors and proteases which support the proliferation of epithelial cells, as well as the production and remodeling of ECM by fibroblasts [58].

Fibroblasts have key role in restoring the ECM homeostasis during inflammation and also exhibit a crucial crosstalk with the immune system. Despite the fact that the process through which immune cells mediate the resolution of inflammation is tightly regulated and self-limiting, this can be hampered leading to abnormalities. Indeed, the profile of cytokines and chemokines persisting at the inflammatory site is decisive for the development of chronic diseases which provoke ECM alteration [14]. In inflamed tissue, infiltrating cells release cytokines such as tumor necrosis factor α (TNF-α), interferon gamma (IFN-γ) and TGF-β, which influence both ECM turnover and protease secretion by tissue-resident cells, thus modulating the expression of a wide range of ECM molecules. On the other hand, the aberrant expression of ECM components can influence immune cell activation, differentiation and survival. Therefore, the remodeled ECM of inflamed tissue influences the perpetuation of inflammatory response and its chronicization.

### 2.2. ECM Damaging and Remodeling during Chronic Inflammation

Tissue damage and the alteration of tissue architecture are common features of chronic inflammation disease, such as inflammatory bowel disease (IBD), and are mainly related to immune components’ activity which activate different MMPs [59]. During inflammatory process, MMPs activity increases and promotes ECM degradation, releasing into circulation its cleaved components. Shimshoni et al. in 2020 [60] provided interesting insights into ECM remodeling during inflammation. This study was conducted on murine models of IBD, in which the ECM dynamic was monitored during intestinal inflammation. They conducted a comparative analysis of matrix structure, stiffness, and composition of a healthy ECM, IBD ECM and ECM of an IBD pre-symptomatic state. The results revealed that in a clinical pre-symptomatic state of IBD, the ECM displayed unique signatures distinct not only from the healthy one, but even from the IBD ECM in the course of full-blown-disease pathology [60]. Thus, the analysis of ECM identified unexpected pre-symptomatic states with its own unique ECM, resulting from progressive changes of the ECM features. Furthermore, the study demonstrated that this progressive change in ECM was related to an increased activity of remodeling enzymes, especially basement membrane degrading gelatinases, which are mediated by the sub-clinical infiltration of immune cells bearing remodeling enzymes in the epithelium [60]. Taken together, these data corroborate the role of ECM in influencing not only immune cells, but also the proteolytic action of proteases which, cleaving ECM components, alters its features, thus influencing inflammation progression.

### 2.3. Key Immune Cells Driving ECM Remodeling during Chronic Inflammation

Neutrophils are the first effectors during inflammation recruited by damage-associated molecular patterns (DAMPs) whose concentration can be altered by ECM components or ECM fragments. Sustained neutrophil recruitment incites the production of pro-inflammatory cytokines and chemokines, promoting angiogenesis and degrading the ECM. In the course of inflammation, neutrophil extracellular traps (NETs) are important sources for ECM-degrading enzymes, such as serine proteases, which function to ingest pathogens, as well as to cleave ECM, aiding in migration [61]. Moreover, the serine proteases can improve MMPs’ activity by cleaving the pro-form of these enzymes and triggering their catalytic activity [61,62,63]. Furthermore, neutrophils store MMP-8 and MMP-9 in granules and release them into the inflammatory milieu where they modulate the action of pro-inflammatory cytokines and chemokines, thereby enabling the recruitment of further leukocytes [64]. Notably, an example of the exacerbation of the inflammation mediated by MMPs’ activity is the increase of MMP-1, MMP-8, MMP-9, MMP-10, MMP-12 and MMP-13 expression within the intestine of IBD patients, which is undetectable in a healthy gut [65]. Aside from MMPs’ and serine activity, another discussed group of enzymes is meprins, zinc-dependent proteases which are responsible for the breakdown of ECM proteins, such as type collagen IV, laminin and nidogen [65].

Proceeding along the inflammation process, once spent, neutrophils secrete many chemoattractant cytokines, such as CCL-2 (MCP-1), CCL-3 (MIP1α), CCL-4 (MIP-1β), CCL-5, TSP-1, IL-1, IL-6, and TNF-α [66,67], which recruit monocytes in situ and induce them to differentiate into mature macrophages and dendritic cells. At the initial stage, the stimulation of most macrophages mediated by IFN-γ, TNF-α and Granulocyte-Macrophage Colony-Stimulating Factor (GM-CSF) induces them to polarize toward a pro-inflammatory M1 phenotypic state, which is responsible for the clearance of the inflammation site from pathogens, dead neutrophils and dead tissue. IL-4, IL-10, IL-13 and TGF-β induce, instead, the M2 anti-inflammatory phenotype to encourage wound healing and stimulate fibroblast migration, proliferation and angiogenesis.

Macrophages are a consistent source of MMP-2 and MMP-9 [68,69]. However, unlike neutrophil-released MMP-9, macrophage-produced MMP-9 is bound by TIMP1, which limits its activity [69]. Therefore, macrophages have an intrinsic modulatory influence on MMPs’ function, which is important in the remodeling phase. In chronic inflammation, macrophages retain their pro-inflammatory phenotype, resulting in persistent inflammation, impeding tissue repair and chronicizatation [70]. MCs are another key effector of the innate immune system which are present in almost all tissue to defend the organism against pathogens. Once activated, MCs degranulate and release pre-made chemokines, cytokines, growth factors, histamine and proteases.

MCs exert a fundamental role in ECM scar formation through proteases release [71], and a protective role through pro-inflammatory cytokines which recruit immune cells [72].

MCs have roles in ECM production, processing and degradation, since MCs proteases can degrade collagen [73], FN [74] and laminin [71], and activate the latent forms of MMP-2 and MMP-9 [75].

### 2.4. ECM-Derived Fragments as Modulators of Chemotactic Activity for Immune Cells

ECM bioactive fragments can exhert chemotactic activity for inflammatory cells such as those derived from collagen types I and IV, elastin, fibronectin, laminins, entactin/ nidogen, thrombospondin and hyaluronan [76]. Interesting to note, Senior et al., demonstrated that the chemotaxis dose response curves to ECM fragments can be comparable to those obtained with formyl met-leu-phe (fMLP) and the complement anaphylatoxin C5a which are classic chemoattractants [77].

ECM fragments’ chemotactic activity is mediated by several cell surface receptors, such as the 67-kDa protein on the neutrophil surface, also called high affinity laminin receptor. These receptors mediate chemotaxis to fragments of type IV collagen, laminins and elastin [76]. Nevertheless, the same ECM component can be recognized by more than one receptor involved in stimulating responses and, moreover, the same ECM component can be split into fragments with opposite activities. For instance, neutrophils present several types of receptors for type IV binding molecules, including elastin binding protein (EBP) and L-selectin [78]. Moreover, while these bind with 7S domains of type IV collagen chains, which have neutrophil chemotactic activity via the EBP complex [77], another peptide from the α3 chain of type IV collagen is reported to suppress neutrophil activation [79].

Other evidence about chemotactic function of ECM-derived fragments concerns the N-acetyl Pro-Gly-Pro (PGP) fragments resulting from MMP-9-mediated hydrolysis of collagen occurring during inflammation [80]. These acetyl-PGP fragments, having a structural homology with CXC-chemokine ligand 8 (CLCL8), mimic the same chemotactic effect on neutrophils as demonstrated in a lung inflammation model [81] and, moreover, it is also detectable in bronchoalveolar lavage samples from patients with obstructive pulmonary disease [82,83].

Entactin/nidogen fragments derived from basement membrane component cleavage also exert neutrophil chemotaxis, which is integrin-mediated [84]. Furthermore, degradation products of elastin have also been shown to be chemotactic for monocytes in chronically inflamed lungs, and in particular those which resulted from macrophage elastase MMP12 activity [85,86].

### 2.5. Alterations in ECM Components Affect Immune Cells’ Activity

ECM-derived fragments and alterations of some molecules, besides exerting chemotactic activity, may also activate immune cells, fostering the inflammatory response [87]. These effects are generally mediated by the Toll-like receptors (TLRs) family, which recognize conserved products (such lipopolysaccharides) defined as pathogen-associated patterns (PAMPs), as well as molecular DAMPs [83]. The activation of TLRs triggers innate immune response which, in turn, also influences the adaptive response. Several fragments of the ECM function as activators, such those derived from interstitial matrix, such as tenascin C isoform, the small leucin-rich proteoglycan biglycan [88], fibronectin [89], heparan sulphate [90,91] and HA [92].

In the study of tenascin C, Midwood K. et al. assessed its activation of TLRs in an in vivo study conducted on a murine model of rheumatoid arthritis (RA). In the in vivo model, just as was observed in RA patients, the level of tenascin C was upregulated in synovia, synovial fluid and cartilage. Upregulated tenascin C was observed to interact with TLR4 on macrophages and synovial fibroblasts, inducing pro-inflammatory cytokine production, such as IL-6, TNF and CXCL8. On the contrary, the mice which lacked tenascin C were protected from synovitis that was induced by zyosan (a TRL2 agonist). Thus, in promoting the production of pro-inflammatory cytokines, tenascin C is proposed to maintain inflammation in the joint and to locally propagate the inflammatory response [93]. Similarly, Schaefer L. et al. described pro-inflammatory effects exerted by soluble, non-ECM-bound biglycan in the course of renal inflammation, through the binding of TLR2 and TLR4 on macrophages. This binding increases the macrophage expression of CXCL2 and TNF and also promotes a positive feed-forward mechanism that sustains macrophage infiltration and fosters the inflammatory response [94]. By contrast, in a later study, Schaefer et al. demonstrated that ECM-bound biglycan can function as cytokine sequesters, controlling their concentration and activity, such as the case of TGF-β anti-inflammatory effects [95].

Even HA has been implicated in the activation of TLR2 and TLR4 [87]. The proteolytic degradation of HA results in the generation of low or high molecular weight fragments with different implications in IME and in TME. Indeed, low-molecular-weight HA fragments have been proposed to be increased in inflamed tissue [96], in that they interact with TLR2 and TLR4 on resident immune cells. This interaction stimulates the expression of pro-inflammatory cytokines and chemokines [97,98,99] and promotes interaction between antigen-presenting cells and T cells [100], fostering the inflammatory response. By contrast, high-molecular-weight HA has been proven to exhibit anti-inflammatory effects and promotes the survival of epithelial cells in a model of acute lung injury [92]. 

### 2.6. ECM-Fragments Influence Gene Expression of Inflammatory Cells

The interaction between inflammatory cells and ECM components has effects on inflammatory cell gene expression [76]. Indeed, fragments derived from ECM cleavage stimulate monocytes/macrophages to product cytokines and proteases. For instance, low molecular weight fragments of HA increase the MMP-12, PAI-1 [101,102] and stimulate the production of several cytokines, such as macrophage inflammatory 1- α (MIP-1α), MIP-1β, MCP-1,KC, IL-8, and IL-12 by macrophages [98,103,104], while fragmented fibronectin provokes an increase in monocyte/macrophage secretion of proteases, such as MMP-9/gelatinase B, MMP-12/macrophage elastase and pro-inflammatory cytokines [105,106,107]. Furthermore, peptides containing a specific sequence of laminin-111, Ser-Ile-Lys-Val-Ala-Val (SIKVAV), induce monocytes/macrophages to expression of MMP-9 and urokinase-type (uPA) [108,109]. Similar peptides Ala-Ser-Lys-Val-Lys-Val (ASKVKV), which are derived from laminin α5 chain, are chemotactic for neutrophils and macrophages in vitro, and also promote the expression of MMP-9 and MMP-14/MT1-MMP production by monocytes and macrophages [110,111]. Additionally, microarray analysis revealed that laminin α5-derived fragments may induce the upregulation of many pro-inflammatory cytokines such as TNF-α and one of its receptors, tumor necrosis factor receptor (TNFR)-II [112].

## 3. From Chronic Inflammation to Cancer: The Main Correlations

The inflammation which may promote cancer initiation is induced and exists long before tumor formation [112]. IBD, chronic hepatitis, Helicobacter-induced gastritis or schistosoma-induced bladder inflammation are correlated with an increased risk of colorectal cancer (CRC), liver, stomach and bladder cancers, respectively [113]. Chronic inflammation responds to one of the main recognized predisposing factors to cancer initiation, which is the accumulation of mutations in healthy cells [114]. Notably, inflammatory cells such as neutrophils and macrophages are the major producers of reactive oxygen species (ROS) and reactive nitrogen intermediate (RNI) species, which increase mutagenesis, predisposing to accumulation of mutations in normal tissue [112]. For instance, chronic intestinal inflammation is correlated with the accumulation of mutation in Tp53 or other cancer-related genes in intestinal epithelial cells [115,116,117,118]. Inflammatory cytokines are also correlated with predisposing cancer mutations. IL-22 induces the expression of DNA damage response genes to repair the genotoxic insult caused by inflammation [119]. Other signaling cytokines such as IL-6, TNF-α and IL-1 activate epigenetic machinery in epithelial cells, including DNA histone modification components (Dnmt1, Dnmt3, DOTL1), miRNA and lncRNA, modulating expression levels of oncogenes and tumor suppressors [120]. The consequences of these epigenetic changes have similar effects of inactivating mutation in tumor suppressor and activating mutation in oncogenes [112]. Another correlation between chronic inflammation and cancer is related to stem cells. The inflammatory process triggers the differentiation of post-mitotic epithelia into tumor-initiating stem-like cells [121] and thus provokes damage to the epithelial barrier, exposing the stem cell compartment to environmental carcinogens or to active inflammatory cells producing genotoxic compounds. This aspect assumes great relevance given that, in many cases, stem cells are triggers for cancer initiation as “seeds” for metastatic outgrowth in different secondary organ sites [112].

### 3.1. Main Inflammatory Pathways Correlated with Tumorigenesis and Colitis-Associated Cancer

The inflammatory process, besides contributing to the transformation of malignant clones even allows their outgrowth in tumor mass [112]. Indeed, cytokine receptors signaling in mutated cells offer a great contribution in the induction of pro-survival pathways, promoting the survival rate of mutated clones [122].

Nuclear factor kappa-light-chain-enhancer of activated B cells (NF-kB) represents a protein complex key orchestrator of innate immunity and inflammation, and has emerged as a key tumor promoter [123]. Once activated in inflammatory cells, NF-kB regulates cell cycle mediators (cyclin D1, c-Myc), anti-apoptotic (c-FLIP, survivin, Bcl-XL) and adhesion molecules (ICAM-1, ELAM-1, VCAM-17), proteolytic enzymes such as MMP-9 and uPA and pro-inflammatory cytokines (such as TNF-α, IL-1, and IL-6), which, altogether, contribute to inflammation-related tissue damage and tumor development and progression [124]. NF-kB, indirectly, through TNF-α production which acts as a potent mutagen, contributes to tumor initiation, inducing ROS release and promoting DNA damage [125], while encoding antiapoptotic regulators, ensuring the survival and proliferation of tumorigenic cells [123]. Moreover, NF-kB may trigger tumor initiation and progression, enhancing angiogenesis through the expression of vascular endothelial growth factors (VEGF), cyclooxygenase 2 (COX-2) and IL-8 [126]. NF-kB activation is often correlated with tumor-associated inflammation in colitis-associated colon cancer (CAC) [127]. In fact, CAC represents a classical example of an inflammation-triggered malignancy [14,112,128]. For this purpose, the STAT3/IL-6 pathway, promoting the survival and proliferation of premalignant intestinal epithelial cells, represents another pathway which triggers CAC [129,130]. In particular, STAT3 suppresses the effects of anti-tumor Th1 cytokines, such as IL-12, IFN-γ, and induces secretion of tumorigenic mediators (e.g., cytokines, pro-angiogenic and growth factors) while promoting the expression of corresponding receptors that, in turn, induce a STAT3 mediated immunoregulatory circuit in the TME [131].

STAT3 activation in IBD, mediated through the interaction between IL-6 with its membrane bound receptor (IL6R), enhances the expression of antiapoptotic factors, causing CD4 T cell resistance and thus promoting the perpetuation of chronic intestinal inflammation [132]. An important role in colonic inflammation and tumorigenesis is attributed even to COX-2. Indeed, elevated expression of COX-2 was detected in IBD patients and in colitis-associated neoplastic tissue [133]. COX-2 may trigger tumor development, likely inducing the expression of antiapoptotic factor STAT3 and increasing levels of MMPs, as well as promoting the migration of malignant cells [134]. As demonstrated in colon cancer cell line CACO-2, programmed to constitutively express COX-2, an increased invasiveness compared with the parental CACO-2 was detected, along with the activation of MMP-2 and increased RNA levels for membrane-type 1 matrix metalloproteinase (MT1-MMP) [135].

The effect of COX-2 in IBD and CRC is mediated by Prostaglandin E2 (PGE_2_) acting through specific cell surface receptors (EP), which include four subtypes, EP1, EP2, EP3, EP4, among which the first is proposed as a mediator of PGE_2_ role in colon carcinogenesis [136]. The cancer-promoting effect of PGE_2_ involves CXCL1 and nuclear hormone receptor peroxisome proliferator-activated receptors (PPARs) [137,138]. CXCL1 is a proangiogenic chemokine which sustains microvascular endothelial cell migration and tube formation to support tumor growth and invasion [137]. PPARs are a downstream target of the COX-2/PGE_2_ pathway, and its activation can induce COX-2 expression in colonic cancer cells. Furthermore, COX-2 derived from PGE_2_ contributes to CAC, stimulating macrophages to produce proinflammatory cytokines. Indeed, as Wang found in 2014, the deletion of PPARs can weaken colonic inflammation and CAC development in various animal models, suggesting that PGE_2_ may mediate the correlation between colonic tumorigenesis and chronic inflammation through a self-amplifying loop between PPARs and COX-2/PGE2 [139]. T-helper IL-17-producing cells (Th17) are involved in both pathogenesis IBD [140] and CAC [141,142]. IL-17 fosters intestinal inflammation, stimulating endothelial cells and macrophages to produce cytokines and chemokines, as well as increasing neutrophil recruitment [143]. Th17 cells can also produce other cytokines which are involved in intestinal inflammation, such as IL-21 and IL-22. Th17 action is modulated by IL-23, while TGF-β and IL-6 drive early its differentiation through the expression of key transcription factor retinoic acid receptor-related orphan receptors (RORs). TGF-β and IL-6 also induce IL-23 receptor (IL21R) expression on Th17 cells to mediate the effects of IL-23, such as the stabilization and expansion of the Th17 response [144].

### 3.2. ECM Deposition in TME

In TME, as well as in a physiological context, the major producers of ECM components are stromal cells (Table 2), which, in this scenario, are recruited and orchestrated by tumor cells through their production of pro-fibrotic growth and inflammatory factors, such as TGF-β, fibroblast growth factor (FGF)-2, platelet-derived growth factor (PDGF) and epidermal growth factor (EGF) [145]. Notably, TGF-β that is a key regulator of myofibroblast differentiation, in that within the ECM it is bound to FN, and connected to the fibrillar meshwork of the ECM through the complex TGF-β-binding protein (LTBP) and the latency-associated protein (LAP). Mechanical tension along the ECM fibrils and partial proteolysis of LAP provokes the release and the activation of TGF-β. Once free to act, TGF-β promotes the differentiation of fibroblasts into more contractile so-called cancer-associated fibroblasts (CAFs), which, increasing tension, foster the release of TGF-β1 in an autocatalytic manner [146,147]. CAFs originate predominantly from tissue resident or bone marrow-derived fibroblasts [148], but also from mesenchymal cells and other cells which have undergone epithelial to mesenchymal transition (EMT) [149]. CAFs exert a variety of tumor promoting functions and, producing several growth factors, chemokines, and owing to their different origin attract other cells, such as endothelial and immune cells to tumor mass and orchestrate their joining the TME [148]. Moreover, driven by cancer cells, CAFs remodel ECM and support tumor growth, producing and depositing large numbers of ECM components in a dysregulated manner, altering ECM composition in TME [150,151]. In so doing, CAFs contribute to the complexity and heterogeneity of the stroma in TME. Nevertheless, some subpopulations of CAFs sustain tumor growth in an manner independent of ECM remodeling, for example, promoting cancer stemness or preventing cancer cell recognition by T-cells [152,153].

### 3.3. Tumorigenic Alterations in ECM Composition

ECM dramatically changes in its composition and relative abundance at the primary site of tumor [5]. These alterations also have consequences for ECM biophysical and biochemical properties, which influence tumor development. Indeed, since ECM components can present both tumor-suppressing and tumor-promoting properties, their alterations during the tumorigenic process can promote one aspect or the other [45]. For instance, as Bohaumilitzky et al. reviewed in a study from 2017, HA, depending on its molecular weight, can act as a tumor suppressor or a tumor promoter [165]. The most frequent tumorigenic alteration of ECM is the increased amount of fibrillar collagen [166], which, instead, has a tumor-promoting effect. In physiologic conditions, the collagen fibers which surround the normal epithelial structures are curly and smooth while, during tumor development, a variable number of the fibers progressively become denser, aligned and stiffer. The increase in parallel orientation in collagen fibers increases their density and concentration, stiffening the ECM. In addition, collagen may undergo post-translational modifications inside and outside the cell, enhance protein complexity, alter 3D organization and, also, affecting matrix interaction with other molecules and cellular receptor, influence ECM degradation [167]. These post-translational modifications affect collagen precursors, procollagens, through various mechanisms, such as the hydroxylation of lysin residues or glycosylation. Modified procollagens form triple helices and are further processed extracellularly by proteases to create collagen fibrils [45]. Collagen fibrils are covalently cross-linked by the action of extracellular ECM-modifying enzymes, such as lysyl oxidases (LOX) and LOX-like proteins (LOXLs). Post-translational modifications or dysregulations of LOX and LOXLs’ activity provoke morphological changes which improve tumorigenesis and tumor progression. Indeed, collagen cross-linking and the ensuing increase of ECM stiffness influence tumor cell motility, improve the recruitment of stromal cells and are associated with poor prognosis [168]. Notably, matrix stiffness provokes an increase in tension which induces integrin clustering, phosphorylation and activation of FAK. Once activated, FAK may promote RAS-mediated phosphorylation of ERK, which can control migration, invasion, proliferation and cell differentiation, myosin contraction and the induction of transcription programs [169]. Growing evidence correlates high tissue tension with cancer progression [168] and is often observed in malignant breast cancer [169].

Additionally, the excessive deposition of fibrillar collagen and cross-linkage, inter alia, causing stiffness of ECM, contributes to a fibrotic phenotype termed desmoplasia, which is a key characteristic of many cancers, such as breast cancer and pancreatic ductal adenocarcinoma (PDAC), and is associated with poor prognosis [170,171].

Another altered component in tumor stroma and typical of desmoplastic reaction is FN, especially in its splice variant expressing the extra domains A and B, which contribute to increased ECM stiffness (ED-A and -B) [146]. Reasonably related to its enhanced integrin-mediated contact cells, the ED-A containing FN variant promotes the TGF-β1 release from its LTBP-LAP complex, especially in a mechanical force-dependent manner without any need for partial proteolysis [146,172], thus promoting CAF differentiation [173]. Moreover, the altered deposition of FN promotes TGF-β1 release and improves the EMT [174]. Furthermore, the fibronectin meshwork, which normally binds several growth factors, if altered, allows the delivery of these molecules to proliferating tumor cells [175].

Another notable consequence of ECM stiffness is the induction of hypoxia status in TME. The hypoxic condition causes a restriction of the delivery of oxygen and nutrients to cancer cells, as a consequence of compressing effects on blood vessels [49]. A deficient blood and oxygen supply, high concentration of lactic acid, acidosis, the ensuing TME acidification and altered redox status favor tumor cells, rewiring them to use glycolysis as primary source of energy [176]. This phenomenon is termed the Warburg effect [177]. These metabolic conditions provoke a change in the gene expression pattern of neoplastic cells, such as the overexpression of HIF [160]. HIF acts to reinforce glycolysis, inducing the expression of glucose transporters (GLUT) and glycolytic enzymes. Moreover, HIF cooperates with Myc, a common oncoprotein gene that encodes the glycolytic proteins while increasing mitochondrial metabolism. Sustained hypoxia, indeed, leads to mitochondrial dysfunction [178] and, furthermore, to an abnormal cellular redox status as a result of reactive oxygen and ROS and reactive RNS accumulation. This leads to an alteration in gene expression levels, signal transduction pathways which regulate cancer cell proliferation, invasion and apoptosis. These oxidative molecules can activate all three members of the mitogen-activated protein kinase (MAPK) family, stress-responsible protein kinases including ERK1/2, JNK, and p38 through the oxidative modification of protein tyrosine phosphatases that dephosphorylate MAPKs. The stimulation of MAPK through ROS and RSN accumulation improves the proliferation, migration, and invasion of human breast, liver, prostate, lung, skin, and pancreatic cancer cells [179]. Furthermore, ROS-induced PI3K/Akt signaling plays a key role in the acquisition of cell malignant phenotypes and the survival of cancer cells through the Akt activation or inactivation of tumor suppressors, such as PTEN or PTPs (protein tyrosine phosphatases) [180].

The migration of cancer cells is also improved in hypoxic conditions through MAPK induction and ROS accumulation. Indeed, the MAPK/ERK pathway increases caveolin-1 expression in cancer cells and, through the induction of RhoA/Rho-associated protein kinase 1 (ROCK1), promotes cell contraction and favors cancer cell migration along the pre-existing collagen matrix [49,181].

Thus, alterations in ECM composition and ECM stiffness involve biochemical and biophysical properties in an equal manner, and promote not only the survival and proliferation of cancer cells, but initiate and promote oncogenic transformation, and may influence somatic transformation rate [182,183].

#### MMPs’ Role in Shaping TME

Cancer and stromal cells within the tumor mass, such as neutrophils, macrophages, lymphocytes, MCs, fibroblasts, endothelial cells, pericytes and adipocytes, are the major source of MMPs in TME. All these distinct stromal cells, producing a specific set of metalloproteinases and its inhibitors, contribute to intratumoral proteolytic equilibrium. CAFs, for example, besides producing ECM molecules, also synthesize elevated level of ECM-degrading enzymes such as MMP-1,2,3,11,13,14,19, ADAM9, 10, 12, 15 and 17 and ADAMTS5, as well as TIMP1, 2 and 3 [184].

Notably, TME contains various MMP substrates, including native fibrillar collagens and gelatin, as well as laminin, making MMPs pivotal in TME shaping in various ways [53].

The proteolytical MMPs’ activity and its ECM remodeling play a crucial role in in varying degrees of tumor dissemination, metastasis cascade and the formation of suitable metastatic niches. Accordingly, the expression and activity of many MMPs correlate with tumor progression [185]. Indeed, the EMT process also depends on the MMPs’ activity, which increases the cell’s ability to migrate and infiltrate. MMPs with a documented role in EMT are MMPs-1,2,3,7,9,14 [186] and 28 [65]. In particular, MMP-14 promotes a mesenchymal phenotype, cleaving BM components and E-cadherin [187]. Soluble MMPs-1,3,7,9,10,11,13,26, and 28, and the membrane-type MMPs -14 and -16 are important for the next steps of the metastatic cascade, being involved in cell migration and invasion [53]. ECM remodeling by MMPs is also important for tumor angiogenesis. In particular, MMPs -1,2,3,8,9,10,11,13, and 14 can also regulate angiogenic balance and, vice versa, angiogenic factors such as VEGF, basic fibroblast growth factor (bFGF), TGF-β and α, and angiogenin can induce MMPs’ activity [65]. Another non-negligible aspect of the degradation of ECM components is related to the production of bioactive fragments; these fragments may have pro- or anti-tumorigenic functions independent of the starting full-length ECM component properties. Due to chemokine- or cytokine-like structure, these fragments are termed matrikines [188] and, through their promoting and inhibiting properties, they are involved in balancing the angiogenic switch [104]. Furthermore, excessive MMP activity observed during tumor progression causes the release into circulation of small ECM fragments, which may be indicative to evaluate tumor activity and invasiveness and could be used as biomarkers [4,189].

### 3.4. Tumorigenic ECM and Its Remodeling Influence on Immune Cells within Tumor Mass

Cancer and immune cells’ migration is frequently guided by gradients of surface-bound molecules, such as the ECM, through a phenomenon termed haptotaxis [190]. Given increased expression of fibronectin and collagen in tumor stroma, as previously discussed, and since immune cells encounter the tumor mass on their route of immunological surveillance, it is reasonable to assume that haptotactic response can enhance immune cell invasion in tumor stroma [30]. Nevertheless, cells migrate faster in stiffer substrates and their persistent migration can be directed up a stiffness gradient, through a process termed durotaxis [9], which could explain why stiffer tumors show a higher infiltration of immune cells. Once immune cells meet desmoplastic ECM, they are kept in an immunosuppressive state [191] which fosters tumor progression [192]. Macrophage polarization is critical for the switch from pro-inflammatory to anti-inflammatory signaling. However, an imbalance in this phase contributes to tumorigenesis. Macrophages polarized towards the M1 phenotypic state are associated with tumor-suppressive functions, such as supporting CD8+ [193] cytotoxic T-cell activity. Macrophages polarized towards the M2 state are anti-inflammatory and associated with tumor-promoting functions such as ECM remodeling, angiogenesis, stimulating cancer cell proliferation and metastasis. Indeed, in the later process of inflammation, macrophages are favorable for tumor progression and often contribute into the process. Monocyte-derived macrophages within the tumor stroma, which differentiate in M2 phenotype, are indicated as TAMs. These cells release in tumor stroma several cytokines and interleukins (ILs), among which is IL-10, which inhibits the expression of major histocompatibility complex (MHC) and co-stimulatory molecules inducing immune suppression and TGF-β release [192]. This latter one, instead, attracts Tregs and other cells of adaptive immune system, such as the myeloid suppressor cells, which collaborate to inhibit the attack of CD8+ -T-cells and natural killer (NK) cells to the tumor mass [194,195]. Tregs, in turn, secrete TGF-β, supporting the activation of ECM-tethered TGF-β and thus reinforcing the tumor supportive action [196].

The role of TAMs is well investigated for their interaction with CAFs. TAMs and CAFs can activate and recruit each other in a similar manner to that observed in normal macrophages and tissue fibroblasts. Indeed, CAFs secrete bFGF, inducing M2 polarization, while M2 cells release TGF-β, provoking fibroblast reprogramming to a tumor-promoting CAF state [70]. It has been demonstrated in an orthotopic CRC model that the recruitment of TAMs to the tumor site regulates several ECM-associated genes, which established key characteristics of the TME. The high remodeling activity mediated by tumor cells and CAFs, as previously pronounced, results in the release ECM components which can act as inflammatory stimuli, creating an inflammatory tumor environment [197]. For instance, matrikines may function as DAMPs, such as the bioactive fragments resulting from the ADAMTS proteolytic degradation of versican known as versikine [198]. Hope C. et al., in 2017, demonstrated that high levels of versikine increase tumor infiltration of CD8+ T-cells in colorectal cancer [199], while in 2019, Dhakal et al. demonstrated infiltration in myeloma [200]. In addition, TAMs and tumor-associated neutrophils (TANs) represent a source for ECM-remodeling proteases, which promote angiogenesis [45]. Neutrophils are attracted to the tumor site by the hypoxic condition [201] and active angiogenesis-inducing MMP-9 [202], which, not being complexed with its inhibitor TIMP-1, has a stronger activity [69]. M2-like TAMs act on tumorigenic ECM, upregulating MMPs’ expression and inducing the proteolytic demolition of interstitial collagen, along with the increasing endocytosis and lysosomal degradation of collagen [203]. Interestingly, as sustained by Afik R. et al. in a paper published in Journal of Experimental Medicine in 2016, in a colorectal cancer model, TAMs contribute not only to ECM degradation, but also to its deposition. Indeed, TAMs upregulate the synthesis and assembly of collagens and induce the deposition, crosslinking and linearization of collagen fibers in the proximity of invasive tumor cells [204].

## 4. Pre-Clinical Models’ Application in TME and IME

An appropriate system of microenvironment and tumor culture is the first step towards a better understanding of the complex interaction between cancer cells and their surroundings [10]. A possible approach is the deconstruction of the complex cellular microenvironment into a simpler and more predictable system. To date, the usual and common approach is the employment of standard animal models, such as murine, that present several advantages: short gestation times, small size, relatively inexpensive maintenance, and easy manipulation of gene expression [205]. However, the average rate of successful translation from animal models to clinical cancer trials is less than 8% [206]. The development of anticancer therapies has traditionally relied on 2D cultures [207], which has helped researchers to understand the biology of cancers cells, identify various carcinogens’ markers and discover new therapies. The 2D culture contribution to the advancement of knowledge in cancer research is related to its several advantages, such as the simple culture and manipulation, the methods for the cytotoxicity evaluation of a molecule are simple, highly standardized and repeatable [200]. Nevertheless, the 2D setting has been increasingly recognized as an over-simplified version of tumor conditions in vivo, often failing to address many of the more dominant pathological problems, such as the TME [208].

Monolayer cells do not grow in a physiological environment that allows them to assume different shapes and behaviors observed in vivo. In 2D cultures, cells are forced to polarize and their exchange area to culture media due to the attachment to rigid and flat substrates [207]. This leads to over-nutrition, over-oxygenation and non-reproducibility of the in vivo molecular gradients. In addition, in 2D settings, the composition, configuration and production of ECM are significantly altered [209]. It was estimated that only 5% of drugs in cellular models were found to be active in clinical trials [210]. This has encouraged the development of more realistic in vitro and reliable models, such as the 3D culture systems. In fact, researchers have found that cells in the 3D culture environment differ morphologically and physiologically from cells in the 2D culture environment. 3D cell culture provides a suitable micro-environment for optimal cell growth, differentiation and function, and the ability to create a tissue-like constructs in vitro [211]. 

### 4.1. Scaffold-Free and Scaffold-Based Systems

According to their morphology, 3D culture can be divided into two types of system: scaffold-free and scaffold-based (natural or synthetic origin). Nowadays, a 3D scaffold-free culture system can be identified as follows: spheroids, mosaic spheroids, spheres, organoids and colonspheres (specifically referred to colorectal 3D cells culture).

In particular, in the case of solid tumors, the 3D structure can mimic the inner avascular structure of neoplastic lesions in terms of growth kinetics, biochemical stimuli, tissue oxygenation and nutrient supply, the fundamental conditions that are completely absent in 2D culture systems [212]. Spheroids have gradients of oxygen, nutrition and metabolism; moreover, they closely resemble the ultrastructure and organization of the same cells when grown as a tumor in vivo. After drug treatment, spheroids exhibit resistance comparable to the in vivo situation. On the other hand, this system has some disadvantages, such as cell number limitation, low stability and the long-term culture complexity [213,214]. Lastly, concerning scaffold-based, there has been immense progress in the development of biomaterials to support scaffold formation, both for drug screening and regenerative medicine [215]. Biomaterials, which consist of ECM components, enhance cell activity and functions. The interactions with biomaterials enhance cell proliferation, differentiation, and biological functions, mimicking cancer cell-environment interaction [216]. An essential feature of biomaterials is biocompatibility, the ability of a material to act by determining an appropriate host response to a given application. 3D scaffolds must meet the following requirements: degree of porosity, enabling them to constitute a percolating pattern that favors cellular growth and disposition, nutrient intake, and disposal of metabolic products [217]; additionally, an appropriate surface characterized by adequate physical-chemical properties to promote adhesion, growth, proliferation, differentiation and migration of cells is required [218]. Natural scaffolds are constituted by polymers derived from animals or plants such as collagen, fibrin, glycosaminoglycan and HA [219]. Among them, gel systems, such as collagen-based ones, represent some of the earliest biomaterials and are still used today, as collagen is one of the most abundant components of the ECM. Magdeldin et al. (2014) developed a 3D tumor in vitro model based on the removal of interstitial fluid from collagen hydrogels with the aim of creating a multiwell drug testing platform [220]. Matrigel is another largely used hydrogel (Peptide Hydrogels—Versatile Matrices for 3D Cell Culture in Cancer Medicine); it consists of a basal membrane preparation in a solution extracted from Engelbreth-Holm-Swarm (EHS) mouse sarcoma, a tumor rich in ECM proteins. The main Matrigel components are laminin, collagen IV, entactin, and proteoglycan heparan-sulphate [221]. At room temperature, Matrigel is polymerized to produce a biologically active matrix that represents the basal membrane of mammalian cells. One limitation of Matrigel is that once polymerized, the matrix is not suitable for long-term storage [222]. In contrast, the mainly used plant-derived polymers are methyl-cellulose, agarose and alginate [223].

### 4.2. Patient-Derived Scaffold Obtained by Tissue Decellularization

Recently, tissue decellularization has emerged as a robust alternative technique in the field of tissue engineering and regenerative medicine [224]. The principal advantage of using biologically derived matrices instead of synthetic polymers for 3D tumor study in vitro, is that the main structural proteins and soluble factors are already present in decellularized scaffolds [225]. The term decellularization indicates the removal of the cellular component from a tissue, preserving its biochemical composition and biological and structural properties [226]. The decellularization of ECM can be achieved by three different approaches, including enzymatic, chemical and physical methods. Chemical methods include the perfusion, agitation or immersion with chemical solutions. These methods achieve the removal of most cellular materials but fail to preserve the ECM molecules. Physical methods include freezing, direct pressure, osmosis, sonication and agitation, which disrupt cellular membranes, provoking cell lysis. These methods prevent the disruption of ECM structures but, since these techniques only disintegrate cellular membranes, do not remove cellular debris, thus requiring the adding of enzymatic or chemical treatments to obtain acellular tissues [227]. Enzymatic decellularization methods require the employment of nucleases, proteases, and chelating agents, which are necessary for the removal or separation of cellular debris. Enzymatic approaches can effectively remove cellular materials preserving most of the collagen components. However, it is necessary to set a maximum treatment time to mitigate the disruptive effects on the architecture and composition of the ECM during the decellularization process [227].

### 4.3. 3D Patient-Derived dECM Models: Novel Strategies to Study IME to Prevent TME?

In Table 3, some of the studies in cancer research field obtained with the employment of patient derived dECM models are reported. These models provided new insights into the role of dECM in oncogenesis and cancer progression and represent a culture substrate for pharmacological and pharmacokinetic analyses to develop novel anticancer drugs [228]. Indeed, 3D patient derived-dECM models, allowing 3D culture of multiple cell populations, mimic cell-matrix interactions and predict treatment responses, as seen in patients diagnosed with PDAC [229,230] and CRC [13,231]. Furthermore, dECM provided new insights into the role of ECM in oncogenesis and cancer progression [232]. This encouraging evidence caused us to reflect on the employment of these models even more for chronic inflammation research, to recapitulate and investigate IME features with the aim to avoid progression toward TME and, thus, cancer initiation. 

These models could represent a potential tool to investigate inflammatory chronic diseases and could prove useful in exploring new patient-specific therapeutic approaches, reducing inflammation conditions and, potentially, also decreasing the risk of cancer initiation.

## 5. Conclusions and Future Perspectives

dECM models are a thriving tool in cancer research to study the complex mechanisms behind tumor invasion, progression, and response to treatment, since they recapitulate both the multicellular nature and 3D stromal environment present in an in vivo tumor. These models, taking advantages from the role of ECM in TME and its influence on cancer cells, can mimic the TME more closely than the conventional 2D system. Nevertheless, less attention has been given to the employment of these models for the study of conditions which could significantly increase the risk of cancer initiation, such as chronic inflammation disease. When the inflammatory responses became chronic, indeed, cell mutations and proliferation can provide a fertile ground conducive to cancer initiation and development. The correlation between the development of cancer from chronic condition is recognized, while the biological changes that underlie this progression are not clear. Therefore, a deeper understanding of the precise mechanisms by which chronic inflammation could trigger and predispose for cancer initiation could provide new insights for novel therapeutic strategies which could target this progressive passage to cancer.

A literature review provides information about the implication of ECM in IME and TME (Figure 1) and, furthermore, how its remodeling process may even trigger the switch between them. 3D dECMs are encouraging for their integration into a new field, such as chronic inflammation diseases, in which the ECM role is equally involved. The dECM models could provide new insight, which could be helpful in reducing the risks resulting from the chronicization of inflammation and thus, potentially, preventing triggers for cancer initiation.

## Figures and Tables

**Figure 1 cancers-14-05903-f001:**
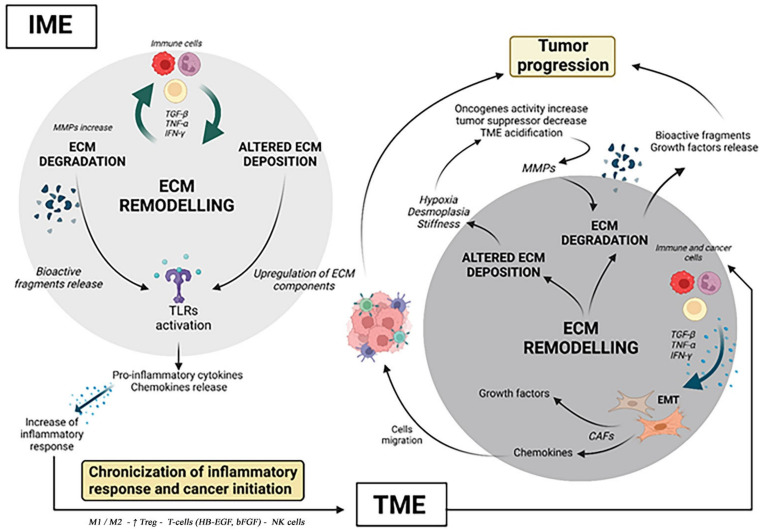
The ECM contribution in IME to TME progression. In the course of inflammation, in IME, immune cells induce the release of TNF-α, IFN-γ,TGF-β, influencing ECM turnover and protease activity. In turn, aberrant expression of ECM components and bioactive fragments resulting from ECM degradation activate TLRs on immune cells. The activation of TLRs induces pro-inflammatory cytokines’ and chemokines’ release, which fosters the inflammatory response. The perpetuation and the relapsing inflammatory response induce its chronicization and the passage from IME to TME. In TME, CAFs orchestrate ECM remodeling. CAFs originate from fibroblasts and mesenchymal cells or other cells in the EMT, induced by immune cells producing TNF-α, IFN-γ and TGF-β. CAFs exert several tumor-promoting functions, such as the release of growth factors and chemokines. In so doing, CAFs recruit endothelial and immune cells to join the TME, promoting tumor progression, while orchestrating the tumorigenic ECM deposition. Furthermore, altered ECM deposition provokes aberrant expression of some ECM components provoking hypoxia, desmoplasia and stiffness. Hypoxia induces the increase of oncogene activity, the inhibition of tumor suppressor genes and TME acidification. The acidification and metabolic changes in TME increase the MMPs’ activity and thus ECM degradation. The result of ECM degradation is the release of bioactive fragments and ECM-bound growth factors which induce the tumor progression. (Created with Biorender.com, accessed on 25 July 2022).

**Table 1 cancers-14-05903-t001:** The major ECM components and functions.

Components	Functions and Properties
Proteins and Structure	
Collagens Formed as fibrils within the ECM (Collagen I, II, III, V and XI)	Maintenance of ECM architecture, conferring of mechanical properties [20]. Influence on cell processes such as adhesion and migration. Binding of extracellular growth factors and cytokines [21].
Elastin Composed of single tropoelastin subunits cross-linked	Tissue elasticity, load bearing and storage of mechanical energy [22].
Glycoproteins	
Laminins Trimeric structure made of three different chains α, β, γ	Resident in the basement membrane. Cell adhesion, survival, migration, differentiation and migration via integrin [22].
Fibronectin (FN) Arranged into a mesh of fibrils and collagen and is linked to cell surface receptors (integrins)	Binds to collagen, fibrin and glycosaminoglycans, influencing cell adhesion, growth, migration, wound healing and differentiation [23].
Fibrillins	Scaffolds for elastin deposition [23].
GAGs	
Hyaluronic acid (HA)	Regulation of numerous cell functions and biological processes through the interaction with cell surface receptors. Lends tissue turgor and facilitates cell migration during tissue morphogenesis and repair [24].
PGs	
e.g., Heparan, chondroitin and keratin sulphates. Core of protein domain covalently linked to GAGs. Negatively charged, it can bind water.	Provide compressive resistance to tissue. Reservoir and sequester of growth factors, chemokines and cytokines at the ECM protecting them from degradation and creating effective gradients along ECM [25].
Matricellular proteins	
Thrombospondins (TSPs), tenascins (TNs), fibulins, osteopontin (OPN), cartilage oligomeric matrix protein (COMP), CNN family proteins, periostin and R-spondins	Cell behavior modulation, mediation of the interaction between cellular transmembrane receptors and fibrous ECM molecules and key functional regulator of cell-matrix communication [26].

**Table 2 cancers-14-05903-t002:** Summary of the main roles of stromal cells in TME.

Cancer-associated Endothelial cells	- Release of proangiogenic factors (PDGF-EC), EGF and VEGF under hypoxic conditions. - Promotion of cancer cell migration, invasion and metastasis. - Undergoing to EMT become CAF organized by TGF-β and bone morphogenetic protein (BMP), which leads to loss of cell-to-cell connections, detachment and elongation, enhanced migration and loss of endothelial properties [154].
CAFs	- Crosstalk between cancer cells and TME mediation. - Release of growth factors, cytokines to attract other cells (e.g., endothelial cells and immune cells) to take part in TME and ECM components’ deposition. - Release of MMP-3 for E-caderin degradation to facilitate cancer cell migration through the TME, induction of EMT and cell invasion promotion. - Induction of an immunosuppressive phenotype through the production of immune-modulatory chemokines and cytokines [155].
Cancer-associated adipocytes	- Secretion of metabolites, enzymes, hormones, growth factors and cytokines involved in matrix remodeling, invasion and cancer cell survival [156]. (e.g., tjroug production of insulin-like growth factor binding protein-2, MMP-11, and IL-6 and Il-1β increase migration and metastasis in human breast cancer cell in vitro and in vivo.) [157].
Tumor-associated immune stroma	- Tumor-associated macrophages (TAMs): promotion of tumor cell migration and invasion [158]. - Myeloid-derived suppressor cells differentiate into TAMs and dendritic cells, contribute to tumor progression, ECM remodeling and promoting EMT [159]. - Natural killer cells distinct in two subpopulation tumor-infiltrating natural killer cells (TINKs) and tumor-associated natural killer cells (TANKs) [160,161] and secrete altered levels of cytokines (e.g., pro-angiogenic factor such as VEGF and stromal derived factor-1), promoting angiogenesis and tumor progression [162]. - Regulatory T cells (Tregs): promotion of cell infiltration through the reduction of antitumor immune response [163]. - Mast cells with anti-tumorigenic activity: exert proinflammatory activity through degranulation process and recruiting innate immune system cells, orchestrate antitumor immune responses [164]. - Protumor activity: relasing of factors (e.g., VEGF) to support angiogenesis and MMP9 to degrade ECM and facilitate the metastasis [164].

**Table 3 cancers-14-05903-t003:** 3D repopulated dECM models.

dECM Origin	dECM Tissue	Cells Cultured on dECM	Results	Ref.
Human	CRC	Human colon adenocarcinoma cells (HT-29—HCT-116)	- 3D CRC model exhibited reduced sensitivity to 5-Fluoruracil and FOLFIRI treatments compared with 2D conventional cell cultures.	[13]
Human	HC—CRC—healthy liver (HL)—CRC and liver metastasis (CRLM)	Human colon adenocarcinoma cells (HT-29 ZsGreen/Luc+ and HCT-116 cells)	- Increased migration ability of CRC cells observed in CRC and CRLM scaffolds compared to HL scaffolds. - At an increasing concentration of drug, the response of the CRC cell line to chemotherapy was scaffold-dependent. HT-29 cells grown in HL scaffolds exhibited a significant reduction in cell proliferation compared to untreated scaffolds.	[233]
Human	CRC	Human colon adenocarcinoma cells (HCT-116 and HT-29)	- Tumor cells and the decellularized tumor matrix induced the differentiation of monocytes in macrophages, characterized by high surface expression of CD206 and reduced expression of MHC-II and CD86. - The exposition of macrophage to tumor cells and to tumor ECM induced a higher expression of IL-6, IL-10, TGF- CCL17, CCL18, and CCL22.	[234]
Human	Healthy colon and CRC	Human colon adenocarcinoma cells (HT-29 cells)	- Over-expression of IL-8 in 3D tumor matrices after 5 days of recellularization with HT-29 cells. - Secretome analysis showed a marked increase of DEFA3 in tumor stroma compared to healthy counterparts.	[235]
Human	PDAC	Human pancreatic adenocarcinoma cell lines (Panc-1 and AsPC-1)	- Immunofluorescence analyses after 7 days of scaffolds, recellularization with Panc-1 and AspC-1 confirmed the biocompatibility of 3D matrices to sustain engraftment, localization and infiltration. - Both PANC-1 and AsPC-1 cells cultured in 3D matrices showed a reduced response to treatment with FOLFIRINOX compared to conventional 2D culture.	[236]
Human	Esophagus from cadavers	Human bone marrow cell line (HM1-SV40)	- After the 72 h of exposure period to culture media conditioned with decellularized esophageal samples, HM1-SV40 cells appeared to be viable and proliferating, reaching about 90% confluence on the growth surface. The MTT assay revealed that, compared with untreated cultures, cells preserved 93.7%, 95.4% and 78.3% viability when treated with a medium conditioned with esophageal matrices (decellularized through Protocols No. 1, No. 2 and No. 3, respectively).	[237]
Rat	Esophageal adenocarcinoma (EAC)	Human mononuclear cells (THP-1) and esophageal epithelial cells (Het-1A)	- Metaplastic and neoplastic esophageal ECM induce distinctive effects upon THP-1 macrophage signalling compared to normal esophageal ECM. - The secretome of macrophages pre-treated with metaplastic and neoplastic ECM increases the migration of normal esophageal epithelial cells, similar behaviour observed in tumor cells.	[238]
Pig	Breast cancer	Human BC cell line (MDA-MB- 231) 3D tumor spheroids Human dermal fibroblasts	- Metastatic breast cancer model consisted of TME-mimetic platform comprising breast tissue-derived porcine d ECM; cancer cell assembled to form spheroids. In vitro generated monotypic and heterotypic dECM-Spheroids models exhibited the invasive profile of metastatic breast cancer cells and evidenced exometabolomic signatures similar to those observed in human tumor, translated in differences to cisplatin, gemcitabine and Palbociclib cancer-cell specific response.	[239]
Human	Breast cancer	Human breast cancer cells (MCF7, T-47D, and MDA-MB-231)	- Cells cultured in patient-derived scaffolds showed a higher drug resistance compared to the 2D cultures, demonstrating that the 3D environment improved cell chemoresistance.	[240]
Human and murine	Normal and tumor biopsies	Fibroblastic cell derived matrices (fCDM)	- This method offers the possibility of obtaining 3D matrices with relevant characteristics for mimicking particular in vivo microenvironments (e.g., normal vs. Desmoplastic or fibrous). - fCDMs can be used to scrutinize cocultures of different cell types (tumor, stroma, immune neural, etc.) that co-exist in the interstitial microenvironment of diverse tissues. - fCDM have been shown to serve as a useful platform to establish pre-clinical studies.	[241]

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
