# Peer review of "The Study of the Extracellular Matrix in Chronic Inflammation: A Way to Prevent Cancer Initiation?"

_cancers, 2022, doi:10.3390/cancers14235903_

Round 1
Reviewer 1 Report (Previous Reviewer 1)
The authors considered the previous comments, and they have made efforts to improve their manuscript. Instead of condensing the manuscript with tables, its length has increased significantly from 25 to 39 pages, but the comprehensibility has only been slightly improved. The phrases “as previously demonstrated (line 94), ~ cited (line 632), ~ described (782), ~ discussed (789), ~ pronounced (line 820), ~ discussed (842)” show that there are still repetitions. For example, lines 792-796 are identical to every word of lines 844-848. This could be streamlined for readability.
Major points:
Line 201: Does Src really stand for „steroid receptor coactivator” in this context, or rather short for sarcoma (SRC, the human orthologue of the Src gene)?
Line 285: As previously objected: filopodia lack MMP14. Contrary to what the authors state, reference 51 shows MMP14 incorporation only for invadopodia, which are a completely different type of cell protrusion compared to filopodia. Either a suitable work should be cited that supports the statement on filopodia, or the authors should refrain from making this statement.
Line 559 “…, in many cases, cancer cells are triggers for cancer initiation …” To my understanding, if cancer cells are present, they need not trigger cancer initiation because the person already has cancer.
Line 684: What is meant with “development of FAK”?
Line 705: What is meant with “solid stress”?
Line 737: What is meant with “although the ECM undergoes constant turnover, its perpetuation limit plasticity of the TME”?
All references should be checked thoroughly by the authors: I have randomly checked reference 189 (Ardi, V.C.; Kupriyanova, T.A.; Deryugina, E.I.; Quigley, J.P.; Chetram, M.A.; Bethea, D.A.; Odero-Marah, V.A.; Don-Salu-Hewage, A.S.; Jones, K.J.; Hinton, C. V; et al. The Role of Extracellular Matrix Components in Inflammatory Bowel Diseases. Biotechnol. Prog. 2014, 9, 63–71, doi:10.3390/jcm10051122.) and found that the doi points to a paper not from 2014 but from 2021 with this title, but by completely different authors (Derkacz A, Olczyk P, Olczyk K, Komosinska-Vassev K.; J. Clin. Med. 10(5):1122; PMID: 33800267). This is also cited as reference 67. There is a PNAS paper from 2007 by Ardi VC, Kupriyanova TA, Deryugina EI, Quigley JP (PMID: 18077379), which is cited as reference 71. And another Mol Cell Biochem-paper by Chetram MA, Bethea DA, Odero-Marah VA, Don-Salu-Hewage AS, Jones KJ, Hinton CV. (PMID: 23315288) which is cited as reference 182. These papers are apparently cobbled together in reference 189. But what is the “et al.” supposed to mean? Also, many references from #6 to #246 are incomplete as page numbers are missing.
Minor suggestions:
The review would benefit greatly if an English speaker revised the text. Understanding the manuscript is made difficult by many grammatical and spelling mistakes. For example:
Line 110: “cell proliferation … sustains cell proliferation”?
Line 152: “manly”
Lines 202, 205: “Akt” vs. “akt” - capitalization of genes and gene products is not arbitrarily and should not be mixed up.
225: “DDrs recptors” – the “r” already stands for “receptor”. Also check grammar of this and the following sentence.
Line 240: check grammar
Line 261: Is heparanase meant with “heparase”?
Line 261-262: check grammar
Line 265: grammar”…which it is secretes as zymogens…”
Line 267: “zinc” instead of “zin”
Line 290 ff.: check grammar; what is meant with “the highest number of substrate specificity”?
Line 294: Is “cleave” meant with “clean”?
Line 302: “on-matrix molecules” or “non-matrix molecules”
Line 411: “would healing”?
Line 427: The name of the author is not “Mertz” but “Metz”
Line 449: “biding”?
Line 493: It must be “biglycan” and not “glycan”.
Line 525: Delete “superfamily” when the protein is meant here.
Line 541: “genetic mutation” is redundant
Line 602: The common abbreviation is “MT1-MMP” instead of “MT-MMP-1”
Line 628: “contest” or “context”?
Table 2 is full of typographical errors: e.g., imme, suppresso, trhough, antitnumor, becam, hightly, imune, angiogenesi
Line 754: "denatured gelatin" is a misnomer. By definition, gelatin is denatured collagen.
Line 825: Is “CS8+” a typo that should read “CD8+”?
Line 892: The term “colonspheres” does not show up in reference 216.
Line 915: “contract limitation”?
The list of abbreviations is incomplete. At least COX-2, EP, RAS, ED-A, LTBP-LAP, MYC, RNS, JNK, PI3K, Akt, PTEN, PTP, MAPK, MHC, NK, MTT, DEFA3, and HC are missing.
Author Response
Response to Reviewer 1 Comments
Dear Editor,
thanks for reconsidering our manuscript for publication in your journal.
The reviewers’ comments have been addressed point by point by the authors. Furthermore, following the reviewer’s comments, we have checked and improved the English language. Any change in the text is traceable by “track changes” function in MS WORD.
The authors considered the previous comments, and they have made efforts to improve their manuscript. Instead of condensing the manuscript with tables, its length has increased significantly from 25 to 39 pages, but the comprehensibility has only been slightly improved. The phrases “as previously demonstrated (line 94), ~ cited (line 632), ~ described (782), ~ discussed (789), ~ pronounced (line 820), ~ discussed (842)” show that there are still repetitions. For example, lines 792-796 are identical to every word of lines 844-848. This could be streamlined for readability.
Thanks for the precise and insightful revision you performed and for your consideration. Multiple changes to the manuscript have been performed in accordance with your suggestions.
Major points:
Line 201: Does Src really stand for „steroid receptor coactivator” in this context, or rather short for sarcoma (SRC, the human orthologue of the Src gene)?
We apologize for the mistakes and we have corrected as follows:
Line 215: “SRC”.
Line 1195: “SRC (Proto-oncogene tyrosine-protein kinase)”.
Line 285: As previously objected: filopodia lack MMP14. Contrary to what the authors state, reference 51 shows MMP14 incorporation only for invadopodia, which are a completely different type of cell protrusion compared to filopodia. Either a suitable work should be cited that supports the statement on filopodia, or the authors should refrain from making this statement.
We kindly thank the reviewer to highlight our imprecision. In accordance with you and the following reference Hohensinner, P. J., Baumgartner, J., Kral-Pointner, J. B., Uhrin, P., Ebenbauer, B., Thaler, B., Doberer, K., Stojkovic, S., Demyanets, S., Fischer, M. B., Huber, K., Schabbauer, G., Speidl, W. S., and Wojta, J. (2017) PAI-1 (Plasminogen Activator Inhibitor-1) Expression Renders Alternatively Activated Human Macrophages Proteolytically Quiescent. Arterioscler., Thromb., Vasc. Biol. 37, 1913– 1922, DOI: 10.1161/ATVBAHA.117.309383, the statement on filopodia was deleted at line 285. The MMP14 activity was demonstrate in polarized macrophage filopodia but, on the contrary to what we stated, this activity was not related to the hypoxia condition.
Line 559 “…, in many cases, cancer cells are triggers for cancer initiation …” To my understanding, if cancer cells are present, they need not trigger cancer initiation because the person already has cancer.
We apologies with the reviewer. We have substituted cancer cell as follows:
Line 606: “stem cells”.
Line 684: What is meant with “development of FAK”?
Following the kindly notification, the text was modified as follows:
Line 730: “Notably, matrix stiffness provokes an increase in tension which induces integrin clustering, phosphorylation and activation of FAK. Once activated, FAK may promote RAS-mediated phosphorylation of ERK which can control migration, invasion, proliferation, and cell differentiation controlling myosin contraction, as well as the induction of transcription programs”
Line 705: What is meant with “solid stress”?
We thank the reviewer for raising the question, we have used this term superficially. Solid stress is defined as the mechanical stress that is contained and transmitted by the solid and elastic elements of the extracellular matrix and cells in TME and is a force distinc from ECM stiffness and fluid pressure. Since we tried to slim the manuscript, we have decided to delete this term and the relative reference to avoid the new information adding: [178] Nia, H.T.; Datta, M.; Seano, G.; Huang, P.; Munn, L.L.; Jain, R.K. Quantifying Solid Stress and Elastic Energy from Excised or in Situ Tumors. Nat. Protoc. 2018, 13, 1091–1105, doi:10.1038/nprot.2018.020
Line 737: What is meant with “although the ECM undergoes constant turnover, its perpetuation limit plasticity of the TME”?
We kindly thank the reviewer to underlie us the confusing sentence. We have decided to delete it from the text.
All references should be checked thoroughly by the authors: I have randomly checked reference 189 (Ardi, V.C.; Kupriyanova, T.A.; Deryugina, E.I.; Quigley, J.P.; Chetram, M.A.; Bethea, D.A.; Odero-Marah, V.A.; Don-Salu-Hewage, A.S.; Jones, K.J.; Hinton, C. V; et al. The Role of Extracellular Matrix Components in Inflammatory Bowel Diseases. Biotechnol. Prog. 2014, 9, 63–71, doi:10.3390/jcm10051122.) and found that the doi points to a paper not from 2014 but from 2021 with this title, but by completely different authors (Derkacz A, Olczyk P, Olczyk K, Komosinska-Vassev K.; J. Clin. Med. 10(5):1122; PMID: 33800267). This is also cited as reference 67. There is a PNAS paper from 2007 by Ardi VC, Kupriyanova TA, Deryugina EI, Quigley JP (PMID: 18077379), which is cited as reference 71. And another Mol Cell Biochem-paper by Chetram MA, Bethea DA, Odero-Marah VA, Don-Salu-Hewage AS, Jones KJ, Hinton CV. (PMID: 23315288) which is cited as reference 182. These papers are apparently cobbled together in reference 189. But what is the “et al.” supposed to mean? Also, many references from #6 to #246 are incomplete as page numbers are missing.
Following the reviewer’s comments, we decided to check and reorganize all references in the text and in the bibliography.
Minor suggestions:
The review would benefit greatly if an English speaker revised the text. Understanding the manuscript is made difficult by many grammatical and spelling mistakes. For example:
Line 110: “cell proliferation … sustains cell proliferation”?
We apologize with the reviewer for the repetition. We have modified the text as follows
Line 118: “The proliferation of cells alone does not cause cancer but, the presence of inflammatory cells, growth factors and DNA-damage-promoting agents contribute to stroma activation, promoting neoplastic risk.”
Line 152: “manly”
We apologize with the reviewer we have modified the text as follows:
Line 164: “mainly”.
Lines 202, 205: “Akt” vs. “akt” - capitalization of genes and gene products is not arbitrarily and should not be mixed up.
We apologize for the typo. We have corrected the text as follows:
Line 220: “Akt”.
225: “DDrs recptors” – the “r” already stands for “receptor”. Also check grammar of this and the following sentence.
We apologize with reviewer for the mistake. We delete “receptors” and further we modify the text as follows:
Line 239: “Cells use actomyosin contractibility to remodel the ECM and to sense its material properties or stiffness while integrins, DDrs and FAKs mediate the cell response”.
Line 240: check grammar
We modify the text at line as follows:
Line 255: “Fibroblasts are the mainly producer of ECM components whether in physiological or pathological conditions”.
Line 261: Is heparanase meant with “heparase”?
The reviewer intuition was correct. We apologize for the mistake, we have substituted with correct word as follows:
Line 278: “heparanase”.
Line 261-262: check grammar.
The text was modified as follows:
Line 279: “MMPs are of outstanding importance in ECM remodeling and are crucially involved in cancer progression more than the other ECM-degrading enzymes”.
Line 265: grammar”…which it is secretes as zymogens…”
We apologize for the grammar mistake we correct the text as follows:
Line 283 “ which are secreted as zymogens”.
Line 267: “zinc” instead of “zin”
Thank to the reviewer we have corrected as follows:
Line 285: “zinc”.
Line 290 ff.: check grammar; what is meant with “the highest number of substrate specificity”?
We thank the reviewer for the suggestion, the text was modified at line as follows
Line 309: “Among MMPs, MT1-MMP also known as MMP-14, exhibits the highest value of substrate specificity cleaving the interstitial collagen I, II, while, activating MMP-2, can also indirectly promote the cleavage of type IV”.
Line 294: Is “cleave” meant with “clean”?
We correct the word as follows:
Line 314: “cleave”
Line 302: “on-matrix molecules” or “non-matrix molecules”
We have decided to delete the sentence to slim the text.
Line 411: “would healing”?
We apologize with the reviewer we have modified the text as follows:
Line 447: “wound healing”.
Line 427: The name of the author is not “Mertz” but “Metz”
We apologize for the typo. We have decided to delete the sentence to slim the text.
Line 449: “biding”?
we apologize for the typo. We have corrected the text as follows:
Line 488: “binding”.
Line 493: It must be “biglycan” and not “glycan”.
We thank the reviewer for the notification. We have corrected the text as follows:
Line 531: “biglycan”.
Line 525: Delete “superfamily” when the protein is meant here.
We thank the reviewer for the kindly suggestion. We have deleted the word “superfamily” at line 567
Line 541: “genetic mutation” is redundant
In agreement with the reviewer the word “genetic” was deleted in the text at line 584.
Line 602: The common abbreviation is “MT1-MMP” instead of “MT-MMP-1”
We apologize with the reviewer we have corrected the abbreviation as follows:
Line 648: “MT1-MMP”.
Line 628: “contest” or “context”?
Thank to the reviewer we have corrected the typo as follows:
Line 675: “context”.
Table 2 is full of typographical errors: e.g., imme, suppresso, trhough, antitnumor, becam, hightly, imune, angiogenesis.
We are desolated we have corrected all the highlighted words in table 2.
Line 754: "denatured gelatin" is a misnomer. By definition, gelatin is denatured collagen.
We thank the reviewer for the suggestion. We have deleted “denaturated” at line 807.
Line 825: Is “CS8+” a typo that should read “CD8+”?
The reviewer intuition was right. We have corrected as follows:
Line 881: “CD8+”.
Line 892: The term “colonspheres” does not show up in reference 216.
Apologizing with the reviewer with for the lack, we have provided the reference 214 in the text.
Line 915: “contract limitation”?
Apologizing with the reviewer we have deleted the wrong sentence in the text.
The list of abbreviations is incomplete. At least COX-2, EP, RAS, ED-A, LTBP-LAP, MYC, RNS, JNK, PI3K, Akt, PTEN, PTP, MAPK, MHC, NK, MTT, DEFA3, and HC are missing.
Following the reviewer’s comment, we decided to update the abbreviations list in the text.

Reviewer 2 Report (Previous Reviewer 3)
No further comments
Author Response
We sincerely thank the reviewer for the previous suggestions which have improved the manuscript.
Reviewer 3 Report (Previous Reviewer 4)
I would like to thank the authors for their response. The cover letter indicates extensive changes have been made that should address most of my original comments. However, I have identified some issues that still need to be resolved:
1. The main issue is readability. I’m afraid the writing significantly impacts on readability. The whole manuscript requires an edit for the appropriate use of the English language.
2. The name listed for Src abbreviation (steroid receptor coactivator) is not correct. The correct name is SRC Proto-Oncogene, Non-Receptor Tyrosine Kinase https://www.genecards.org/cgi-bin/carddisp.pl?gene=SRC
3. The abbreviation is DDR for discoidin domain receptor. Originally this had been inconsistently applied. There is now a conflict with DDR used for dna damage response. However, terms that are used infrequently do not need abbreviations.
4. Line 240 does not read “These cells, once activated become myofibroblasts” as stated in the cover letter
5. Lines 335-336 “Since we proceed to describe the role of ECM in IME and then in TME, a brief overview of the immune microenvironment and its interaction with the ECM is need” are not required and can be removed. The same is true for a similar introductory sentence in section 3 at lines 529-531.
6. New text at line 386 (beginning with the word neutrophils) should be a new paragraph
7. Line 425 formatting indent error
8. Please rewrite line 460: “Proceeding through the description, entactin/nidogen fragments from basement membrane components also exert neutrophil chemotaxis which is, in this case, is integrin mediated [86].” There is a double ‘is’ and starting the sentence with ‘proceeding through the description’ does not seem appropriate.
9. Line 630 remove ‘recapitulated in’
Author Response
Response to Reviewer 3 Comments
Dear Editor,
thanks for reconsidering our manuscript for publication in your journal.
The reviewers’ comments have been addressed point by point by the authors. Furthermore, following the reviewer’s comments, we have checked and improved the English language. Any change in the text is traceable by “track changes” function in MS WORD.
I would like to thank the authors for their response. The cover letter indicates extensive changes have been made that should address most of my original comments. However, I have identified some issues that still need to be resolved:
- The main issue is readability. I’m afraid the writing significantly impacts on readability. The whole manuscript requires an edit for the appropriate use of the English language.
Thanks for the precise and insightful revision you performed and for your consideration. Multiple changes to the manuscript have been performed in accordance with your suggestions.
- The name listed for Src abbreviation (steroid receptor coactivator) is not correct. The correct name is SRC Proto-Oncogene, Non-Receptor Tyrosine Kinase https://www.genecards.org/cgi-bin/carddisp.pl?gene=SRC
We apologize for the mistake, we have removed the wrong abbreviation and corrected as follows:
Line 215: “SRC”.
Line 1195: “SRC (Proto-oncogene tyrosine-protein kinase)”.
- The abbreviation is DDR for discoidin domain receptor. Originally this had been inconsistently applied. There is now a conflict with DDR used for dna damage response. However, terms that are used infrequently do not need abbreviations.
We thank the reviewer for the suggestion and we have deleted the abbreviation inappropriately used “DDR” for dna damage response at line 591.
- Line 240 does not read “These cells, once activated become myofibroblasts” as stated in the cover letter
We apologize for the mistake and we have corrected in the manuscript as follows:
Line 257: “These cells, once activated become myofibroblasts”.
- Lines 335-336 “Since we proceed to describe the role of ECM in IME and then in TME, a brief overview of the immune microenvironment and its interaction with the ECM is need” are not required and can be removed. The same is true for a similar introductory sentence in section 3 at lines 529-531.
Following the reviewer’s comment, we decided to delete the introductory sentences in section 2 and 3.
- New text at line 386 (beginning with the word neutrophils) should be a new paragraph
In agreement with the reviewer, we have modified the title of the paragraph 2.2 as follows:
Line 389: “ECM damaging and remodeling during chronic inflammation”.
Further, we have provided a new paragraph 2.3 as follows:
Line 418: “Key immune cells driving ECM remodelling during chronic inflammation”.
- Line 425 formatting indent error
We thank the reviewer and we have corrected the wrong word as follows:
Line 384: “remodeled”
- Please rewrite line 460: “Proceeding through the description, entactin/nidogen fragments from basement membrane components also exert neutrophil chemotaxis which is, in this case, is integrin mediated [86].” There is a double ‘is’ and starting the sentence with ‘proceeding through the description’ does not seem appropriate.
Following the reviewer’s comment, we have corrected the text as follows:
Line 498: “Entactin/nidogen fragments derived from basement membrane components cleavege, also exhert neutrophil chemotaxis which is integrin-mediated
- Line 630 remove ‘recapitulated in’
We thank the reviewer for the suggestion and we have deleted “recapitulated in” at line 676.

Reviewer 4 Report (Previous Reviewer 5)
The manuscript has been improved well. I recommend the publication.
Author Response
We sincerely thank the reviewer for the previous suggestions that have improved the manuscript.
Round 2
Reviewer 1 Report (Previous Reviewer 1)
Marangio and co-authors have made many efforts to take all comments into account.
This manuscript is a resubmission of an earlier submission. The following is a list of the peer review reports and author responses from that submission.
Round 1
Reviewer 1 Report
In their review article "The study of the Extracellular Matrix in chronic inflammation: a way to prevent cancer initiation?" the authors summarize the major components of the ECM with their main properties and functions. They also describe the interaction of cells with their surrounding ECM. They turn their attention on pathophysiological conditions, particularly chronic inflammation and cancer, where they focus on the tumor microenvironment. They continue with reporting on studies that use three-dimensional decellularized matrices of tumor tissues, and they recommend these as tools to study chronic inflammatory diseases and cancer initiation.
With regard to matrix molecules, matrix receptors and matrix-modifying enzymes, this review does not stand out from other reviews on these topics. The way in which the interrelation of ECM components and their receptors is presented is difficult to understand. For example, line 241-243: "in alternative, ECM ligands can be responsible of the recruitment of talin and kindlin to integrin cytoplasmic tail, binding integrin extracellular domain." This could be misconstrued to mean that talin and kindlin bind to the extracellular domain of integrins.
The authors repeatedly emphasize the utility of patient-derived decellularized three-dimensional extracellular matrices. Although decellularization, which is explicitly mentioned as a keyword, is of great importance in the context of this review, the method(s) for obtaining patient-derived decellularized ECM scaffolds are neither described nor discussed. For example, treatment with detergents and nucleases is usually required. It would be interesting for the reader to know which components of the extracellular space are preserved and which are lost. Methods for repopulating a decellularized matrix with different cell types could also be discussed.
Major points:
A quick Pubmed search using the search terms listed in this review "Chronic Inflammation Cancer 3D culture model Extracellular Matrix Decellularization” yielded a 2020 paper titled “Engineering clinically-relevant human fibroblastic cell-derived extracellular matrices“ by Franco-Barraza et al. in Methods Cell Biol. (PMID 32222216). I wonder why it is not cited by the authors in the context of patient-derived decellularized ECM. Using the search terms “Chronic Inflammation Cancer Decellularization”, Pubmed points to a 2021 paper by Saldin et al. in J Immunol Regen Med with the title “The effect of normal, metaplastic, and neoplastic esophageal extracellular matrix upon macrophage activation” (PMID 34027260) that is also not cited by the authors. Another Pubmed search using the search terms "decellularized matrix cancer" yielded 23 review articles in the last five years alone. I wonder why none of these are cited by the authors. The review by Hoshiba (2019) Decellularized Extracellular Matrix for Cancer Research. Materials (Basel) 12(8): 1311 (PMID 31013621) would be definitely interesting to the readers of this review. In my opinion, the paper by Jin et al. (2019) Decellularization of tumours: A new frontier in tissue engineering. J Cell Physiol 2019 Vol. 234(4), 3425-3435 (PMID 35495097) is also worth citing.
The text is cumbersome and sometimes uses words imprecisely. For example, line 79: "procollagens fibers form triple helices" There are no procollagen fibers: peptide chains self-assemble to procollagen that is processed to tropocollagen, which self-assembles to fibrils that ultimately form fibers.
Line 370: "In physiologic condition, the collagen fibers which surrounding the normal epithelial structures are curly and anisotropic" - collagen does not show a preferred orientation and is thus isotropically arranged in this setting. Anisotropy means that collagen fibers are preferentially aligned in the same orientation.
Line 445: Instead of emphasizing the somewhat exotic and still poorly understood subcellular localization of some MMPs in mitochondrial and nuclear compartments, more information on the various MMPs, their different substrates and their regulation would be desirable.
Line 451: Do filopodia have MMP activity?
Lines 551-552, 578: MMP-2 and -9 are gelatinases and cannot degrade collagens.
Line 585-586: Neither elastase (isolated from sputum in ref. #124) nor MMP12 (the macrophage elastase) are chemotactic signals in contrast to elastokines that can be generated by these proteases. If the authors mean with “MMP12 digests” elastin degradation products, they should make that clear.
Line 598: The reference "Midwood K. et al." is not given in this somewhat garbled sentence.
Line 638: Neither MMP12 nor MT1-MMP are mentioned in reference #147.
Line 676-677:"...in many cases, cancer cells are considered as "seeds" for cancer initiation [148]." In reference #148, seeding is discussed in the context of metastasis and not cancer initiation.
Lines 688, 711, and others: It is important to discriminate between different MMPs, because they exert very different functions. For example, I suppose that in line 689, MMP9 is meant, whereas in line 711 MMP2 and MT1-MMP are meant.
Line 737: Patient-derived decellularized matrices are indeed very interesting. However, their production is anything but easy. At least a brief description of the protocol would be desirable. It would also be helpful to estimate which matrix components are retained during production or are inevitably lost.
Line 749 ff: Some information on the different cell lines (e.g., the type of the cancer from which they are derived) would be helpful in a (missing) legend of the table. MCF-7 definitely differs for MDA-MB-231, yet both are indiscriminately described here as breast cancer cells.
Line 764: I understand that three-dimensional dECMs are suitable to investigate processes of chronic inflammation in cancer progression. However, I do not understand how they might be used as a "possible prevention tool" as stated by the authors.
Minor points:
Line 125: The name is “Dvorak” and not “Dvorakin”
Line 126: “would healing” – wound healing?
Line 143: What is meant by “cancer risk initiation”?
Line 161: “manly”?
Line 188: what is meant by “ECM stroma”?
Often more words than necessary are used. For example, line 193: “intracellular cytoskeleton”. The cytoskeleton is of course within a cell. Or in line 464-466: “production of bioactive shorter fragments”. Fragments are always shorter than the original molecule.
Line 197: “cell surface receptors such as integrin, collagen and fibrin” sounds misleading.
Line 223: What does the term “(glycol) proteins” mean?
Line 240: "ligand biding"?
Line 290: "linearized collagen" and line 372: “linearized fibers”? I assume that parallel alignment is meant.
Line 387: “integrin grouping” – clustering?
Line 402, 404: “LTBP-LAP”, “TGB-beta1” -A list of abbreviations would be helpful to the reader. Line 421: ROS = reactive oxygen species; line 731: ROS=retinoic orphan receptor: Different abbreviations would be useful here. Does the abbreviation in line 731 make sense, since this expression is only used once? Table 1: what does “FOLFIRI” stand for?
Line 440: “cleaved degraded”?
Lines 511, 680: “addiction”?
Line 514: “side” or site?
Line 583-585: what “list“ is meant here? Besides, the grammar is wrong.
Lines 607, 608: “bind”?
Line 735: "... and its related manly related the promotion angiogenesis ..." - What statement is intended here? In addition, many other sentences are very long and yet incomplete: For example, lines 72-75, 184-186, 610-611, 621-623
758: "From our review resulted a laking data ..."?
774: In the figure legend "TGF-beta, FGF2, PDGF" are mentioned, whereas in the figure "TGF-beta, TNF-alpha, IFN-gamma" are shown.
Author Response
Response to Reviewer 1 Comments
Oct 28th, 2022
Dear Editor,
we kindly thank to all the reviewers for the brilliant suggestions.
We carefully considered all the comments by the Reviewers and we revised the manuscript accordingly. We feel that the manuscript has been significantly improved thanks to the Reviewers’ comments and we hope the paper will now be considered acceptable for publication. In accordance with Reviewer’s requests, we decided to make extensive changes in manuscript’s schedule. To make more readable and to streamline the text, we decided to modify the structure of the review as follows:
1.Introduction
1.1. ECM More Than A Physical Support: An Overview
1.2. ECM Major Components, Properties and Functions
1.3. ECM-Cells Interplay
1.4 ECM deposition
1.5 Degradation of ECM and MMPs activity
1.6 ECM dynamics under pathological conditions
- Role of ECM in IME
2.2 Immune cells components drive ECM remodeling during inflammation2.3 ECM-derived fragments as modulators of chemotactic activity for immune cells
2.4 Alteration in ECM components affect immune cells activity.
2.5 ECM-fragments influence gene expression of inflammatory cells.
- From chronic inflammation to cancer: the main correlations
3.1. Chronic inflammation and tumor initiation
3.2. Main inflammatory pathways correlated to tumorigenesis and colitis associated cancer
3.3 ECM deposition in TME
3.4 Tumorigenic Alterations in ECM composition
3.4.1 MMPs role in shaping TME
3.5 Tumorigenic ECM and its remodeling influence immune cells within tumor mass.
- Pre-clinical models application in TME and IME
4.1 Scaffold-free and Scaffold-based systems
4.2 Patient-derived scaffold obtained by tissue decellularization
4.1 3D patient-derived dECM models: novel strategies to study IME to prevent TME?
- Conclusions and Future Perspectives
Further, following the Reviewer’s suggestions we provided table 1 for ECM components properties and functions, table 2 for stromal cells description. In conclusion, as cleverly suggested, we updated the table 3 for 3D repopulated decellularized ECM models and we added an abbreviations list.
In their review article "The study of the Extracellular Matrix in chronic inflammation: a way to prevent cancer initiation?" The authors summarize the major components of the ECM with their main properties and functions. They also describe the interaction of cells with their surrounding ECM. They turn their attention on pathophysiological conditions, particularly chronic inflammation and cancer, where they focus on the tumor microenvironment. They continue with reporting on studies that use three-dimensional decellularized matrices of tumor tissues, and they recommend these as tools to study chronic inflammatory diseases and cancer initiation.
With regard to matrix molecules, matrix receptors and matrix-modifying enzymes, this review does not stand out from other reviews on these topics. The way in which the interrelation of ECM components and their receptors is presented is difficult to understand. For example, line 241-243: "in alternative, ECM ligands can be responsible for the recruitment of talin and kindlin to integrin cytoplasmic tail, binding integrin extracellular domain." This could be misconstrued to mean that talin and kindlin bind to the extracellular domain of integrins.
We thank the reviewer for notifying us of the confusing sentence.
We modify the text as follows:
Line 192: “Indeed, biochemical signals coming from intracellular space can induce conformational change in integrin extracellular domain, facilitating ECM ligand binding through promotion of talin and kindlin recruitment to the cytoplasmatic tails to facilitate ECM ligand binding. Regarding the outside-in signaling in ECM ligands bind to the extracellular domain of integrin and contribute to the recruitment of talin and kindlin to their cytoplasmatic tail.
The authors repeatedly emphasize the utility of patient-derived decellularized three-dimensional extracellular matrices. Although decellularization, which is explicitly mentioned as a keyword, is of great importance in the context of this review, the method(s) for obtaining patient-derived decellularized ECM scaffolds are neither described nor discussed. For example, treatment with detergents and nucleases is usually required. It would be interesting for the reader to know which components of the extracellular space are preserved and which are lost. Methods for repopulating a decellularized matrix with different cell types could also be discussed.
We thank the reviewer for the interesting suggestions. On this purpose we enrich Section 4. “Pre-clinical models application in TME and IME” as follows:
Line 853: “4. Pre-clinical models application in TME and IME”
“The best experimental way to develop an effective anticancer therapy is to comprehensively understand the mechanisms responsible for the onset, the development and the tumor diffusion [208]. Using an appropriate system of microenvironment and tumor culture is the first step towards a better understanding of the complex interaction between cancer cells and their surroundings [10]. In fact, a possible approach is the deconstruction of the complex cellular microenvironment into a simpler and more predictable system. To date, the usual and common approach is to use standard animal models such as mouse model that present several advantages: short time gestation times, small size, relatively inexpensive maintenance, and easy manipulation of gene expression [209]. However, the average rate of successful translation from animal models to clinical cancer trials is less than 8% [210]. The development of anticancer therapies has traditionally relied on 2D cultures [211] which helped researchers to understand the biology of cancers cells, identify various carcinogens markers, and discover new therapies.. The 2D culture contribution to the advancement of knowledge in cancer research is related to its several advantages such as the easily culture and manipulation, the methods for the cytotoxicity evaluation of a molecule are simple, highly standardized and repeatable.[200] Nevertheless, the 2D setting limitation have been increasingly recognized, as a fully reliable pre-clinical tumor model, mainly as an over-simplified version of tumor conditions in vivo, often failing to address many of the more dominant pathological problems, such as the TME. [212].
Monolayer cells do not grow in a physiological environment that allows them to assume the different shapes and behaviours observed in vivo. In 2D cultures, cells are forced to polarize and increase their exchange area to culture media due to the attachment to rigid and flat substrates [211]. This leads to an over-nutrition, over-oxygenation and non-reproducibility of the in vivo molecular gradients. In addition, in 2D settings, the composition, configuration and production of ECM are significantly altered [213]. It was estimated that only 5% of drugs in cellular models were found to be active in clinical trials [214]. This has encouraged the development of more realistic in vitro and reliable models such as the 3D culture systems. In fact, researchers have found that cells in the 3D culture environment differ morphologically and physiologically from cells in the 2D culture environment. 3D cell culture provides a suitable micro-environment for optimal cell growth, differentiation and function, and the ability to create a tissue-like constructs in vitro [215]”.
4.1 Scaffold-free and Scaffold-based systems
According to their morphology, 3D culture can be divided in two types of system: scaffold-free and scaffold-based (natural or synthetic origin). Nowadays, a 3D scaffold-free culture system can be defined by the following terms: spheroids, mosaic spheroids spheres, organoids and colonspheres (specifically referred to colorectal 3D cells culture).
In particular, in the case of solid tumors, the 3D structure can mimic the inner avascular structure of neoplastic lesions in terms of growth kinetics, biochemical stimuli, tissue oxygenation, and nutrients supply, the fundamental conditions that are completely absent in 2D culture systems [216]. Spheroids have gradients of oxygen, nutrition and metabolism; moreover, they closely resembled the ultrastructure and organization of the same cells when grown as a tumor in vivo. After drug treatment, spheroids exhibit resistance comparable to the in vivo situation. On the other hand, this system has some disadvantages such as cell number limitation, low stability and the long-term culture complexity [217]. Lastly, concerning scaffold-based, there has been immense progress in the development of biomaterials to support scaffold formation both for drug screening and regenerative medicine [218]. Biomaterials, which consist of ECM components, enhance the cell activity or functions. The interaction with biomaterials enables cells to enhance their proliferation, differentiation, and biological functions, leading to the realization of cancer cell–environment interaction [219]. An essential feature of biomaterials is the biocompatibility, the ability of a material to act by determining an appropriate host response to a given application. 3D scaffold must meet the following requirements: degree of porosity, enabling to constitute a percolating pattern that favours cellular growth and disposition, nutrient intake, and disposal of metabolic products [220]; also an appropriate surface characterized by adequate physical-chemical properties to promote adhesion, growth, proliferation, differentiation and migration of cells [221]. The 3D synthetic model showed a significantly higher drug resistance [222]. Despite their biocompatibility, the presence of polyvinylidene fluoride (PVDF) fibers generates an artificial barrier between the pathological tissue and the surrounding environment. This contract limitation ends up excluding the flow of large molecules such as antibodies [223]. Dainiak et al. (2008) have developed microporous hydrogels (MHs), it consists of the gelation of the polyacrylamide at subzero temperatures, followed by its functionalization by covalent immobilization of type collagen I and by copolymerization of cell-adhesive peptide RGD mimetics during its production [224].
Furthermore, natural scaffolds are constituted by polymers derived from animals or plants [225]. The mainly used animal-derived polymers are collagen, fibrin, glycosaminoglycan and HA. Among them, gel systems, such as collagen-based ones, represent one of the earliest biomaterials and are still used nowadays since collagen is one of the most abundant components of the ECM. Magdeldin et al. (2014) developed a 3D tumor in vitro model based on the removal of interstitial fluid from collagen hydrogels with the aim to create a multiwell drug testing platform [226]. Matrigel is another largely used hydrogel (Peptide Hydrogels - Versatile Matrices for 3D Cell Culture in Cancer Medicine), it consists of a basal membrane preparation in a solution extracted from Engelbreth-Holm-Swarm (EHS) mouse sarcoma, a tumor rich in ECM proteins. The main components are laminin, collagen IV, entactin, and proteoglycan heparan-sulfate [227]. At room temperature, Matrigel polymerized to produce a biologically active matrix that represents the basal membrane of mammalian cells. One limitation of Matrigel is that once polymerized, the matrix is not suitable for long-term storage [228]. In contrast, the mainly used plant-derived polymers are methyl-cellulose, agarose and alginate [229].
4.2 Patient-derived scaffold obtained by tissue decellularization
Recently, tissue decellularization has emerged as a robust alternative technique in the field of tissue engineering and regenerative medicine [230]. The principal advantage of using biological-derived matrices, instead of synthetic polymers for 3D tumor study in vitro, is that the main structural proteins and soluble factors are already present in decellularized scaffolds [231]. The term decellularization indicate the removal of the cellular component from a tissue by minimally altering its biochemical composition and biological and structural properties [232]. The decellularization of ECM can be achieved by three different approaches: enzymatic, chemical and physical methods. Chemical methods include the perfusion, agitation or immersion with chemical solutions. These methods achieve removal of the majority of cellular materials but fail to preserve the ECM molecules. That is why, the decellularization process with combined chemical treatments should be optimized for the least damage to ECMs [230]. Physical methods include freezing, direct pressure, osmosis, sonication and agitation. They can be used to decellularized tissues by disrupting cellular membranes, with corresponding cell lysis. Physical prevent the disruption of ECM structures, but since these techniques only disintegrate cellular membranes, but do not remove cellular debris, requiring the adding of enzymatic or chemical treatments to obtain acellular tissues [233]. Enzymatic decellularization uses nucleases, proteases, and chelating agents (necessary for the removal or separation of cellular debris). Enzymatic approaches can effectively remove cellular materials preserving most of the collagen components. However, it is necessary to set a maximum treatment time to mitigate the disruptive effects on the architecture and composition of the ECM during the decellularization process [233]”.
Major points:
A quick Pubmed search using the search terms listed in this review "Chronic Inflammation Cancer 3D culture model Extracellular Matrix Decellularization” yielded a 2020 paper titled “Engineering clinically-relevant human fibroblastic cell-derived extracellular matrices“ by Franco-Barraza et al. in Methods Cell Biol. (PMID 32222216). I wonder why it is not cited by the authors in the context of patient-derived decellularized ECM. Using the search terms “Chronic Inflammation Cancer Decellularization”, Pubmed points to a 2021 paper by Saldin et al. in J Immunol Regen Med with the title “The effect of normal, metaplastic, and neoplastic esophageal extracellular matrix upon macrophage activation” (PMID 34027260) that is also not cited by the authors. Another Pubmed search using the search terms "decellularized matrix cancer" yielded 23 review articles in the last five years alone. I wonder why none of these are cited by the authors. The review by Hoshiba (2019) Decellularized Extracellular Matrix for Cancer Research. Materials (Basel) 12(8): 1311 (PMID 31013621) would be definitely interesting to the readers of this review. In my opinion, the paper by Jin et al. (2019) Decellularization of tumours: A new frontier in tissue engineering. J Cell Physiol 2019 Vol. 234(4), 3425-3435 (PMID 35495097) is also worth citing.
We thank the reviewer for the relevant suggestion. We have decided to reorganize the table 3 adding the following studies
- Sensi, F.; D’Angelo, E.; Biccari, A.; Marangio, A.; Battisti, G.; Crotti, S.; Fassan, M.; Laterza, C.; Giomo, M.; Elvassore, N.; et al. Establishment of a Human 3D Pancreatic Adenocarcinoma Model Based on a Patient-Derived Extracellular Matrix Scaffold. Transl. Res. 2022, doi:10.1016/j.trsl.2022.08.015.
- Barbon, S.; Biccari, A.; Stocco, E.; Capovilla, G.; D’Angelo, E.; Todesco, M.; Sandrin, D.; Bagno, A.; Romanato, F.; Macchi, V.; et al. Bio-Engineered Scaffolds Derived from Decellularized Human Esophagus for Functional Organ Reconstruction. Cells 2022, 11, doi:10.3390/cells11192945.
- Saldin, L.T.; Klimak, M.; Hill, R.C.; Cramer, M.C.; Huleihel, L.; Li, X.; Quidgley-Martin, M.; Cardenas, D.; Keane, T.J.; Londono, R.; et al. The Effect of Normal, Metaplastic, and Neoplastic Esophageal Extracellular Matrix upon Macrophage Activation. J. Immunol. Regen. Med. 2021, 13, doi:10.1016/j.regen.2020.100037.
- Hoshiba, T. Decellularized Extracellular Matrix for Cancer Research. Mater. (Basel, Switzerland) 2019, 12, doi:10.3390/ma12081311.
- Franco-Barraza, J.; Raghavan, K.S.; Luong, T.; Cukierman, E. Engineering Clinically-Relevant Human Fibroblastic Cell-Derived Extracellular Matrices. Methods Cell Biol. 2020, 156, 109–160, doi:10.1016/bs.mcb.2019.11.014.
The text is cumbersome and sometimes uses words imprecisely. For example, line 379: "procollagens fibers form triple helices "There are no procollagen fibers: peptide chains self-assemble to procollagen that is processed to tropocollagen, which self-assembles to fibrils that ultimately form fibers.
We kindly thank the reviewer for notifying us this inaccurate description regarding the process which lead to collagen assembly. We have modified the text as follows
Line 675: “Modified procollagens form triple helices and are further processed extracellularly by proteases to create collagen fibrils".
Line 370: "In physiologic condition, the collagen fibers which surround the normal epithelial structures are curly and anisotropic" - collagen does not show a preferred orientation and is thus isotropically arranged in this setting. Anisotropy means that collagen fibers are preferentially aligned in the same orientation.
We thank the reviewer for raising this issue. We change the text deleting the term “anisotropic” used improperly and we modify the text as follows:
Line 666: “In physiologic condition, the collagen fibers which surrounding the normal epithelial structures are curly and smooth while, during tumor development, a variable amount of the fibers progressively became denser, aligned and stiffer”.
Line 445: Instead of emphasizing the somewhat exotic and still poorly understood subcellular localization of some MMPs in mitochondrial and nuclear compartments, more information on the various MMPs, their different substrates and their regulation would be desirable.
We thank the reviewer for the kind suggestion, so we modify the paragraphs 1.5 Degradation of ECM and MMPs activity adding more information about what requested.
Line 254: “1.5 Degradation of ECM and MMPs activity”
“The ECM components are normally cleaved and degraded by pericellular target-specific remodeling and degrading enzymes. Among them, various proteases such as soluble and membrane-bound metalloproteinases (MMPs), disintegrin and metalloproteinases (ADAMs), disintegrin and metalloproteinase with thrombospondin motifs (ADAMTS), cathepsins, bone morphogenic protein1 and Tolloid-like proteinases, as well as hyaluronindases and heparase are mainly involved. More than the other ECM-degrading enzymes, MMP are of outstanding importance in ECM remodeling, and it are crucially involved in cancer progression [46]. MMPs family comprises 28 members and, depending on their type, these can have gelatinolytic and collagenolytic activity toward ECM. MMPs proteolytic activity is triggered by an activation cascade through which it is secreted as zymogens (pro-MMPs) without biological activity and kept inactive till the disruption of the interaction between the conserved cysteine in the propetide domain and the zin ion bound to the catalytic domain. The activation cascade can include endogenous inhibitors such as the case of MMP-2, which requires a tissue inhibitor of metalloprotineases (TIMP)-2 to be activated by MMP-13. TIMPs are a protein family which, forming a complex with their own N-terminal domain and chelating the catalytic zinc ion in the active center of MMP, function as natural MMP inhibitors. Nevertheless, TIMPs can also interact with their C-terminal domain with the hemopexin domain of MMPs and thereby activate them [47]. In particular, forming a ternary complex of pro-MMP-9 with MMP-3 and TIMP-1 can activate MMP-9, while TIMP-2 contributes to MMP-2 activation forming a complex with proMMP-2 and MMP-14[48]. TIMPs have also been shown to modulate other biological functions such as apoptosis, cell growth, and angiogenesis. Notably, many cancer types exhibit TIMP dysregulation, which affects ECM integrity and promotes metastatic ability [47].
The several MMPs are stringently regulated from transcriptional to post-translational level to maintain their expression and activity limited in spatio-temporal distribution and activity. This regulation process is frequently lost in many cancers [49]. Nevertheless, metabolic condition also influences MMPs activity. Indeed, MMPs increase under hypoxia conditions, in particular hypoxia-inducible factor (HIF)-1 increases the degrading action of MMP-1 and MMP-9 [50]. Further, TME acidification induces translocation of MMP14 to membrane protrusions such as filopodia and invadopodia via small GTPase RhoA [51].
MMPs exert different molecular functions. It can cleave the insoluble ECM components into soluble fragments, can activate or inactivate soluble proteins and can release soluble ectodomains of membrane-bound proteins as autocrine or paracrine signals [52]. Among MMPs, MT1-MMP also known as MMP-14, have the highest number of substrate specificity cleaving the interstitial collagen I, II and III while cannot directly act of type IV, but it is able to activate MMP-2 which, instead, can cleave it, especially during the carcinoma formation [53]. Other membrane-bound MMPs, such as MMP-15 can cleave type I collagen while MMP-16 is able to clean type III collagen and both can also activate pericellular proMMP-2. [54,55]. Nevertheless, MMPs activity is not limited to the degradation of ECM components. Indeed, some MMPs substrates and some cleavage fragments can regulate cell proliferation, differentiation, apoptosis and, further, chemotaxis, migration, and angiogenesis.
Some pericellular molecules are also susceptible to MMP’s cleavage activity such as protease inhibitors like α 1- anti-chymotrypsin, α 1-proteinase inhibitor, α 2-macroglobulin, plasminogen activator inhibitor-1 (PAI-1), plasmin C1-inhibitor, and serine proteinase inhibitor-E2 (SERPINE2) and further other on-matrix molecules like cytokines (e.g. pro-IL-1β, pro-IL-8, Motif Chemochine Ligand 5 (CXCL5), CXCL9, CXCL10, CXCL11 precursor, and CXCL11, CXCL12 (stromal cell-derived factor 1 - SDF) and growth factors (e.g pro-TGF-β).
Additionally, several MMPs substrates are intracellular proteins involved in transcription, translation and apoptosis processes such the case of many cytoskeletal protein nuclear laminins, chaperones, regulators of transcription, translation [55]”.
Furthermore, we decide to discuss the information about MMPs activity in TME in a dedicated paragraph as follows:
Lines 744: “3.4.1 MMPs role in shaping TME”
“Cancer and stromal cells within the tumor mass such as neutrophils, macrophages, lymphocytes, MCs, fibroblasts, endothelial cells, pericytes and adipocytes are the major source of metalloproteinases in TME. All these distinct stromal cells, producing a specific set of metalloproteinases and inhibitors, contribute to intratumoral proteolytic equilibrium. CAFs, for example, beside to produce ECM molecules, also synthesize elevated level of ECM-degrading enzymes such as MMP-1,-2,-3, ‐11, ‐13, ‐14, ‐19, ADAM9, 10, 12, 15 and 17 and ADAMTS5 as well as TIMP1, 2, and 3 [186].
Notably, TME contains various MMP substrates, including native fibrillar collagens and denaturated gelatin as well as laminin, making MMPs pivotal in TME shaping in various ways [55].
The proteolytical MMPs activity and its ECM remodeling plays a crucial role in in varying degrees of tumor dissemination, metastasis cascade and the formation of suitable metastatic niches. Accordingly, the expression and activity of many MMPs are correlate with tumor progression. [187]. Indeed, epithelial-to-mesenchymal transition also depends on the MMPs activity which increase the cell’s ability to migrate and infiltrate. MMPs with a documented role in EMT are MMPs 1,-2,-3,-7,-9,-14 [188] and 28 [189]. In particular, MMP-14 promote a mesenchymal phenotype cleaving BM components and E-cadherin [190]. Regarding the next steps of metastatic cascade, soluble MMPs-1-3 -7, -9, -10, -11, -13, -26, and -28, and the membrane-type MMPs -14 and -16 are mainly involved in migration and invasion [55]. ECM remodeling by MMPs is important also for tumor angiogenesis. In particular, MMPs -1, -2, -3, -8, -9, -10, -11, -13, and -14 can also regulate angiogenic balance and ,vice versa, angiogenic factors such as VEGF, basic fibroblast growth factor (bFGF), TGF- β and α, and angiogenin can induce MMPs activity [189]. Another non-negligible aspect of degradation of ECM components is related to the production of bioactive fragments. These fragments may have pro or anti tumorigenic function independently to the starting full-length ECM component properties. Due to chemokines or cytokines-like structure, these fragments are termed matrikines [191] and, through their promoting and inhibiting properties, they are involved in balancing the angiogenic switch [106]. Further, an excessive MMPs activity observed during tumor progression, causes the release into circulation of small ECM fragments which may be indicative to evaluate tumor activity and invasiveness and could be used as biomarkers [4,192]”.
Line 451: Do filopodia have MMP activity?
We apologize for the misleading sentence. We modify the text as follows:
Line 285: “Further TME acidification induces translocation of MMP14 to membrane protrusions such as filopodia and invadopodia via small GTPase RhoA.”
We also substitute relative reference Muñoz-Nájar, U.M.; Neurath, K.M.; Vumbaca, F.; Claffey, K.P. Hypoxia stimulates breast carcinoma cell invasion through MT1-MMP and MMP-2 activation. Oncogene 2006, 25, 2379–2392.
Lines 551-552, 578: MMP-2 and -9 are gelatinases and cannot degrade collagens.
We thank the reviewer for highlighting this oversight. We have deleted the whole sentence containing the wrong information and we modify the text as follows:
Line 453 “Other evidence about chemotactic function of ECM-derived fragments, concerns the N-acetyl Pro-Gly-Pro (PGP) fragments resulted from MMP-9 mediated hydrolysis of collagen occurred during inflammation “.
The MMP-9 activity in degradation of collagen occurred in inflammation is also reported in reference “Xu X, Jackson PL, Tanner S, Hardison MT, Abdul Roda M, Blalock JE, Gaggar A. A self-propagating matrix metalloproteinase-9 (MMP-9) dependent cycle of chronic neutrophilic inflammation. PLoS One. 2011 Jan 13;6(1):e15781. doi: 10.1371/journal.pone.0015781. PMID: 21249198; PMCID: PMC3020950”
Line 585-586: Neither elastase (isolated from sputum in ref. #124) nor MMP12 (the macrophage elastase) are chemotactic signals in contrast to elastokines that can be generated by these proteases. If the authors mean “MMP12 digests” elastin degradation products, they should make that clear.
We apologize to the reviewer if the sentence was not clear enough. We modify the text as follows:
Line 462: “Further degradation products of elastin have also been shown to be chemotactic for monocytes in chronically inflamed lungs and in particular whose resulted from macrophage elastase MMP12 activity”.
Further we have added the reference reporting the description of this pathophysiological mechanism: “Houghton AM, Quintero PA, Perkins DL, Kobayashi DK, Kelley DG, Marconcini LA, Mecham RP, Senior RM, Shapiro SD. Elastin fragments drive disease progression in a murine model of emphysema. J Clin Invest. 2006 Mar;116(3):753-9. doi: 10.1172/JCI25617. Epub 2006 Feb 9. PMID: 16470245; PMCID: PMC1361346.”
Line 598: The reference "Midwood K. et al." is not given in this somewhat garbled sentence.
We are desolated for the lack. We add the right reference and also modify the text as follows:
Lines 478: “To what concern tenascin C, Midwood K. et al. assessed its activation of TLRs in an in vivo study conducted on a murine model of rheumatoid arthritis (RA). In the in vivo model, like observed in RA patients, the level of tenascin C was upregulated in synovia, synovial fluid, and cartilage. The upregulation of tenascin C was observed to interact with TLR4 on macrophages and synovial fibroblasts, inducing pro-inflammatory cytokines production, such as IL-6, TNF and CXCL8. On the contrary the mices which lacked tenascin C were protected from synovitis that was induced by zyosan (a TRL2 agonist). Thus, promoting the production of pro-inflammatory cytokines, tenascin C is proposed to maintain inflammation in the joint and to locally propagate the inflammatory response”
Line 638: Neither MMP12 nor MT1-MMP are mentioned in reference #147.
We apologize for the mistake, we removed the wrong references and added the following references:
Line 519: ”Khan, K.M.; Falcone, D.J. Role of Laminin in Matrix Induction of Macrophage Urokinase-Type Plasminogen Activator and 92-KDa Metalloproteinase Expression. J. Biol. Chem. 1997, 272, 8270–8275, doi:10.1074/jbc.272.13.8270.”
Line 519: “Corcoran, M.L.; Kibbey, M.C.; Kleinman, H.K.; Wahl, L.M. Laminin SIKVAV Peptide Induction of Monocyte/Macrophage Prostaglandin E2 and Matrix Metalloproteinases. J. Biol. Chem. 1995, 270, 10365–10368, doi:10.1074/jbc.270.18.10365.”
Further we modified the text as follows:
Line 519:” whereas similar peptides Ala-Ser-Lys-Val-Lys-Val (ASKVKV), derived from laminin α5 chain, beside to be a neutrophil chemotactic for neutrophils and macrophage in vitro, also promote the expression of MMP-9 and MMP-14 production by monocytes and macrophages”.
We also added the following references:
Line 522: “Adair-Kirk, T.L.; Atkinson, J.J.; Kelley, D.G.; Arch, R.H.; Miner, J.H.; Senior, R.M. A Chemotactic Peptide from Laminin Alpha 5 Functions as a Regulator of Inflammatory Immune Responses via TNF Alpha-Mediated Signaling. J. Immunol. 2005, 174, 1621–1629, doi:10.4049/jimmunol.174.3.1621”
“Mydel, P.; Shipley, J.M.; Adair-Kirk, T.L.; Kelley, D.G.; Broekelmann, T.J.; Mecham, R.P.; Senior, R.M. Neutrophil Elastase Cleaves Laminin-332 (Laminin-5) Generating Peptides That Are Chemotactic for Neutrophils. J. Biol. Chem. 2008, 283, 9513–9522, doi:10.1074/jbc.M706239200.”
Line 676-677:"...in many cases, cancer cells are considered as "seeds" for cancer initiation [148]." In reference #148, seeding is discussed in the context of metastasis and not cancer initiation.
We apologize for the unproperly use of the term “seed”. We modify the text as follows
Line 559 “This aspect assumes great relevance given that, in many cases, the pathologic clones in the very first phase of pre-malignant lesion are considered as triggers for cancer initiation”.
Lines 688, 711, and others: It is important to discriminate between different MMPs, because they exert very different functions. For example, I suppose that in line 689, MMP9 is meant, whereas in line 711 MMP2 and MT1-MMP are meant.
We apologize for the superficial information provided. We modify the text as follows:
Line 571: “Once activated in inflammatory cells, NF-kB regulates cell cycle mediators (cyclin D1, c-Myc), anti-apoptotic (c-FLIP, survivin, Bcl-XL) and adhesion molecules (ICAM-1, ELAM-1, VCAM-17), proteolytic enzymes, such as MMP9 and uPa and pro-inflammatory cytokines (such as TNF-α, IL-1, and IL-6) which, all together, contribute to inflammation-related tissue damage, tumor development and progression”.
Line 598 “As demonstrate in colon cancer cell line CACO-2, programmed to constitutively express COX-2, were detected an increased invasiveness if compared with the parental CACO-2, along with an activation of MMP-2 and increased RNA levels for MT-MMP-1”.
Further we provide a new reference in the text as follows:
Line 602 “Tsujii, M.;M, Kawano S, DuBois RN. Cyclooxygenase-2 expression in human colon cancer cells increases metastatic potential. Proc Natl Acad Sci U S A. 1997 Apr 1;94(7):3336-40. doi: 10.1073/pnas.94.7.3336. PMID: 9096394; PMCID: PMC20370”.
Line 737: Patient-derived decellularized matrices are indeed very interesting. However, their production is anything but easy. At least a brief description of the protocol would be desirable. It would also be helpful to estimate which matrix components are retained during production or are inevitably lost.
We thank the reviewer for the interesting suggestion. We decided to add a brief description of the protocols to obtain patient-derived decellularized matrices and how the ECM components are retained or removed after decellularization protocols in the text as follows:
Line 939: “Recently, tissue decellularization has emerged as a robust alternative technique in the field of tissue engineering and regenerative medicine [230]. The principal advantage of using biological-derived matrices, instead of synthetic polymers for 3D tumor study in vitro, is that the main structural proteins and soluble factors are already present in decellularized scaffolds [231]. The term decellularization indicate the removal of the cellular component from a tissue by minimally altering its biochemical composition and biological and structural properties [232]. The decellularization of ECM can be achieved by three different approaches: enzymatic, chemical and physical methods. Chemical methods include the perfusion, agitation or immersion with chemical solutions. These methods achieve removal of the majority of cellular materials, but fail to preserve the ECM molecules. That is why, the decellularization process with combined chemical treatments should be optimized for the least damage to ECMs [230]. Physical methods include freezing, direct pressure, osmosis, sonication and agitation. They can be used to decellularized tissues by disrupting cellular membranes, with corresponding cell lysis. Physical prevent the disruption of ECM structures, but since these techniques only disintegrate cellular membranes, but do not remove cellular debris, requiring the adding of enzymatic or chemical treatments to obtain acellular tissues [233]. Enzymatic decellularization uses nucleases, proteases, and chelating agents (necessary for the removal or separation of cellular debris). Enzymatic approaches can effectively remove cellular materials preserving most of the collagen components. However, it is necessary to set a maximum treatment time to mitigate the disruptive effects on the architecture and composition of the ECM during the decellularization process [233]”.
Line 749 : Some information on the different cell lines (e.g., the type of the cancer from which they are derived) would be helpful in a (missing) legend of the table. MCF-7 definitely differs for MDA-MB-231, yet both are indiscriminately described here as breast cancer cells.
Following the reviewer's comment, we decided to indicate in Table 3 the type of cell lines used in each article and the respective abbreviations as follows at line 975:
dECM origin |
dECM tissue |
Cells cultured on dECM |
Results |
Ref. |
Human |
CRC |
Human colon adenocarcinoma cells
(HT-29 – HCT-116) |
-3D CRC model exhibited reduced sensitivity to 5-Fluoruracil and FOLFIRI treatments compared with 2D conventional cell cultures. |
[13] |
Human |
HC - CRC – Healthy Liver (HL) – CRC and liver metastasis (CRLM) |
Human colon adenocarcinoma cells (HT-29 ZsGreen/Luc+ and HCT-116 cells)
|
-Increased migration ability of CRC cells observed in CRC and CRLM scaffolds if compared to HL scaffolds. -At an increasing concentration of drug, the response of the CRC cell line to chemotherapy was scaffold-dependent. HT-29 cells grown in HL scaffolds exhibited a significant reduction in cell proliferation compared to untreated scaffolds. |
[238] |
Human |
CRC |
Human colon adenocarcinoma cells (HCT-116 and HT-29) |
-Tumor cells and the decellularized tumor matrix induce the differentiation of monocytes in macrophages, characterized by high surface expression of CD206 and reduced expression of MHC-II and CD86. -The exposition of macrophage to tumor cells and to tumor ECM induce a higher expression of IL-6, IL-10, TGF- CCL17, CCL18, and CCL22. |
[239] |
Human |
Healthy colon and CRC |
Human colon adenocarcinoma cells (HT-29 cells) |
-Over-expression of IL-8 in 3D tumor matrices after 5 days of recellularization with HT-29 cells. -Secretoma analysis showed a marked increase of DEFA3 in tumor stroma compared to healthy counterpart. |
[240] |
Human |
PDAC |
Human pancreatic adenocarcinoma cell lines (Panc-1 and AsPC-1) |
-Immunofluorescence analyses after 7 days of scaffolds recellularization with Panc-1 and AspC-1, confirmed the biocompatibility of 3D matrices to sustain engraftment, localization and infiltration. -Both PANC-1 and AsPC-1 cells cultured in 3D matrices showed a reduced response to treatment with FOLFIRINOX if compared to conventional bi-dimensional culture. |
[241] |
Human |
Esophagus from cadavers |
Human bone marrow cell line (HM1-SV40) |
- After the 72 hours of exposure period to culture media conditioned with decellularized esophageal samples, HM1-SV40 cells appeared to be viable and proliferating, reaching about 90% confluence on the growth surface. The MTT assay revealed that, compared with untreated cultures, cells preserved 93.7%, 95.4% and 78.3% viability when treated with medium conditioned with esophageal matrices decellularized through Protocols No. 1, No. 2 and No. 3, respectively). |
[242] |
Rat |
Esophageal adenocarcinoma (EAC) |
Human mononuclear cells (THP-1) and oesophageal eipithelial cells (Het-1A) |
-Metaplastic and neoplastic esophageal ECM induce distinctive effects upon THP-1 macrophage signaling compared to normal esophageal ECM. -The secretome of macrophages pre-treated with metaplastic and neoplastic ECM increases the migration of normal esophageal epithelial cells, similar behavior to that shown by tumor cells. |
[243] |
Pig |
Breast cancer |
Human BC cell line (MDA-MB- 231) 3D tumor spheroids Human dermal fibroblasts |
- Metastatic breast cancer model which constitute a TME-mimetic platform comprising breast tissue-derived porcine d ECM; cancer cell assembled to form spheroids. In vitro generated monotypic and heterotypic dECM-Spheroids model exhibited the invasive profile of metastatic breast cancer cells and evidenced exometabolomic signatures similar to those observed for human tumor translated in differences to cisplatin, gemcitabine and Palbociclib cancer-cell specific response. |
[244] |
Human |
Breast Cancer |
Human breast cancer cells (MCF7, T‐47D, and MDA‐MB‐231) |
Cells cultured in patient-derived scaffolds showed a higher drug resistance if compared to the 2D cultures, demonstrating that the 3D environment improves cell chemoresistance. |
[245] |
Human and Animal |
Different Tissue/organ-derived |
Human cancer cells, human monocytes and fibroblasts. |
-dECM will provide new insights into the role of ECM in oncogenesis and cancer progression. -dECM represents a culture substrate for pharmacological and pharmacokinetic analyses to develop novel anticancer drugs. -proteomic analysis of dECM will lead to the identification of candidates for anticancer drugs. |
[246] |
Human and murine
|
Normal and Tumor biopsies |
Fibroblastic cell derived matrices (fCDM)
|
-The method offers the possibility to obtain 3D matrices with relevant characteristics for mimicking particular in vivo microenvironments (e.g. normal vs. desmoplastic or fibrous). -fCDMs can be used to scrutinize cocultures of different cell types (tumor, stroma, immune neural etc.) that co-exist in the interstitial microenvironment of diverse tissues. -fCDM have shown to serve as a useful platform to establish pre-clinical studies. |
[247] |
Line 764: I understand that three-dimensional dECMs are suitable to investigate processes of chronic inflammation in cancer progression. However, I do not understand how they might be used as a "possible prevention tool" as stated by the authors
We thank the reviewer for raising this perplexity. Maybe the concept expressed is too ambitious for a working progress model. To better express our idea, we modify the text as follows:
Line 980: “These models could represent a potential tool to investigate in the deep on inflammatory chronic diseases and useful to explore new patient-specific therapeutic approaches, reducing inflammation condition and, potentially, also decreasing the risk of cancer initiation”.
Minor points:
Line 125: The name is “Dvorak” and not “Dvorakin”
We apologize for the mistake , we modified and wrote “Dvorak” at line 116.
Line 126: “would healing” – wound healing?
We apologize for the mistake , we modified and wrote “wound healing” at line 117.
Line 143: What is meant by “cancer risk initiation”?
We thank the reviewer’s comment, we modified LINE 133 as follows “chronic inflammation which increases the risk for cancer initiation”.
Line 161: “manly”?
We apologize for the mistake , we modified and wrote “mainly” at lines 162 and 177.
Line 188: what is meant by “ECM stroma”?
We apologies with the reviewer for the improper use of the term “stroma”. We meant the whole set of ECM components. We have substituted the term at line 183 with “ECM structure”.
Often more words than necessary are used. For example, line 193: “intracellular cytoskeleton”. The cytoskeleton is of course within a cell. Or in line 464-466: “production of bioactive shorter fragments”. Fragments are always shorter than the original molecule.
Following the reviewer's comment, at line 190 we delete the word “intracellular” and keep only “cytoskeleton” nouns, while at line 769 “production of bioactive fragments”.
Line 197: “cell surface receptors such as integrin, collagen and fibrin” sounds misleading.
Following the kind reviewer's suggestion about the length of some parts of our review, we decided to focus our attention on the central argument of our review. For this reason, the line cited was deleted and we report the ECM components, functions and properties in Table 1 as follows at line 157:
Components |
Functions and Properties |
Proteins and Structure |
|
Collagens. Formed as fibrils within the ECM (Collagen I, II, III, V and XI) |
ECM architecture, mechanical properties [20]. Influence cell process such as adhesion and migration. Binding of extracellular growth factors and cytokine [21]. |
Elastin Composed of single tropoelastin subunit cross-linked |
Tissue elasticity, load bearing and storage of mechanical energy [22]. |
Glycoproteins |
|
Laminins Trimeric structure made of three different chain α, β, γ |
Resident in the basement membrane. It play a role in cell adhesion, survival, migration, differentiation and migration via integrin [22]. |
Fibronectin (FN) Arranged into a mesh of fibrils to collagen and is linked to cell surface receptors (integrins) |
Binds to collagen, fibrin and glycosaminoglycans, influencing cell adhesion, growth, migration, wound healing and differentiation [23]. |
Fibrillins |
Scaffolds for elastin deposition [23]. |
GAGs |
|
Hyaluronic acid (HA) |
Regulates numerous cell functions and biological processes through the interaction with cell surface receptors. Lend tissue turgor and facilitates cell migration during tissue morphogenesis and repair. Intact high molecular weight of HA has antiangiogenic and anti-inflammatory properties, whereas lower HA fragments exhibit completely opposite functions [24]. |
PGs |
|
e.g Heparan, chondroitin and keratin sulphates. Core of protein domain covalently linked to GAGs. Negavitely charged it can bind water. |
Provide compressive resistance to tissue. Reservoir and sequester of growth factors, chemokines and cytokines at the ECM protecting them from degradation and creating effective gradients along ECM [25]. |
Matricellular proteins |
|
Thrombospondins (TSPs), tenascins (TNs), fibulins, osteopontin (OPN), cartilage oligomeric matrix protein (COMP), CNN family proteins, periostin, and R-spondins |
Cell behaviour modulation, mediation of the interaction between cellular transmembrane receptors and fibrous ECM molecules and key functional regulator of cell-matrix communication [26]. |
Line 223: What does the term “(glycol) proteins” mean?
We thank the reviewer for notifying us of the error and even in this case the line cited was deleted. The proteins indicated were cited in table 1 as Matricellular protein category.
Line 240: "ligand biding"?
Following the reviewer's comment, we modified and wrote “ligand binding” at line 195.
Line 290: "linearized collagen" and line 372: “linearized fibers”? I assume that parallel alignment is meant.
We thank the reviewer for raining the imprecisions. As the the revisor supposed with the term “linearized” we indicated the parallel orientation of fibers. To clarify what we meant we modified as follows:
Line 322: “This feature explains why linearization of cross-linked collagen bundles promotes cell migration, whereas the presence of denser network of stiff cross-linked matrix fibers, instead, impedes migration unless matrix metalloproteinases are simultaneously activated.”
Line 668: “The increase of parallel orientation in collagen fibers increases its density and concentration, stiffening the E.CM”
Line 387: “integrin grouping” – clustering?
Following the reviewer's comment, we modified and wrote “integrin clustering” at line 684.
Line 402, 404: “LTBP-LAP”, “TGB-beta1” -A list of abbreviations would be helpful to the reader. Line 421: ROS = reactive oxygen species; line 731: ROS=retinoic orphan receptor: Different abbreviations would be useful here. Does the abbreviation in line 731 make sense, since this expression is only used once? Table 1: what does “FOLFIRI” stand for?
Following the reviewer's comment, we created an abbreviations list that should help the reader to understand as follows at line 1036:
2D (two-dimensional)
3D (three-dimensional)
ADAMs (disintegrin and metalloproteinases)
ADAMTS (thrombospondin motifs)
ASKVKV (Ala-Ser-Lys-Val-Lys-Val)
BFGF (fibroblast growth factor)
BM (basement membranes)
BMP (bone morphogenetic protein)
CXCL (C-X-C motif ligand)
CAC (colitis-associated colon cancer)
CAFs (cancer-associated fibroblasts)
COMP (cartilage oligomeric matrix protein)
CRC (colorectal cancer)
CRLM (colorectal liver metastasis)
DAMPs (damage-associated molecular patterns)
DDR (DNA damage response)
DDrs (discoidin domain receptors)
dECM (decellularized ECM)
EAC (esophageal adenocarcinoma)
EBP (elastin binding protein)
ECM (extracellular matrix)
EGF (epidermal growth factor)
EHS (Engelbreth-Holm-Swarm) c
EMT (epithelial to mesenchymal transition)
ERK (extracellular-regulated kinase)
FAK (focal adhesion kinase)
FGF-2 (fibroblast growth factor)
FMLP (formyl met-leu-phe)
FOLFIRI (folinic acid, fluorouracil acid and irinotecan)
FOLFIRINOX (folinic acid, fluoruracil acid, irinotecan and oxaliplatin)
GAGs (glycosaminoglycans)
GLUT (glucose transporters)
GM-CSF (Granulocyte-Macrophage Colony-Stimulating Factor)
HA (hyaluronic acid)
HIF (hypoxia-inducible factor)
HL (Healthy Liver)
IBD (inflammatory bowel disease)
IFNγ (interferon gamma)
ILs (interleukins)
IM (interstitial matrix)
IME (inflammatory microenvironment)
LAP (latency-associated protein)
LOX (lysyl oxidases)
LOXLs (LOX-like proteins)
MCs (mast cells)
MT1-MMP (membrane-type 1 matrix metalloproteinase)
MHs (microporous hydrogels)
MIP (macrophage inflammatory protein)
MMPs (metalloproteinases)
NETs (neutrophil extracellular traps)
NF-kB (nuclear factor kappa-light-chain-enhancer of activated B cells) (
OPN (Osteopontin)
PAI (plasminogen activator inhibitor)
PAMPs (pathogen-associated patterns)
PDAC (pancreatic ductal adenocarcinoma)
PDGF (platelet-derived growth factor)
PDGF-EC (platelet derived endothelial cell growth factor)
PGP (Pro-Gly-Pro)
PGs (proteoglycans)
PPARs (peroxisome proliferator-activated receptors)
PGE2 (Prostaglandin E2)
PVDF (polyvinylidene fluoride)
RA (rheumatoid arthritis)
RNI (reactive nitrogen intermediate)
ROCK1 (Rho‐associated protein kinase 1)
RONS (nitrogen species reactive oxygen)
ROS (reactive oxygen species)
SDF (stromal cell-derived factor 1)
SERPINE2 (serine proteinase inhibitor-E2)
SIKVAV (Ser-Ile-Lys-Val-Ala-Val)
Src (steroid receptor coactivator)
TAMs (Tumor associated macrophages)
TANs (tumor-associated neutrophils)
TGF-β (transforming growth factor β)
Th17 (T-helper IL-17 producing cells)
TINKs (tumor-infiltrating natural killer cells)
TIMP (tissue inhibitor of metalloproteinases)
TLRs (Toll-like receptors)
TME (tumor microenvironment)
TNF-α (tumor necrosis factor α)
TNFR (tumor necrosis factor receptor superfamily)
Tregs (Regulatory T cells)
TSPs (Thrombospondins)
VEGF (vascular endothelial growth factors)
Regarding the line 622, we modify the abbreviation as follow “RORs”.
Line 440: “cleaved degraded”?
Following the reviewer's comments, we decide to modify in the text as follows:
line 256: “The ECM components are normally cleaved and degraded by pericellular target-specific remodeling and degrading enzymes.”
Lines 511, 680: “addiction”?
We apologize for the typos, we modified and wrote “addition” at lines 670, 827 and 878.
Line 514: “side” or site?
We apologize for the mistake, we modified and wrote “site” at line 829.
Line 583-585: what “list“ is meant here? Besides, the grammar is wrong.
Following the reviewer's comment, we modified and wrote “proceeding through the description” at line 460.
Lines 607, 608: “bind”?
We apologize for the mistake, we wrote “binding” at lines 488 and 489.
Line 735: "... and its related mainly related to the promotion angiogenesis ..." - What statement is intended here? In addition, many other sentences are very long and yet incomplete: For example, lines 72-75, 184-186, 610-611, 621-623
We apologize for the mistake, we decided to delete the whole sentence at the cited line because it needed deeper explanations which emulate from the aims of our review and we modified the text as follows:
Line 70: “Indeed, in course of tumor progression, the acquisition of mutations in cancer cells and the changing of surrounding stromal cells phenotype impacts on ECM, leading to bio-functional and biomechanical changes''.
Line 170: “In each existing version, ECM is not mere intercellular filling, but represents a bioactive element with multiple functions related to its physical, biochemical and biomechanical properties that are essential for regulating cell behavior. Regarding its physical properties, the molecular architecture, rigidity, porosity, insolubility, spatial arrangement and orientation, allows the function of ECM to shape and maintain the form of tissue and organs integrity.”
Line 473: “The activation of TLRs triggers innate immune responses which also influence the adaptive response in turn. Several fragments of the ECM function as activator such those derived from interstitial matrix like tenascin C isoform, the small leucin-rich proteoglycan biglycan [90], fibronectin [91], heparan sulphate [92,93]and HA[94]”.
Line 484 “Thus, promoting the production of pro-inflammatory cytokines, tenascin C maintains inflammation in the joint and locally propagates the inflammatory response [95].”
758: "From our review resulted a laking data ..."?
Following the reviewer's comment, we modified and wrote “Our literature review” at line 1003.
In the figure legend "TGF-beta, FGF2, PDGF" are mentioned, whereas in the figure "TGF-beta, TNF-alpha, IFN-gamma" are shown.
We apologize for the mistake, we modified and wrote “TGF-beta, TNF-alpha, IFN-gamma” at line 1019.

Reviewer 2 Report
This is a reasonable topic for review - there are lots of reviews related to ECM and cancer, but the link between inflammation and cancer is less clear, and figure 1 is a good schematic. I have some suggestions for the authors:
- Please proof-read English as language is unclear in places.
- This occasionally makes some points inaccurate/difficult to follow. An example would be lines 675-677 which I did not understand.
- The links between ECM stiffness and tumour suppressors in lines 424-426 are not clearly explained, not clear how these concepts link together here.
- Also unclear lines 427-428 how ECM stiffness induces hypoxia. Needs explaining.
- I would recommend re-ordering the review to discuss IME before cancer, as logically this would seem to be the chronological order of events?
- I suggest including a little more information on the role of macrophages/immune cells and their contribution to ECM remodelling pre-cancer since the review is detailing inflammatory links to cancer.
- line 742 explain what "2D limitations" are. In what way?
- check details in table 1 - I was a bit confused by some of the descriptions which don't seem to relate closely to the cells listed. Eg the example linked to reference 14 mentions SW480/SW620/RKO cells but description seems to discuss immune cell components. Also reference 193 is provided in one line, but text describes other work (Baker et al), referencing seems inconsistent. This table is useful and needs to be clear.
-
Author Response
Response to Reviewer 2 Comments
Oct 28th, 2022
Dear Editor,
we kindly thank to all the reviewers for the brilliant suggestions.
We carefully considered all the comments by the Reviewers and we revised the manuscript accordingly. We feel that the manuscript has been significantly improved thanks to the Reviewers’ comments and we hope the paper will now be considered acceptable for publication. In accordance with Reviewer’s requests, we decided to make extensive changes in manuscript’s schedule. To make more readable and to streamline the text, we decided to modify the structure of the review as follows:
1.Introduction
1.1. ECM More Than A Physical Support: An Overview
1.2. ECM Major Components, Properties and Functions
1.3. ECM-Cells Interplay
1.4 ECM deposition
1.5 Degradation of ECM and MMPs activity
1.6 ECM dynamics under pathological conditions
- Role of ECM in IME
2.2 Immune cells components drive ECM remodeling during inflammation2.3 ECM-derived fragments as modulators of chemotactic activity for immune cells
2.4 Alteration in ECM components affect immune cells activity.
2.5 ECM-fragments influence gene expression of inflammatory cells.
- From chronic inflammation to cancer: the main correlations
3.1. Chronic inflammation and tumor initiation
3.2. Main inflammatory pathways correlated to tumorigenesis and colitis associated cancer
3.3 ECM deposition in TME
3.4 Tumorigenic Alterations in ECM composition
3.4.1 MMPs role in shaping TME
3.5 Tumorigenic ECM and its remodeling influence immune cells within tumor mass.
- Pre-clinical models application in TME and IME
4.1 Scaffold-free and Scaffold-based systems
4.2 Patient-derived scaffold obtained by tissue decellularization
4.1 3D patient-derived dECM models: novel strategies to study IME to prevent TME?
- Conclusions and Future Perspectives
Further, following the Reviewer’s suggestions we provided table 1 for ECM components properties and functions, table 2 for stromal cells description. In conclusion, as cleverly suggested, we updated the table 3 for 3D repopulated decellularized ECM models and we added an abbreviations list.
This is a reasonable topic for review - there are lots of reviews related to ECM and cancer, but the link between inflammation and cancer is less clear, and figure 1 is a good schematic. I have some suggestions for the authors:
Please proof-read English as language is unclear in places. This occasionally makes some points inaccurate/difficult to follow. An example would be lines 675-677 which I did not understand.
We apologize for the unproperly use of the term “seed” and we modify the text as follows:
Line 559: “This aspect assumes great relevance given that, in many cases, the pathologic clones in the very first phase of pre-malignant lesion are considered as triggers for cancer initiation”.
The links between ECM stiffness and tumour suppressors in lines 424-426 are not clearly explained, not clear how these concepts link together here.
We apologize with the reviewer for the unclear explanation. We provide some different reference and we further modify the text as follows:
Line 718: “Accumulation of these oxidative molecules, caused by an imbalance between antioxidants system and ROS and RNS generation, lead to an alteration in gene expression levels, signal transduction pathways which regulate cancer cell proliferation, invasion, and apoptosis. These oxidative molecules can activate all three members of MAPK family stress-responsible protein kinases including ERK1/2, JNK, and p38 through the oxidative modification of protein tyrosine phosphatases that dephosphorylate MAPKs. Stimulation of MAPK through ROS and RSN accumulation improves proliferation, migration, and invasion of human breast, liver, prostate, lung, skin, and pancreatic cancer cells [181]. Further, ROS induced PI3K/Akt signaling which plays a key role in acquisition of malignant phenotype by normal cells and survival of cancer cells through Akt activation or inactivation of tumor suppressors such as PTEN or PTPs (protein tyrosine phosphatases) [182].”
Also unclear lines 427-428 how ECM stiffness induces hypoxia. Needs explaining.
We thank the reviewer for raising the issue. We modify the text as follows:
Line 729: “Induction of hypoxia within the TME acts also as a barrier for the uptake and delivery of chemotherapeutic drugs. Migration of cancer cells is also improved in hypoxic conditions through MAPK induction and ROS accumulation. Indeed, MAPK/ERK pathway increases caveolin-1 expression in cancer cells and, through the induction of RhoA/Rho‐associated protein kinase 1 (ROCK1), promotes cell contraction and favors cancer cell migration along the pre-existing collagen matrix [50,183]”.
I would recommend re-ordering the review to discuss IME before cancer, as logically this would seem to be the chronological order of events?
We thank the reviewer for the interesting suggestion. We decided to accept your suggestion describing the role of ECM in IME first and consequently in the TME as anticipated in the preamble. Therefore, the new structure of the review could be observed as indicated at the beginning of the reply.
I suggest including a little more information on the role of macrophages/immune cells and their contribution to ECM remodelling pre-cancer since the review is detailing inflammatory links to cancer.
Following the reviewer’s suggestion, we have decided to report a better description on the role of macrophages/immune cells in paragraph 2.1 as follows:
Line 338: “The immune microenvironment and ECM components interactions. The immune microenvironment is strongly influenced by ECM features and many immune cells type have key role in ECM deposition and remodeling process, in homeostasis as well as in pathological condition. Inflammation is a multifactorial network of chemical signals which initiate and maintain a host response aimed to heal the injured tissue. Inflammation process involves primary cell players such as neutrophils, monocyte-macrophage, mast cells (MCs) and T lymphocytes also produce cytokines, chemokines, growth factors and proteases which support proliferation of epithelial cells as well as production and remodeling of ECM by fibroblasts [60]. Fibroblasts have key role in restoring the ECM homeostasis during inflammation and have also a crucial crosstalk with the immune system. The outcome inflammation process is related to the tightly regulated interaction between all cell type involved as well as with the ECM. Despite the process through which immune cells mediate the resolution of inflammation is tightly regulated and self-limiting this can be hampered leading to abnormalities. Indeed, the profile of cytokines and chemokines persisting at the inflammatory site is decisive for the development of chronic disease which provoke ECM alteration [14]. In inflamed tissue infiltrating cells release cytokines, such as tumor necrosis factor α (TNF-α), interferon gamma (IFNγ) and TGFβ, which influence both ECM turnover and protease secretion by tissue-resident cells, thus modulating the expression of a wide range of ECM molecules. On the other hand, the aberrant expression of ECM components can influence immune cell activation, differentiation, and survival. Therefore, the remodeled ECM of inflamed tissue influence the perpetuation of inflammatory response and its chronicization”.
line 742 explain what "2D limitations" are. In what way?
We thank the reviewer for the interesting question and we added 2D limitations description in the text as follows:
Line 870: “Nevertheless, the 2D setting limitation have been increasingly recognized, as a fully reliable pre-clinical tumor model, mainly as an over-simplified version of tumor conditions in vivo, often failing to address many of the more dominant pathological problems, such as the tumor microenvironment (TME).[212] Monolayer cells do not grow in a physiological environment that allows them to assume the different shapes and behaviours observed in vivo. In 2D cultures, cells are forced to polarize and increase their exchange area to culture media due to the attachment to rigid and flat substrates [211]. This leads to an over-nutrition, over-oxygenation and non-reproducibility of the in vivo molecular gradients. In addition, in 2D settings, the composition, configuration and production of ECM are significantly altered [213]. It was estimated that only 5% of drugs in cellular models were found to be active in clinical trials [214]”.
check details in table 1 - I was a bit confused by some of the descriptions which don't seem to relate closely to the cells listed. Eg the example linked to reference 14 mentions SW480/SW620/RKO cells but description seems to discuss immune cell components. Also reference 193 is provided in one line, but text describes other work (Baker et al), referencing seems inconsistent. This table is useful and needs to be clear.
Following the reviewer's comment, we decided to update the experimental study in Table 3 and we have modified its description and organization, based on the tumor type. We have indicated the type of cell lines used in each article and the respective abbreviations as follows at line 975:
dECM origin |
dECM tissue |
Cells cultured on dECM |
Results |
Ref. |
Human |
Colorectal cancer (CRC) |
Human colon adenocarcinoma cells
(HT-29 – HCT-116) |
-3D CRC model exhibited reduced sensitivity to 5-Fluoruracil and FOLFIRI treatments compared with 2D conventional cell cultures. |
[13] |
Human |
HC - CRC – Healthy Liver (HL) – CRC and liver metastasis (CRLM) |
Human colon adenocarcinoma cells (HT-29 ZsGreen/Luc+ and HCT-116 cells)
|
-Increased migration ability of CRC cells observed in CRC and CRLM scaffolds if compared to HL scaffolds. -At an increasing concentration of drug, the response of the CRC cell line to chemotherapy was scaffold-dependent. HT-29 cells grown in HL scaffolds exhibited a significant reduction in cell proliferation compared to untreated scaffolds. |
[238] |
Human |
CRC |
Human colon adenocarcinoma cells (HCT-116 and HT-29) |
-Tumor cells and the decellularized tumor matrix induce the differentiation of monocytes in macrophages, characterized by high surface expression of CD206 and reduced expression of MHC-II and CD86. -The exposition of macrophage to tumor cells and to tumor ECM induce a higher expression of IL-6, IL-10, TGF- CCL17, CCL18, and CCL22. |
[239] |
Human |
Healthy colon and CRC |
Human colon adenocarcinoma cells (HT-29 cells) |
-Over-expression of IL-8 in 3D tumor matrices after 5 days of recellularization with HT-29 cells. -Secretoma analysis showed a marked increase of DEFA3 in tumor stroma compared to healthy counterpart. |
[240] |
Human |
Pancreatic ductal adenocarcinoma (PDAC) |
Human pancreatic adenocarcinoma cell lines (Panc-1 and AsPC-1) |
-Immunofluorescence analyses after 7 days of scaffolds recellularization with Panc-1 and AspC-1, confirmed the biocompatibility of 3D matrices to sustain engraftment, localization and infiltration. -Both PANC-1 and AsPC-1 cells cultured in 3D matrices showed a reduced response to treatment with FOLFIRINOX if compared to conventional bi-dimensional culture. |
[241] |
Human |
Esophagus from cadavers |
Human bone marrow cell line (HM1-SV40) |
- After the 72 hours of exposure period to culture media conditioned with decellularized esophageal samples, HM1-SV40 cells appeared to be viable and proliferating, reaching about 90% confluence on the growth surface. The MTT assay revealed that, compared with untreated cultures, cells preserved 93.7%, 95.4% and 78.3% viability when treated with medium conditioned with esophageal matrices decellularized through Protocols No. 1, No. 2 and No. 3, respectively). |
[242] |
Rat |
Esophageal adenocarcinoma (EAC) |
Human mononuclear cells (THP-1) and oesophageal eipithelial cells (Het-1A) |
-Metaplastic and neoplastic esophageal ECM induce distinctive effects upon THP-1 macrophage signaling compared to normal esophageal ECM. -The secretome of macrophages pre-treated with metaplastic and neoplastic ECM increases the migration of normal esophageal epithelial cells, similar behavior to that shown by tumor cells. |
[243] |
Pig |
Breast cancer |
Human BC cell line (MDA-MB- 231) 3D tumor spheroids Human dermal fibroblasts |
- Metastatic breast cancer model which constitute a TME-mimetic platform comprising breast tissue-derived porcine d ECM; cancer cell assembled to form spheroids. In vitro generated monotypic and heterotypic dECM-Spheroids model exhibited the invasive profile of metastatic breast cancer cells and evidenced exometabolomic signatures similar to those observed for human tumor translated in differences to cisplatin, gemcitabine and Palbociclib cancer-cell specific response. |
[244] |
Human |
Breast Cancer |
Human breast cancer cells (MCF7, T‐47D, and MDA‐MB‐231) |
Cells cultured in patient-derived scaffolds showed a higher drug resistance if compared to the 2D cultures, demonstrating that the 3D environment improves cell chemoresistance. |
[245] |
Human and Animal |
Different Tissue/organ-derived |
Human cancer cells, human monocytes and fibroblasts. |
-dECM will provide new insights into the role of ECM in oncogenesis and cancer progression. -dECM represents a culture substrate for pharmacological and pharmacokinetic analyses to develop novel anticancer drugs. -proteomic analysis of dECM will lead to the identification of candidates for anticancer drugs. |
[246] |
Human and murine
|
Normal and Tumor biopsies |
Fibroblastic cell derived matrices (fCDM)
|
-The method offers the possibility to obtain 3D matrices with relevant characteristics for mimicking particular in vivo microenvironments (e.g. normal vs. desmoplastic or fibrous). -fCDMs can be used to scrutinize cocultures of different cell types (tumor, stroma, immune neural etc.) that co-exist in the interstitial microenvironment of diverse tissues. -fCDM have shown to serve as a useful platform to establish pre-clinical studies. |
[247] |

Reviewer 3 Report
In this current manuscript, Marangio et al elaborately summarized the evidences that deal with the role of ECM in cancer initiation from a pathological chronic inflammation condition. Chronic inflammation often leading to oncogenic transformation in stomach, pancreases, colon and liver and various other organs. How inflammation leading to cancer initiation is still an unsolved question, despite numerous studies in last 2 decades. This review highlights the major studies done in this field. They organized, summarized the evidences and draw conclusion perfectly. The manuscript written nicely and sufficiently informative. I personally enjoy reading it, and hope will attract attention from other researchers involve in inflammation biology. Following some comments to improve the quality of this manuscript.
Major comment:
No
Minor comments:
1. Targeting ECM component in Chronic inflammatory disease condition can be a good paragraph to add.
2. In figure 1, labeling of major immune cells type in IME to TME conversion will improve the quality of the figure. (m1/2, Tc, Treg, NK cells)
3. Please discuss about TIMP in 2.4 section.
4. How ECM regulate M1 to M2 conversion during IME to TME conversion. Can author add few studies?
5. Typo mistake in line 25, 126.
Author Response
Response to Reviewer 3 Comments
Oct 28th, 2022
Dear Editor,
we kindly thank to all the reviewers for the brilliant suggestions.
We carefully considered all the comments by the Reviewers and we revised the manuscript accordingly. We feel that the manuscript has been significantly improved thanks to the Reviewers’ comments and we hope the paper will now be considered acceptable for publication. In accordance with Reviewer’s requests, we decided to make extensive changes in manuscript’s schedule. To make more readable and to streamline the text, we decided to modify the structure of the review as follows:
1.Introduction
1.1. ECM More Than A Physical Support: An Overview
1.2. ECM Major Components, Properties and Functions
1.3. ECM-Cells Interplay
1.4 ECM deposition
1.5 Degradation of ECM and MMPs activity
1.6 ECM dynamics under pathological conditions
- Role of ECM in IME
2.2 Immune cells components drive ECM remodeling during inflammation2.3 ECM-derived fragments as modulators of chemotactic activity for immune cells
2.4 Alteration in ECM components affect immune cells activity.
2.5 ECM-fragments influence gene expression of inflammatory cells.
- From chronic inflammation to cancer: the main correlations
3.1. Chronic inflammation and tumor initiation
3.2. Main inflammatory pathways correlated to tumorigenesis and colitis associated cancer
3.3 ECM deposition in TME
3.4 Tumorigenic Alterations in ECM composition
3.4.1 MMPs role in shaping TME
3.5 Tumorigenic ECM and its remodeling influence immune cells within tumor mass.
- Pre-clinical models application in TME and IME
4.1 Scaffold-free and Scaffold-based systems
4.2 Patient-derived scaffold obtained by tissue decellularization
4.1 3D patient-derived dECM models: novel strategies to study IME to prevent TME?
- Conclusions and Future Perspectives
Further, following the Reviewer’s suggestions we provided table 1 for ECM components properties and functions, table 2 for stromal cells description. In conclusion, as cleverly suggested, we updated the table 3 for 3D repopulated decellularized ECM models and we added an abbreviations list.
In this current manuscript, Marangio et al elaborately summarized the evidences that deal with the role of ECM in cancer initiation from a pathological chronic inflammation condition. Chronic inflammation often leading to oncogenic transformation in stomach, pancreases, colon and liver and various other organs. How inflammation leading to cancer initiation is still an unsolved question, despite numerous studies in last 2 decades. This review highlights the major studies done in this field. They organized, summarized the evidences and draw conclusion perfectly. The manuscript written nicely and sufficiently informative. I personally enjoy reading it, and hope will attract attention from other researchers involve in inflammation biology. Following some comments to improve the quality of this manuscript.
Major comment:
No
Minor comments:
Targeting ECM component in Chronic inflammatory disease condition can be a good paragraph to add.
We kindly thank the reviewer for the important point of reflection. Therefore, in our review we mainly focused on the discussion of ECM role in chronic inflammation and its contribution to cancer initiation, trying to provide information to sustain the value of dECM models. Indeed, these models are widely used for cancer research, but less information are provided for chronic inflammation diseases studies. We think that only when these models will become more solid in the study of chronic inflammation diseases, the investigation of targeting ECM components for therapeutic approaches could be possible. Thus, this could represent a good starting point for a next publication.
In figure 1, labeling of major immune cells type in IME to TME conversion will improve the quality of the figure. (m1/2, Tc, Treg, NK cells)
We thank the reviewer for the interesting suggestion. We have labelled the major immune cells type in conversion from IME to TME in figure 1 as follows:
- Please discuss about TIMP in 2.4 section.
We updated the 2.4 paragraph in 1.5 and following the reviewer’s comment, we added the TIMP description in the text as follows:
line 268: “The activation cascade can include endogenous inhibitors such as the case of MMP-2, which requires a tissue inhibitor of metalloprotineases (TIMP)-2 to be activated by MMP-13. TIMPs are a protein family which, forming a complex with their own N-terminal domain and chelating the catalytic zinc ion in the active center of MMP, function as natural MMP inhibitors. Nevertheless, TIMPs can also interact with their C-terminal domain with the hemopexin domain of MMPs and thereby activate them [47]. In particular, forming a ternary complex of pro-MMP-9 with MMP-3 and TIMP-1 can activate MMP-9, while TIMP-2 contributes to MMP-2 activation forming a complex with proMMP-2 and MMP-14[48]. TIMPs have also been shown to modulate other biological functions such as apoptosis, cell growth, and angiogenesis. Notably, many cancer types exhibit TIMP dysregulation, which affects ECM integrity and promotes metastatic ability [47]”.
- How ECM regulate M1 to M2 conversion during IME to TME conversion. Can author add few studies?
We thank the reviewer for the suggestions to enrich our review on this specific topic. The polarization and role of M1 and M2 macrophages in IME have been discussed as follows:
Line 403: “Proceeding along the inflammation process, once spent, neutrophils secrete many chemoattracts cytokines, e.g., CCL-2 (MCP-1), CCL-3 (MIP1α), CCL-4 (MIP-1β), CCL-5, TSP-1, IL-1, IL-6, and TNF-α [68] [69], which recruit monocyte in situ and inducing them to differentiate in mature macrophage and dendritic cells. At the initial stage, stimulation of most macrophages mediated by IFN -γ TNF -α and Granulocyte-Macrophage Colony-Stimulating Factor (GM-CSF) are polarized toward a pro-inflammatory M1 phenotypic state, which are responsible for the clearance of the inflammation site from pathogens, dead neutrophils and dead tissue. IL-4, IL-13, IL-10, and TGF-β induce the M2 anti-inflammatory phenotype to encourage would healing, stimulate fibroblast migration, proliferation and angiogenesis. In chronic inflammation, macrophages retain their pro-inflammatory phenotype, resulting in persistent inflammation, impeding tissue repair. Macrophages are a consistent source of MMP-2 and MMP-9 [70,71]. However, unlike neutrophil-released MMP-9, macrophage-produced MMP-9 is bound by TIMP1, which limits MMP-9 activity [71]. Therefore, macrophages have an intrinsic modulatory influence on MMPs function, which is important in the remodeling phase. In chronic inflammation, macrophages retain their pro-inflammatory phenotype, resulting in persistent inflammation, impeding tissue repair and chronicitation [72]”;
Line 798: Macrophages polarized towards the M1 phenotypic state are associated with tumour-suppressive functions such as supporting CD8+ [196] cytotoxic T-cell activity. Macrophages polarised towards the M2 state are anti-inflammatory and associated with tumour-promoting functions such as ECM remodelling, angiogenesis, stimulating cancer cell proliferation and metastasis. Indeed, in the later process of inflammation, macrophages are favourable for tumour progression and often contribute to the process. Monocyte-derived macrophages within the tumor stroma which differentiate in M2 phenotype is indicated as tumor-associated (TAMs). These cells release in tumor stroma several cytokines and interleukins (ILs), among which IL-10 that inhibits the expression of MHC and co-stimulatory molecules inducing immune suppression and TGF-β [195]. This latter one, instead, attracts regulatory T-cells (Tregs) and other cells of adaptive immune system such as the myeloid suppressor cells which collaborates to inhibit the attack of CD8+ -T-cells and natural killer (NK) cells to the tumor mass [197,198]. Tregs, in turn, secrete TGF-β support the activation of ECM-tethered TGF-β, thus reinforcing the tumor supportive action [199]. The role of TAMs is well investigated for their interaction with CAFs. TAM and CAFs can activate and recruit each other in a similar manner to that observed in normal macrophages and tissue fibroblasts. Indeed, CAFs secrete bFGF inducing M2 polarization while M2 cells release TGF-β provoking fibroblast reprogramming to a tumor-promoting CAF state [72]. It has been demonstrating in an orthotopic CRC model, that recruitment of TAMs to the tumour site regulated several ECM-associated genes which established key characteristics of the TME. Many of these ECM components were expressed by the TAMs directly, while many others were produced by CAFs under the influence of TAMs”.
- Typo mistake in line 25, 126.
We apologize for the mistake, we have corrected and wrote “wound” in line 117.

Reviewer 4 Report
Summary
The authors provide a review about the extracellular matrix (ECM) and its role in the relationship between inflammatory and tumour microenvironments. In particular, the authors provide a distinctive viewpoint through their consideration of the use of decellularized ECM (dECM) models.
In this reviewers’ opinion the topic being covered is interesting, important and relevant to the cancer field in general. The authors are well positioned to provide this review, with the lead author’s laboratory having published in this field for numerous years.
However, there are some issues that I feel need to be addressed before publication.
Primary Issues:
1.1. For this reviewer there appeared to be a lack of connection between the larger coverage of the role of the ECM in the TME provided (section 2) and shorter coverage of the role in IME (sections 3 and 4) and dECM model section (4.3). Unless my understanding was wrong, this was the whole premise of the review i.e. to provide a comparison between the role of the ECM in TME and IME? This reviewer has some expertise, and a decent understanding of the role of ECM in the TME, but I struggled to relate the information provided about the IME to the TME.
2. For section 4.3 about the dECM a much more detailed introduction to the relevance of this area is required. The information needs to be discussed in relation to the sections about TME/IME above otherwise why include the information?
2. The scope of the review is perhaps too ambitious. You begin with and introduction (section 1) which provides historical background but was perhaps too long, and then moved on to the role of the ECM in the TME (section 2 – perhaps headed a bit misleadingly ‘ECM features under pathological conditions’). Much of what was covered in these sections has been featured in many excellent reviews before. Unless the authors wish to provide this information over again, I feel it would be better to condense these sections and provide more details regarding the IME/TME relationship.
3. The role of forces and biomechanics is a theme that runs through the whole review but how is it different in TME/IME?
Other issues:
51. In general there are many long paragraphs e.g. 1.1. Please shorten and break up the text more where appropriate.
6. 2. Typo line 270 ‘DDrs’ not DDRs
7. 3. Line 298. Is the term ‘reciprocity’ the most appropriate to describe ECM turnover and reorganisation?
8. 4. Lines 324-325: ‘These cells, once activated in myofibroblasts,’ should perhaps be ‘activated to become’?
9. 5. It is not clear how the meaning of section ‘2.2. Tumorigenic ECM deposition’ covers different aspects to that of section ‘2.3. Alterations in ECM composition’?
1 6. Could the authors provide some additional informative figures? At present the review is very text heavy.
1 7. Lines 526-532: The text in the introduction section 3 requires editing for clarity of meaning.
. 8. eading ‘4.2. Main inflammatory pathways correlated to tumorigenesis and colitis associated cancer example’ requires revision to aid readability
1 9. able 1. Expand legend to provide full details of what is shown. How is the table organised? Are the entries grouped in a specific way? Disease or date? A logical order will help the reader to extract the relevant information.
Author Response
Response to Reviewer 4 Comments
Oct 28th, 2022
Dear Editor,
we kindly thank to all the reviewers for the brilliant suggestions.
We carefully considered all the comments by the Reviewers and we revised the manuscript accordingly. We feel that the manuscript has been significantly improved thanks to the Reviewers’ comments and we hope the paper will now be considered acceptable for publication. In accordance with Reviewer’s requests, we decided to make extensive changes in manuscript’s schedule. To make more readable and to streamline the text, we decided to modify the structure of the review as follows:
1.Introduction
1.1. ECM More Than A Physical Support: An Overview
1.2. ECM Major Components, Properties and Functions
1.3. ECM-Cells Interplay
1.4 ECM deposition
1.5 Degradation of ECM and MMPs activity
1.6 ECM dynamics under pathological conditions
- Role of ECM in IME
2.2 Immune cells components drive ECM remodeling during inflammation2.3 ECM-derived fragments as modulators of chemotactic activity for immune cells
2.4 Alteration in ECM components affect immune cells activity.
2.5 ECM-fragments influence gene expression of inflammatory cells.
- From chronic inflammation to cancer: the main correlations
3.1. Chronic inflammation and tumor initiation
3.2. Main inflammatory pathways correlated to tumorigenesis and colitis associated cancer
3.3 ECM deposition in TME
3.4 Tumorigenic Alterations in ECM composition
3.4.1 MMPs role in shaping TME
3.5 Tumorigenic ECM and its remodeling influence immune cells within tumor mass.
- Pre-clinical models application in TME and IME
4.1 Scaffold-free and Scaffold-based systems
4.2 Patient-derived scaffold obtained by tissue decellularization
4.1 3D patient-derived dECM models: novel strategies to study IME to prevent TME?
- Conclusions and Future Perspectives
Further, following the Reviewer’s suggestions we provided table 1 for ECM components properties and functions, table 2 for stromal cells description. In conclusion, as cleverly suggested, we updated the table 3 for 3D repopulated decellularized ECM models and we added an abbreviations list.
Summary
The authors provide a review about the extracellular matrix (ECM) and its role in the relationship between inflammatory and tumour microenvironments. In particular, the authors provide a distinctive viewpoint through their consideration of the use of decellularized ECM (dECM) models.
In this reviewers’ opinion the topic being covered is interesting, important and relevant to the cancer field in general. The authors are well positioned to provide this review, with the lead author’s laboratory having published in this field for numerous years.
However, there are some issues that I feel need to be addressed before publication.
Primary Issues:
1.1. For this reviewer there appeared to be a lack of connection between the larger coverage of the role of the ECM in the TME provided (section 2) and shorter coverage of the role in IME (sections 3 and 4) and dECM model section (4.3). Unless my understanding was wrong, this was the whole premise of the review i.e. to provide a comparison between the role of the ECM in TME and IME? This reviewer has some expertise, and a decent understanding of the role of ECM in the TME, but I struggled to relate the information provided about the IME to the TME.
We kindly thank the reviewer for raising these reasonable doubts. Following reviewers’ comments we decided, as previously announced in the preface, to pose the IME section, after the description of ECM in physiologic condition and before the TME section. We decided these changes hoping to focus more attention on the IME and to improve the chronological order of events. Further, we enrich the relative Section 2 adding more information about the immune microenvironment in IME and ECM components interactions, and the immune cells components in ECM remodeling during inflammation. We modified text as follows:
Line 338: “The immune microenvironment is strongly influenced by ECM features and many immune cells type have key role in ECM deposition and remodeling process, in homeostasis as well as in pathological condition.
Inflammation is a multifactorial network of chemical signals which initiate and maintain a host response aimed to heal the injured tissue. Inflammation process involves primary cell players such as neutrophils, monocyte-macrophage, mast cells (MCs) and T lymphocytes also produce cytokines, chemokines, growth factors and proteases which support proliferation of epithelial cells as well as production and remodeling of ECM by fibroblasts [60].
Fibroblasts have key role in restoring the ECM homeostasis during inflammation and have also a crucial crosstalk with the immune system. The outcome inflammation process is related to the tightly regulated interaction between all cell type involved as well as with the ECM. Despite the process through which immune cells mediate the resolution of inflammation is tightly regulated and self-limiting this can be hampered leading to abnormalities. Indeed, the profile of cytokines and chemokines persisting at the inflammatory site is decisive for the development of chronic disease which provoke ECM alteration [14]. In inflamed tissue infiltrating cells release cytokines, such as tumor necrosis factor α (TNF-α), interferon gamma (IFNγ) and TGFβ, which influence both ECM turnover and protease secretion by tissue-resident cells, thus modulating the expression of a wide range of ECM molecules. On the other hand, the aberrant expression of ECM components can influence immune cell activation, differentiation, and survival. Therefore, the remodeled ECM of inflamed tissue influence the perpetuation of inflammatory response and its chronicization”.
Line 386: “Neutrophils are the first effectors during inflammation recruited by damage-associated molecular patterns (DAMPs) which can be altered by ECM component or ECM fragments as discuss above. Sustained neutrophils recruitment incites the production of pro-inflammatory cytokines and chemokines, promoting angiogenesis and degrading the ECM. In course of inflammation, neutrophil extracellular traps (NETs) are important sources for ECM-degrading enzymes such as serine proteases which function to ingest pathogens, as well as to break down ECM to aid in migration [63]. Moreover, the serine proteases can improve MMP activity cleaving the pro-form of these enzymes and triggering their catalytic activity [64][63][65]. Further, modulating pro-inflammatory cytokines and chemokines, enabling the recruitment of further leukocytes [66]. Notably, an example of exacerbation of the inflammation mediated by MMPs activity is the increase of MMP-1, MMP-8, MMP-9, MMP-10, MMP-12, MMP-13 expression within the intestine of IBD patients which are undetectable in health gut [67]. Apart from MMPs and serine activity, another discussed group of enzymes are meprins, a zinc-dependent proteases necessary which are responsible for the breakdown of ECM proteins such as type IV collagen, laminin and nidogen [67].
Proceeding along the inflammation process, once spent, neutrophils secrete many chemoattracts cytokines such as CCL-2 (MCP-1), CCL-3 (MIP1α), CCL-4 (MIP-1β), CCL-5, TSP-1, IL-1, IL-6, and TNF-α [68] [69], which recruit monocytes in situ and inducing them to differentiate in mature macrophage and dendritic cells. At the initial stage, stimulation of most macrophages mediated by IFN -γ TNF -α and Granulocyte-Macrophage Colony-Stimulating Factor (GM-CSF) are polarized toward a pro-inflammatory M1 phenotypic state, which are responsible for the clearance of the inflammation site from pathogens, dead neutrophils and dead tissue. IL-4, IL-13, IL-10, and TGF-β induce the M2 anti-inflammatory phenotype to encourage would healing, stimulate fibroblast migration, proliferation and angiogenesis. In chronic inflammation, macrophages retain their pro-inflammatory phenotype, resulting in persistent inflammation, impeding tissue repair.
Macrophages are a consistent source of MMP-2 and MMP-9 [70,71]. However, unlike neutrophil-released MMP-9, macrophage-produced MMP-9 is bound by TIMP1, which limits MMP-9 activity [71]. Therefore, macrophages have an intrinsic modulatory influence on MMPs function, which is important in remodeling phase. In chronic inflammation, macrophages retain their pro-inflammatory phenotype, resulting in persistent inflammation, impeding tissue repair and chronicitation [72]. MCs are another key effector of innate immune system which are present in almost all tissue where defend the organism against pathogens. Once activated, MCs degranulate and release pre-made chemokines, cytokines, growth factors, histamine, and proteases.
MCs exert a fundamental role in ECM scar formation through proteases release [73], and a protective role through pro-inflammatory cytokines which recruit of immune cells [74].
MCs have roles in ECM production, processing, and degradation, since MCs proteases can degrade collagen [75], FN [76], laminin [73] and activate the latent forms of MMP-2 and MMP-9 [77]. Further, as reported by Mertz and colleagues in 2007 a wide range of possible functions were identified for MCs in promoting (or suppressing) many features of chronic inflammation [74]”.
1.2 For section 4.3 about the dECM a much more detailed introduction to the relevance of this area is required. The information needs to be discussed in relation to the sections about TME/IME above otherwise why include the information?
Following the reviewer's comment, we decided to add a description regarding the decellularization in the text as follows:
Line 939: “Recently, tissue decellularization has emerged as a robust alternative technique in the field of tissue engineering and regenerative medicine [229]. The principal advantage of using biological-derived matrices, instead of synthetic polymers for 3D tumor study in vitro, is that the main structural proteins and soluble factors are already present in decellularized scaffolds [230]. The term decellularization indicates the removal of the cellular component from a tissue by minimally altering its biochemical composition and biological and structural properties [231]. The decellularization of ECM can be achieved by three different approaches including enzymatic, chemical and physical methods. Chemical methods include the perfusion, agitation or immersion with chemical solutions. These methods achieve removal of the majority of cellular materials but fail to preserve the ECM molecules. That is why, the decellularization process with combined chemical treatments should be optimized for the least damage to ECMs [229]. Physical methods include freezing, direct pressure, osmosis, sonication and agitation. They can be used to decellularized tissues by disrupting cellular membranes, with corresponding cell lysis. Physical prevent the disruption of ECM structures, but since these techniques only disintegrate cellular membranes, but do not remove cellular debris, requiring the adding of enzymatic or chemical treatments to obtain acellular tissues [232]. Enzymatic decellularization uses nucleases, proteases, and chelating agents (necessary for the removal or separation of cellular debris). Enzymatic approaches can effectively remove cellular materials preserving most of the collagen components. However, it is necessary to set a maximum treatment time to mitigate the disruptive effects on the architecture and composition of the ECM during the decellularization process [232].”
- The scope of the review is perhaps too ambitious. You begin with and introduction (section 1) which provides historical background but was perhaps too long, and then moved on to the role of the ECM in the TME (section 2 – perhaps headed a bit misleadingly ‘ECM features under pathological conditions’). Much of what was covered in these sections has been featured in many excellent reviews before. Unless the authors wish to provide this information over again, I feel it would be better to condense these sections and provide more details regarding the IME/TME relationship.
We are grateful to the reviewer for the suggestion. On this purpose we decide to summarize the first section of the ECM components in Table 1 as follows:
Components |
Functions and Properties |
Proteins and Structure |
|
Collagens. Formed as fibrils within the ECM (Collagen I, II, III, V and XI) |
ECM architecture, mechanical properties [20]. Influence cell process such as adhesion and migration. Binding of extracellular growth factors and cytokine [21]. |
Elastin Composed of single tropoelastin subunit cross-linked |
Tissue elasticity, load bearing and storage of mechanical energy [22]. |
Glycoproteins |
|
Laminins Trimeric structure made of three different chain α, β, γ |
Resident in the basement membrane. It play a role in cell adhesion, survival, migration, differentiation and migration via integrin [22]. |
Fibronectin (FN) Arranged into a mesh of fibrils to collagen and is linked to cell surface receptors (integrins) |
Binds to collagen, fibrin and glycosaminoglycans, influencing cell adhesion, growth, migration, wound healing and differentiation [23]. |
Fibrillins |
Scaffolds for elastin deposition [23]. |
GAGs |
|
Hyaluronic acid (HA) |
Regulates numerous cell functions and biological processes through the interaction with cell surface receptors. Lend tissue turgor and facilitates cell migration during tissue morphogenesis and repair. Intact high molecular weight of HA has antiangiogenic and anti-inflammatory properties, whereas lower HA fragments exhibit completely opposite functions [24]. |
PGs |
|
e.g Heparan, chondroitin and keratin sulphates. Core of protein domain covalently linked to GAGs. Negavitely charged it can bind water. |
Provide compressive resistance to tissue. Reservoir and sequester of growth factors, chemokines and cytokines at the ECM protecting them from degradation and creating effective gradients along ECM [25]. |
Matricellular proteins |
|
Thrombospondins (TSPs), tenascins (TNs), fibulins, osteopontin (OPN), cartilage oligomeric matrix protein (COMP), CNN family proteins, periostin, and R-spondins |
Cell behaviour modulation, mediation of the interaction between cellular transmembrane receptors and fibrous ECM molecules and key functional regulator of cell-matrix communication [26]. |
In addition, as suggested, we enrich the section 2 adding information about the immune microenvironment first and then the immune cells and their contribution to EMC remodeling in text as follows:
Line 340: The immune microenvironment is strongly influenced by ECM features and many immune cells type have key role in ECM deposition and remodeling process, in homeostasis as well as in pathological condition. Inflammation is a multifactorial network of chemical signals which initiate and maintain a host response aimed to heal the injured tissue. Inflammation process involves primary cell players such as neutrophils, monocyte-macrophage, mast cells (MCs) and T lymphocytes also produce cytokines, chemokines, growth factors and proteases which support proliferation of epithelial cells as well as production and remodeling of ECM by fibroblasts [60]. Fibroblasts have key role in restoring the ECM homeostasis during inflammation and have also a crucial crosstalk with the immune system. The outcome inflammation process is related to the tightly regulated interaction between all cell type involved as well as with the ECM. Despite the process through which immune cells mediate the resolution of inflammation is tightly regulated and self-limiting this can be hampered leading to abnormalities. Indeed, the profile of cytokines and chemokines persisting at the inflammatory site is decisive for the development of chronic disease which provoke ECM alteration [14]. In inflamed tissue infiltrating cells release cytokines, such as tumor necrosis factor α (TNF-α), interferon gamma (IFNγ) and TGFβ, which influence both ECM turnover and protease secretion by tissue-resident cells, thus modulating the expression of a wide range of ECM molecules. On the other hand, the aberrant expression of ECM components can influence immune cell activation, differentiation, and survival. Therefore, the remodeled ECM of inflamed tissue influence the perpetuation of inflammatory response and its chronicization”.
Line 386: Neutrophils are the first effectors during inflammation recruited by damage-associated molecular patterns (DAMPs) which can be altered by ECM component or ECM fragments as discuss above. Sustained neutrophils recruitment provokes the production of pro-inflammatory cytokines and chemokines, promoting angiogenesis and degrading the ECM. In course of inflammation, neutrophil extracellular traps (NETs) are important sources for ECM-degrading enzymes such as serine proteases which function to ingest pathogens, as well as to break down ECM to aid in migration [63] . Moreover, the serine proteases can improve MMP activity cleaving the pro-form of these enzymes and triggering their catalytic activity [64][63][65]. Further, modulating proinflammatory cytokines and chemokines, enabling the recruitment of further leukocytes [66]. Notably, an example of exacerbation of the inflammation mediated by MMPs activity is the increase of MMP-1, MMP-8, MMP-9, MMP-10, MMP-12, MMP-13 expression within the intestine of IBD patients which are undetectable in health gut [67]. Apart from MMPs and serine activity, another discussed group of enzymes are meprins, a zinc-dependent proteases necessary which are responsible for the breakdown of ECM proteins such as type IV collagen, laminin and nidogen [67].
Proceeding along the inflammation process , once spent, neutrophils secrete many chemoattracts cytokines, e.g. CCL-2 (MCP-1), CCL-3 (MIP1α), CCL-4 (MIP-1β), CCL-5, TSP-1, IL-1, IL-6, and TNF-α [68] [69], which recruit monocyte in situ and inducing them to differentiate in mature macrophage and dendritic cells. At the initial stage, stimulation of most macrophages mediated by IFN -γ TNF -α and Granulocyte-Macrophage Colony-Stimulating Factor (GM-CSF) are polarized toward a pro-inflammatory M1 phenotypic state, which are responsible for the clearance of the inflammation site from pathogens, dead neutrophils and dead tissue. IL-4, IL-13, IL-10, and TGF-β induce the M2 anti-inflammatory phenotype to encourage would healing, stimulate fibroblast migration, proliferation and angiogenesis. In chronic inflammation, macrophages retain their pro-inflammatory phenotype, resulting in persistent inflammation, impeding tissue repair.
Macrophages are a consistent source of MMP-2 and MMP-9 [70,71]. However, unlike neutrophil-released MMP-9, macrophage-produced MMP-9 is bound by TIMP1, which limits MMP-9 activity [71]. Therefore, macrophages have an intrinsic modulatory influence on MMPs function, which is important in remodeling phase. In chronic inflammation, macrophages retain their pro-inflammatory phenotype, resulting in persistent inflammation, impeding tissue repair and chronicitation [72]. MCs are another key effector of innate immune system which are present in almost all tissue where defend the organism against pathogens. Once activated, MCs degranulate and release pre-made chemokines, cytokines, growth factors, histamine, and proteases.
MCs exert a fundamental role in ECM scar formation through proteases release [73], and a protective role through pro-inflammatory cytokines which recruit of immune cells [74].
MCs have roles in ECM production, processing, and degradation, since MCs proteases can degrade collagen [75], FN [76], laminin [73] and activate the latent forms of MMP-2 and MMP-9 [77]. Further, as Mertz at in a study of 2007 have suggested a wide range of possible functions for MCs in promoting (or suppressing) many features of chronic inflammation [74]”.
- The role of forces and biomechanics is a theme that runs through the whole review but how is it different in TME/IME?
We thank for this interest point regarding the role of forces and biomechanics in different contexts. Dissimilar from the well-maintained tissue homeostasis and mechanical equilibrium in normal physiological conditions, the pre-neoplastic inflammatory state and the loss of growth control and the disrupted tissue organization during tumor evolution lead to unbalanced physical forces. Because the behavior, structure and organization of tissue in inflammatory state and tumors are continuously changing, the mechanical state also evolves as the tumor develops. The mechanical characteristics during tumorigenesis (pre-tumor niche, hyperplasia, carcinoma in situ, invasive lesion and metastasis lesion) can be very different. An initial dysplastic lesion typically involves the loss of normal cell polarization and organization, the changes in cell–cell contacts and cell–ECM interactions, which result in altered cellular tension and mechano-sensing and transduction. Increased matrix deposition, cell proliferation and altered cell tension result in the thickening and remodeling of the basement membrane architecture. Furthermore, inefficient transport and dense ECM networks result in further increases in interstitial fluid pressure within the tumor. During tumor growth various mechanical forces, such as interstitial compression and shear, interstitial fluidic pressure and ECM stiffness can also critically influence the rate and direction of tumor cell migration. We strongly believe that the role of forces and biomechanics are fundamental points of study both in pre-tumor condition, such as in inflammatory disease, and in tumor condition. The mechanical microenvironment may promote tumour progression by influencing processes such as epithelial-to-mesenchymal transition, enhancing cell survival through autophagy, but also affects sensitivity of neoplastic cells to therapeutics. Overall, TME mechanics are significantly different from IME mechanics and from normal physiological tissue. Therefore, we think that this feature should be further explored for use in inflammatory disease and cancer prevention, detection and also treatment.
Other issues:
In general there are many long paragraphs e.g. 1.1. Please shorten and break up the text more where appropriate.
We completely agree with the reviewer's comment and on this purpose, we have decided to summarize the main components, functions and properties of the ECM in Table 1 as follows:
Components |
Functions and Properties |
Proteins and Structure |
|
Collagens. Formed as fibrils within the ECM (Collagen I, II, III, V and XI) |
ECM architecture, mechanical properties [20]. Influence cell process such as adhesion and migration. Binding of extracellular growth factors and cytokine [21]. |
Elastin Composed of single tropoelastin subunit cross-linked |
Tissue elasticity, load bearing and storage of mechanical energy [22]. |
Glycoproteins |
|
Laminins Trimeric structure made of three different chain α, β, γ |
Resident in the basement membrane. It play a role in cell adhesion, survival, migration, differentiation and migration via integrin [22]. |
Fibronectin (FN) Arranged into a mesh of fibrils to collagen and is linked to cell surface receptors (integrins) |
Binds to collagen, fibrin and glycosaminoglycans, influencing cell adhesion, growth, migration, wound healing and differentiation [23]. |
Fibrillins |
Scaffolds for elastin deposition [23]. |
GAGs |
|
Hyaluronic acid (HA) |
Regulates numerous cell functions and biological processes through the interaction with cell surface receptors. Lend tissue turgor and facilitates cell migration during tissue morphogenesis and repair. Intact high molecular weight of HA has antiangiogenic and anti-inflammatory properties, whereas lower HA fragments exhibit completely opposite functions [24]. |
PGs |
|
e.g Heparan, chondroitin and keratin sulphates. Core of protein domain covalently linked to GAGs. Negavitely charged it can bind water. |
Provide compressive resistance to tissue. Reservoir and sequester of growth factors, chemokines and cytokines at the ECM protecting them from degradation and creating effective gradients along ECM [25]. |
Matricellular proteins |
|
Thrombospondins (TSPs), tenascins (TNs), fibulins, osteopontin (OPN), cartilage oligomeric matrix protein (COMP), CNN family proteins, periostin, and R-spondins |
Cell behaviour modulation, mediation of the interaction between cellular transmembrane receptors and fibrous ECM molecules and key functional regulator of cell-matrix communication [26]. |
6.2. Typo line 270 ‘DDrs’ not DDRs
We apologise for the mistake, we have corrected and wrote “DDrs” in lines 210 and 212.
7.3. Line 298. Is the term ‘reciprocity’ the most appropriate to describe ECM turnover and reorganisation?
We thank the review for raising this doubt.
We used this term at line 325 took cue from the reference “Lu P, Takai K, Weaver VM, Werb Z. Extracellular matrix degradation and remodeling in development and disease. (Cold Spring Harb Perspect Biol. 2011 Dec 1;3(12):a005058. doi: 10.1101/cshperspect.a005058. PMID: 21917992; PMCID: PMC3225943”. In this paper cell–ECM interactions are described as reciprocal. Further following reference “Thorne JT, Segal TR, Chang S, Jorge S, Segars JH, Leppert PC. Dynamic reciprocity between cells and their microenvironment in reproduction. Biol Reprod. 2015 Jan;92(1):25. doi: 10.1095/biolreprod.114.121368. Epub 2014 Nov 19. PMID: 25411389; PMCID: PMC4434933” refers to bidirectional interaction between cells and the ECM using the terms “dynamic reciprocity”.
8.4. Lines 324-325: ‘These cells, once activated in myofibroblasts,’ should perhaps be ‘activated to become’?
We kindly thank the reviewer about our imprecision. We modify text as follows:
Line 240: “These cells, once activated become myofibroblasts”.
9.5. It is not clear how the meaning of section ‘2.2. Tumorigenic ECM deposition’ covers different aspects to that of section ‘2.3. Alterations in ECM composition’?
We are so grateful to the reviewer to give us this point of reflection. We decided for a new paragraphs sequence and deleting the paragraph named “Alteration in ECM composition” keeping separated the description about the alterations of ECM in IME and then in TME.
1 6. Could the authors provide some additional informative figures? At present the review is very text heavy.
Following the reviewer’s comment, we decided to recapitulate paragraph 1.2 in Table 1 and, further we summarized the stromal cells functions and properties in Table 2 as follows:
Cancer associated Endothelial cells
|
-Releasing of proangiogenic factor (PDGF-EC), EGF and VEGF under hypoxic conditions. -Promoting cancer cell migration, invasion and metastasis. -Undergoing to “Endothelial–mesenchymal transition” become cancer-associated fibroblasts organized by TGF-b and bone morphogenetic protein (BMP) that leads to loss of cell to cell connections, detachment and elongation, enhanced migration and loss of endothelial properties [156]. |
CAFs
|
-Mediate the crosstalk between cancer cells and TME -Release of growth factors, cytokines to attract other cell (e.g. endothelial cell and imme cells) to joint TME and ECM components deposition -Release of MMP-3 for E-caderin degradation To facilitate migration of cancer cells through theTME, inducion of EMT and promotion of cell invasion -Promotion an immunosuppressive phenotype through the production of immune-modulatory chemokines and cytokines [157]. |
Cancer-associated adipocytes
|
-Secretion of metabolites, enzymes, hormones, growth factors and cytokines involved in matrix remodelling, invasion and survival of cancer cells, [158]. (e.g. producion of insulin-like growth fator biding protein-2, MMP-11 and IL-6 and Il-1β increase migration and metastasis in human breast cancer cell in vitro and in vivo. [159]. |
Tumor associated immune stroma
|
-Tumor associated macrophages (TAMs) promote tumor cell migration and invasion [160]. -Myeloid-derived suppresso cells differentiate into TAMs and dendritic cells, contribute to tumor progression though tumor imme invasion, ECM remodelling and promoting EMT [161]. -Natural killer cells distinct I two subpopulation tumor-infiltrating natural killer cells (TINKs) and tumor-associated natural killer cells (TANKs) [162,163] and secrete altered levels of cytokines (e.g. pro-angiogenic factor such as VEGF and stromal derived factor-1) promoting angiogenesis and tumor progression [164]. -Regulatory T cells (Tregs) promote infiltration of tumorigenesis trhough the reduction of antitnumor immune response [165]. -Mast cell; anti-tumorigenic activity: through degranulation process becam hightly proinflammatory and actively recuit cells of innate immune system to orchestrate antitumor imune responses [166]. Protumor activity: relasing of factors (e.g. VEGF) to support angiogenesi and MMP9 to degrade ECM and facilitate the metastasis. [166] |
1 7. Lines 526-532: The text in the introduction section 3 requires editing for clarity of meaning.
We thank the reviewer for the suggestion and we modified the introduction of the “Role of ECM in IME” in the text as follows:
Line 335 : “Since we proceed to describe the role of ECM in IME and then in TME, a brief overview of the immune microenvironment and its interaction with the ECM is need”.
1 8. ‘4.2. Main inflammatory pathways correlated to tumorigenesis and colitis associated cancer example’ requires revision to aid readability
We thank reviewer for the suggestion and we modified the text as follows:
Line 571: “ Once activated in inflammatory cells, NF-kB regulates cell cycle mediators (cyclin D1, c-Myc), anti-apoptotic (c-FLIP, survivin, Bcl-XL) and adhesion molecules (ICAM-1, ELAM-1, VCAM-17), proteolytic enzymes such as MMP-9 and uPA and pro-inflammatory cytokines (such as TNF-α, IL-1, and IL-6) which, all together, contribute to inflammation-related tissue damage, tumor development and progression”;
Lines 591: “In IBD activation of STAT3, mediated through the interaction of the complex of IL-6 with its membrane bound receptor (IL6R), enhances the expression of antiapoptotic factors, causing CD4 T cell resistance and thus promotes the perpetuation of chronic intestinal inflammation”;
Lines 593: “As demonstrate in colon cancer cell line CACO-2, programmed to constitutively express COX-2, were detected an increased invasiveness if compared with the parental CACO-2, along with an activation of MMP-2 and increased RNA levels for MT-MMP-1”.
- Table 1. Expand legend to provide full details of what is shown. How is the table organised? Are the entries grouped in a specific way? Disease or date? A logical order will help the reader to extract the relevant information.
Following the reviewer's comment, we decided to update Table 3, description and organization, based on the tumor type to help the reader. We have indicated the type of cell lines used in each article and the respective abbreviations as follows:
dECM origin |
dECM tissue |
Cells cultured on dECM |
Results |
Ref. |
Human |
CRC |
Human colon adenocarcinoma cells
(HT-29 – HCT-116) |
-3D CRC model exhibited reduced sensitivity to 5-Fluoruracil and FOLFIRI treatments compared with 2D conventional cell cultures. |
[13] |
Human |
HC - CRC – Healthy Liver (HL) – CRC and liver metastasis (CRLM) |
Human colon adenocarcinoma cells (HT-29 ZsGreen/Luc+ and HCT-116 cells)
|
-Increased migration ability of CRC cells observed in CRC and CRLM scaffolds if compared to HL scaffolds. -At an increasing concentration of drug, the response of the CRC cell line to chemotherapy was scaffold-dependent. HT-29 cells grown in HL scaffolds exhibited a significant reduction in cell proliferation compared to untreated scaffolds. |
[238] |
Human |
CRC |
Human colon adenocarcinoma cells (HCT-116 and HT-29) |
-Tumor cells and the decellularized tumor matrix induce the differentiation of monocytes in macrophages, characterized by high surface expression of CD206 and reduced expression of MHC-II and CD86. -The exposition of macrophage to tumor cells and to tumor ECM induce a higher expression of IL-6, IL-10, TGF- CCL17, CCL18, and CCL22. |
[239] |
Human |
Healthy colon and CRC |
Human colon adenocarcinoma cells (HT-29 cells) |
-Over-expression of IL-8 in 3D tumor matrices after 5 days of recellularization with HT-29 cells. -Secretoma analysis showed a marked increase of DEFA3 in tumor stroma compared to healthy counterpart. |
[240] |
Human |
PDAC |
Human pancreatic adenocarcinoma cell lines (Panc-1 and AsPC-1) |
-Immunofluorescence analyses after 7 days of scaffolds recellularization with Panc-1 and AspC-1, confirmed the biocompatibility of 3D matrices to sustain engraftment, localization and infiltration. -Both PANC-1 and AsPC-1 cells cultured in 3D matrices showed a reduced response to treatment with FOLFIRINOX if compared to conventional bi-dimensional culture. |
[241] |
Human |
Esophagus from cadavers |
Human bone marrow cell line (HM1-SV40) |
- After the 72 hours of exposure period to culture media conditioned with decellularized esophageal samples, HM1-SV40 cells appeared to be viable and proliferating, reaching about 90% confluence on the growth surface. The MTT assay revealed that, compared with untreated cultures, cells preserved 93.7%, 95.4% and 78.3% viability when treated with medium conditioned with esophageal matrices decellularized through Protocols No. 1, No. 2 and No. 3, respectively). |
[242] |
Rat |
Esophageal adenocarcinoma (EAC) |
Human mononuclear cells (THP-1) and oesophageal eipithelial cells (Het-1A) |
-Metaplastic and neoplastic esophageal ECM induce distinctive effects upon THP-1 macrophage signaling compared to normal esophageal ECM. -The secretome of macrophages pre-treated with metaplastic and neoplastic ECM increases the migration of normal esophageal epithelial cells, similar behavior to that shown by tumor cells. |
[243] |
Pig |
Breast cancer |
Human BC cell line (MDA-MB- 231) 3D tumor spheroids Human dermal fibroblasts |
- Metastatic breast cancer model which constitute a TME-mimetic platform comprising breast tissue-derived porcine d ECM; cancer cell assembled to form spheroids. In vitro generated monotypic and heterotypic dECM-Spheroids model exhibited the invasive profile of metastatic breast cancer cells and evidenced exometabolomic signatures similar to those observed for human tumor translated in differences to cisplatin, gemcitabine and Palbociclib cancer-cell specific response. |
[244] |
Human |
Breast Cancer |
Human breast cancer cells (MCF7, T‐47D, and MDA‐MB‐231) |
Cells cultured in patient-derived scaffolds showed a higher drug resistance if compared to the 2D cultures, demonstrating that the 3D environment improves cell chemoresistance. |
[245] |
Human and Animal |
Different Tissue/organ-derived |
Human cancer cells, human monocytes and fibroblasts. |
-dECM will provide new insights into the role of ECM in oncogenesis and cancer progression. -dECM represents a culture substrate for pharmacological and pharmacokinetic analyses to develop novel anticancer drugs. -proteomic analysis of dECM will lead to the identification of candidates for anticancer drugs. |
[246] |
Human and murine
|
Normal and Tumor biopsies |
Fibroblastic cell derived matrices (fCDM)
|
-The method offers the possibility to obtain 3D matrices with relevant characteristics for mimicking particular in vivo microenvironments (e.g. normal vs. desmoplastic or fibrous). -fCDMs can be used to scrutinize cocultures of different cell types (tumor, stroma, immune neural etc.) that co-exist in the interstitial microenvironment of diverse tissues. -fCDM have shown to serve as a useful platform to establish pre-clinical studies. |
[247] |

Reviewer 5 Report
The paper is interesting and valuable, but some description or discussion is poor. Taken together, major revisions should be made before re-submission. The paper would be re-considered only when all the comments were responded.
1. New section
Biomaterials for 3D cancer models should be introduced. Biomaterials composed of ECM components can be useful to enhance cell activity and establish 3D models. Because the title includes the ECM, the section is essential. To reduce the authors’ burden, I suggest at least these references to be added for the revision.
Review (for concept)
Cancers 2020, 12(10), 2754
Research papers
Alginate Biomaterials 55 (2015) 110-118
Chitosan Biomaterials 25 (2004) 5147–5154
Gelatin Tissue Eng. Part C Methods 2019, 25, 711–720 https://doi.org/10.1089/ten.tec.2019.0189
Collagen doi.org/10.1016/j.actbio.2018.06.003
Hyaluronic acid Adv. Healthcare Mater.2015, 4, 1664–1674
2. New table
The stromal cells in TME should be added.
3. New table
ECM and the properties in TME should be added.
4. Future perspectives
More detailed sentences should be added.
Author Response
Response to Reviewer 5 Comments
Oct 28th, 2022
Dear Editor,
we kindly thank to all the reviewers for the brilliant suggestions.
We carefully considered all the comments by the Reviewers and we revised the manuscript accordingly. We feel that the manuscript has been significantly improved thanks to the Reviewers’ comments and we hope the paper will now be considered acceptable for publication. In accordance with Reviewer’s requests, we decided to make extensive changes in manuscript’s schedule. To make more readable and to streamline the text, we decided to modify the structure of the review as follows:
1.Introduction
1.1. ECM More Than A Physical Support: An Overview
1.2. ECM Major Components, Properties and Functions
1.3. ECM-Cells Interplay
1.4 ECM deposition
1.5 Degradation of ECM and MMPs activity
1.6 ECM dynamics under pathological conditions
- Role of ECM in IME
2.2 Immune cells components drive ECM remodeling during inflammation2.3 ECM-derived fragments as modulators of chemotactic activity for immune cells
2.4 Alteration in ECM components affect immune cells activity.
2.5 ECM-fragments influence gene expression of inflammatory cells.
- From chronic inflammation to cancer: the main correlations
3.1. Chronic inflammation and tumor initiation
3.2. Main inflammatory pathways correlated to tumorigenesis and colitis associated cancer
3.3 ECM deposition in TME
3.4 Tumorigenic Alterations in ECM composition
3.4.1 MMPs role in shaping TME
3.5 Tumorigenic ECM and its remodeling influence immune cells within tumor mass.
- Pre-clinical models application in TME and IME
4.1 Scaffold-free and Scaffold-based systems
4.2 Patient-derived scaffold obtained by tissue decellularization
4.1 3D patient-derived dECM models: novel strategies to study IME to prevent TME?
- Conclusions and Future Perspectives
Further, following the Reviewer’s suggestions we provided table 1 for ECM components properties and functions, table 2 for stromal cells description. In conclusion, as cleverly suggested, we updated the table 3 for 3D repopulated decellularized ECM models and we added an abbreviations list.
Comments and Suggestions for Authors
The paper is interesting and valuable, but some description or discussion is poor. Taken together, major revisions should be made before re-submission. The paper would be re-considered only when all the comments were responded.
1.New section: Biomaterials for 3D cancer models should be introduced. Biomaterials composed of ECM components can be useful to enhance cell activity and establish 3D models. Because the title includes the ECM, the section is essential. To reduce the authors’ burden, I suggest at least these references to be added for the revision.
Review (for concept)
Cancers 2020, 12(10), 2754
Research papers
Alginate Biomaterials 55 (2015) 110-118
Chitosan Biomaterials 25 (2004) 5147–5154
Gelatin Tissue Eng. Part C Methods 2019, 25, 711–720 https://doi.org/10.1089/ten.tec.2019.0189
Collagen doi.org/10.1016/j.actbio.2018.06.003
Hyaluronic acid Adv. Healthcare Mater.2015, 4, 1664–1674
Following the reviewer's comment, we added biomaterials description in the text as follows:
line 903: “Biomaterials, which consist of ECM components, enhance the cell activity or functions. The interaction with biomaterials enables cells to enhance their proliferation, differentiation, and biological functions, leading to the realization of cancer cells–environment interaction [219]. An essential feature of biomaterials is the biocompatibility, the ability of a material to act by determining an appropriate host response to a given application. 3D scaffold must meet the following requirements: degree of porosity, enabling to constitute a percolating pattern that favours cellular growth and disposition, nutrient intake, and disposal of metabolic products [220]; also an appropriate surface characterized by adequate physical-chemical properties to promote adhesion, growth, proliferation, differentiation and migration of cells [221]. The 3D synthetic model showed a significantly higher drug resistance [222]. Despite their biocompatibility, the presence of PVDF fibers generates an artificial barrier between the pathological tissue and the surrounding environment. This contract limitation ends up excluding the flow of large molecules such as antibodies [223]. Dainiak et al. (2008) have developed microporous hydrogels (MHs), it consists of the gelation of the polyacrylamide at subzero temperatures, followed by its functionalization by covalent immobilization of type collagen I and by copolymerization of cell-adhesive peptide RGD mimetics during its production [224]. Furthermore, natural scaffolds are constituted by polymers derived from animals or plants [225]. The mainly used animal-derived polymers are collagen, fibrin, glycosaminoglycan and hyaluronic acid. Among them, gel systems, such as collagen-based ones, represent one of the earliest biomaterials and are still used nowadays since collagen is one of the most abundant components of the ECM. Magdeldin et al. (2014) developed a 3D tumor in vitro model based on the removal of interstitial fluid from collagen hydrogels with the aim to create a multiwell drug testing platform [226]. Matrigel is another largely used hydrogel (Peptide Hydrogels - Versatile Matrices for 3D Cell Culture in Cancer Medicine), it consists of a basal membrane preparation in a solution extracted from Engelbreth-Holm-Swarm (EHS) mouse sarcoma, a tumor rich in ECM proteins. The main components are laminin, collagen IV, entactin, and proteoglycan heparan-sulfate [227]. At room temperature, Matrigel polymerized to produce a biologically active matrix that represents the basal membrane of mammalian cells. One limitation of Matrigel is that once polymerized, the matrix is not suitable for long-term storage [228]. In contrast, the mainly used plant-derived polymers are methyl-cellulose, agarose and alginate [229]”.
2.New table: The stromal cells in TME should be added.
Response 2: We thank the reviewer for the interesting suggestion. We decided to add the stromal cells description in table 2 in the text as follows:
Cancer associated Endothelial cells
|
- Releasing of proangiogenic factor (PDGF), epidermal growth factor (EGF) and VEGF under hypoxic conditions. - Promoting cancer cell migration, invasion and metastasis. - Undergoing to “Endothelial–mesenchymal transition” become cancer-associated fibroblasts organized by TGF-b and bone morphogenetic protein (BMP) that leads to loss of cell to cell connections, detachment and elongation, enhanced migration and loss of endothelial properties [156]. |
CAFs
|
- Mediate the crosstalk between cancer cells and TME - Release of growth factors, cytokines to attract other cell (e.g. endothelial cell and imme cells) to joint TME and ECM components deposition - Release of MMP-3 for E-caderin degradation To facilitate migration of cancer cells through theTMEt, inducion of EMT and promotion of cell invasion -Promotion an immunosuppressive phenotype through the production of immune-modulatory chemokines and cytokines [157]. |
Cancer-associated adipocytes
|
- Secretion of metabolites, enzymes, hormones, growth factors and cytokines involved in matrix remodelling, invasion and survival of cancer cells, [158]. (e.g. producion of insulin-like growth fator biding protein-2, MMP-11 and IL-6 and Il-1β increase migration and metastasis in human breast cancer cell in vitro and in vivo. [159] |
Tumor associated immune stroma
|
- Tumor associated macrophages (TAMs) promote tumor cell migration and invasion [160]. - Myeloid-derived suppresso cells differentiate into TAMs and dendritic cells, contribute to tumor progression though tumor imme invasion, ECM remodelling and promoting EMT [161]. - Natural killer cells distinct I two subpopulation tumor-infiltrating natural killer cells (TINKs) and tumor-associated natural killer cells (TANKs) [162,163] and secrete altered levels of cytokines (e.g. pro-angiogenic factor such as VEGF and stromal derived factor-1) promoting angiogenesis and tumor progression [164]. -Regulatory T cells (Tregs) promote infiltration of tumorigenesis trhough the reduction of antitnumor immune response [165]. - Mast cell; anti-tumorigenic activity: through degranulation process becam hightly proinflammatory and actively recuit cells of innate immune system to orchestrate antitumor imune responses [166] Protumor activity: relasing of factors (e.g. VEGF) to support angiogenesi and MMP9 to degrade ECM and facilitate the metastasis. [166] |
3.New table: ECM and the properties in TME should be added.
We really thank the reviewer for the kindly and clever suggestion. Therefore, and following all the reviewers comments, to make the text more fluent, we decide to condense the description of ECM components and its properties in a new Table 1. To add a similar table for TME could be redundant. Furthermore, the aim of our review was not to provide a comparison between the qualitative and quantitative changes occurred in ECM between the physiological condition, TME and IME. Rather, we tried focus our attention about the description of the major and recognized ECM-influenced cellular components and pathways, implicated into the progression from IME toward TME. For this reason, we tried to describe this information in an extensive manner.
4.Future perspectives
Thanks to the reviewer 's suggestion we have increased the final paragraph of the Conclusions and future perspectives as below:
Line 987: “dECM models are a thriving tool in cancer research to study the complex mechanisms behind tumor invasion, progression, and response to treatment while replicating both the multicellular nature and 3D stromal environment present in vivo tumor. These models, taking advantages from the role of ECM in TME and its influence on cancer cells, can mimic the TME more closely than the conventional 2D system. Nevertheless, less attention has been given to the employment of these models for the study of conditions which could significantly increase the risk for cancer initiation such as chronic inflammation disease. When the inflammatory responses became chronic, indeed, cell mutation and proliferation can create a fertile ground conducive for cancer initiation and development. The correlation between the development of cancer from chronic condition is recognized, while the biological changes that underline this progression are not clear. Therefore, a deeper understanding of the precise mechanisms by which chronic inflammation triggers genetic alterations and predispose for cancer initiation could provide new insights for novel therapeutic strategies which could target this progressive passage to cancer. The correlation between these pathological conditions aimed us to focus our attention to the role of ECM in IME, finding in literature data could sustain the employment of dECM model even in chronic inflammation research. Our review reveals us information about the implication of ECM in IME and TME and, further, how its remodeling process may even trigger the passage from IME to TME. In the view of these insights, the data obtained in cancer research with 3D dECM are encouraging for its integration in a new field such as chronic inflammation diseases. The dECM models, indeed, could provide new insight which could be helpful to reduce the risks resulted from the chronicization of inflammation, thus, potentially, preventing the cancer initiation.”
